# A strong–weak duality for the 1d long-range Ising model

Dario Benedetti[1], Edoardo Lauria[2], Dalimil Mazáč[3,4], and Philine van Vliet[5]

[1] *CPHT, CNRS, École polytechnique, Institut Polytechnique de Paris, 91120 Palaiseau, France*

[2] *Laboratoire de Physique de l'École Normale Supérieure, Mines Paris, Inria, CNRS, ENS-PSL, Sorbonne Université, PSL Research University, Paris, France*

[3] *Institut de Physique Théorique, Université Paris-Saclay, CEA, CNRS, 91191 Gif-sur-Yvette, France*

[4] *Institut des Hautes Études Scientifiques, 91440 Bures-sur-Yvette, France*

[5] *Laboratoire de Physique, École Normale Supérieure, Université PSL, CNRS, Sorbonne Université, Université Paris Cité, 24 rue Lhomond, 75005 Paris, France*

## Abstract

We investigate the one-dimensional Ising model with long-range interactions decaying as $1/r^{1+s}$. In the critical regime, for $1/2 \leq s \leq 1$, this system realizes a family of nontrivial one-dimensional conformal field theories (CFTs), whose data vary continuously with $s$. For $s > 1$ the model has instead no phase transition at finite temperature, as in the short-range case. In the standard field-theoretic description, involving a generalized free field with quartic interactions, the critical model is weakly coupled near $s = 1/2$ but strongly coupled in the vicinity of the short-range crossover at $s = 1$. We introduce a dual formulation that becomes weakly coupled as $s \to 1$. Precisely at $s = 1$, the dual description becomes an exactly solvable conformal boundary condition of the two-dimensional free scalar. We present a detailed study of the dual model and demonstrate its effectiveness by computing perturbatively the CFT data near $s = 1$, up to next-to-next-to-leading order in $1 - s$, by two independent approaches: (i) standard renormalization of our dual field-theoretic description and (ii) the analytic conformal bootstrap. The two methods yield complete agreement.

# 1   Introduction and motivation

An important source of insights and results in quantum field theory (QFT) is provided by IR dualities, where two different UV models share the same IR behavior. If the latter is described by a conformal field theory (CFT), the UV models provide alternative ways of constructing the same CFT as an IR limit.

While closely related to the concept of universality, the term "IR duality" is commonly reserved for a more specific and surprising case of universality, where the two UV models are defined in terms of completely different degrees of freedom.

In other words, IR dualities mimic exact dualities, such as the one between the compact free boson and the free photon in 3d, or that between the sine-Gordon model and the massive Thirring model in 2d [1]. However, in contrast to these examples, they are only true up to IR-irrelevant operators [2]. Famous examples of IR dualities are the Seiberg duality [3], the particle/vortex duality [4, 5], and the "web of dualities" for Chern-Simons-matter theories [6, 7, 8].

One particularly useful feature of IR dualities is that they are often of the strong–weak type: one side is weakly coupled when the other is strongly coupled, and vice versa. A notable example, proposed in [9, 10], occurs in the context of the long-range Ising (LRI) model in $d \geq 2$ dimensions. On the $\mathbb{Z}^d$ lattice, the LRI model with long-range parameter $s$ is described by the classical Hamiltonian:

$$\beta \, \mathcal{H}_{\mathrm{LRI}} = \frac{\mathcal{J}}{2} \sum_{i \neq j} \frac{(\sigma_i - \sigma_j)^2}{|i - j|^{d+s}}, \quad \mathcal{J} > 0 \,, \tag{1.1}$$

where $\sigma_i = \pm 1$ are the Ising variables at sites $i \in \mathbb{Z}^d$, and $\beta = 1/T$ is the inverse temperature.

In the continuum limit, for $0 < s < 2$, the LRI model is described by a generalized free field (GFF) $\varphi$ of scaling dimension $\Delta_\varphi = (d - s)/2$, perturbed by quartic and quadratic self-interactions [11, 12]:

$$S_{\mathrm{LRI}}[\varphi] = S_{\mathrm{GFF}(\Delta_\varphi)}[\varphi] + \int \mathrm{d}^d x \left( \frac{\lambda_2}{2} \varphi(x)^2 + \frac{\lambda_4}{4} \varphi(x)^4 \right) . \tag{1.2}$$

This model is weakly coupled when $s$ approaches $d/2$, below which the critical behavior is governed by mean-field theory (MFT). For $s > d/2$, the critical behavior deviates from MFT, and becomes more and more strongly coupled with growing $s$.

The dual description proposed in [9, 10], and valid for $d \geq 2$, is in terms of the standard local (short-range) Ising CFT, denoted SRI. It features the Ising field $\sigma$, of scaling dimension $\Delta_\sigma$, linearly coupled to a GFF $\chi$ of scaling dimension $\Delta_\chi = (d + s)/2$:

$$\widetilde{S}_{\mathrm{LRI}}[\sigma, \chi] = S_{\mathrm{SRI}}[\sigma] + S_{\mathrm{GFF}(\Delta_\chi)}[\chi] + g \int \mathrm{d}^d x \, \sigma(x)\chi(x) \,. \tag{1.3}$$

This model is weakly coupled near $s^\star = d - 2\Delta_\sigma$, where (1.2) is strongly coupled. The value $s^\star$ had previously been conjectured [12, 13] to correspond to a crossover point for the LRI model, above which the LRI phase transition falls into the SRI universality class. One can use (1.3) to systematically compute the CFT data of the LRI near the crossover, see e.g. [9, 10, 14].

In this article, we consider the $d = 1$ case of the LRI–SRI crossover, where (1.3) does not apply because the local 1d Ising model does not give rise to a CFT. Let us summarize the state of the art about this model prior to our work. For $0 < s \leq 1$, the model exhibits a continuous phase transition at a finite value of $\mathcal{J}$ [15, 16, 17]. The critical exponents take mean-field values for $0 < s \leq 1/2$ [18]. For $s > 1$, by contrast, no finite-$\mathcal{J}$ transition occurs [19] — the model is disordered at all positive temperatures. The analogue of the crossover point is thus located at $s^\star = 1$, with divergent correlation length and discontinuous magnetization at the transition [20, 17].

For $s = 1/2 + \epsilon/2$, with $\epsilon \ll 1$, the 1d version of (1.2) provides a weakly coupled description of the near-critical lattice model (1.1). This description is strongly coupled near $s = 1$. At the same time, the dual description (1.3) does not have an immediate 1d analogue. Indeed, the one-dimensional SRI model exhibits a phase transition only at zero temperature, where it is not a CFT but a topological theory, with constant correlation functions. This theory is equivalent to that of a single qubit with two degenerate ground states $|\pm\rangle$. The spin field is a topological operator acting as $\sigma|\pm\rangle = \pm|\pm\rangle$.

Thus, while the generalized free field $\chi$, with $\Delta_\chi = (1 + s)/2$, coupled to the topological operator $\sigma$, with $\Delta_\sigma = 0$, via $\sigma\chi$ correctly identifies the crossover location at $s^\star = 1$, it fails to provide a correct description of the critical theory. Indeed, $\sigma$ never becomes a genuine local field, it does not acquire an anomalous dimension, and the relevant operator controlling the deviation from criticality is also absent. A different perspective is therefore needed in order to capture the full operator spectrum and scaling behavior near the crossover to short range in one dimension.

Such a perspective has been provided in our recent work [21], where we constructed a candidate dual description, weakly coupled near $s = 1$. Our proposal was inspired by the old works of Anderson and Yuval [22] and of Kosterlitz [23], which capture the physical essence of the problem. These references identified the weakly coupled degrees of freedom at $s \approx 1$ as the domain walls, namely the sites $i$ where the spins $\sigma_i$ flip from $-1$ to $+1$, or vice versa. For the 1d LRI with $s = 1$, Anderson and Yuval observed that the domain walls are dilute at low temperature and rewrote the model as a Coulomb gas of alternating kinks and antikinks, noticing also a connection with the Kondo model. Using their Coulomb gas model, and a primordial version of Wilsonian RG, they derived a system of beta functions and established a phase diagram at $s = 1$, which resembles that of the Berezinskii–Kosterlitz–Thouless (BKT) transition, except for a different physical dictionary. Later, Kosterlitz extended their analysis to small positive values of $1 - s$, constructing a dilute-gas description – hereafter referred to as the Anderson–Yuval–Kosterlitz (AYK) model – and identifying a weakly interacting fixed point. In [21], we have presented a field theory, taking the form of an impurity model that generalizes the bosonized Kondo model, whose perturbative expansion reproduces the AYK model, and that allows to perform systematic perturbative computations of the CFT data in the small parameter $1 - s$. Moreover, using the CFT data from the fixed point at $s = 1$ as a seed, we have developed a perturbative analytic conformal bootstrap that reproduces and extends the renormalization group (RG) results, thus providing an independent check of the field theoretic model and

of the fact that the fixed-point theory is a CFT.

In this paper, we expand on the construction of [21], presenting more details and results on the IR duality between the 1d LRI model and our generalized Kondo model. The main results of our work can be summarized as follows:

- At $s = 1$, we refine the known duality between LRI and Kondo, originally established by Anderson and Yuval. In particular, we show that the correct treatment requires restricting the Kondo model to its $U(1)$-singlet sector, which allows us to identify the full spectrum of the LRI CFT at $s = 1$.

- For all $s \leq 1$, we propose a field theory that is weakly coupled near the crossover, exactly solvable at $s = 1$, that reduces to the Kondo model at $s = 1$, and that upon perturbative expansion reproduces the AYK model. We derive several predictions for CFT data at next-to-leading or next-to-next-to-leading order in perturbation theory, thus showing that such model provides not only a conceptual framework to recast the AYK model as a field theory, but also a computational tool to treat the near-crossover regime with systematic perturbative methods.

- We recover the same results – along with further predictions – through analytic conformal bootstrap. This approach uses only the CFT data from $s = 1$, together with the existence of protected operators $\sigma$, $\chi$ for all $s \leq 1$, and the assumption that the CFT data admit an asymptotic expansion in nonnegative powers of $\sqrt{1 - s}$. Therefore, the bootstrap provides an independent validation of our proposed model at $s < 1$, and together they represent strong evidence for the conformal invariance of the IR fixed point.

## 1.1 Outline

The paper is organized as follows.

In Section 2, we review what is known (and what is not known) about the 1d LRI model, starting with the Ginzburg-Landau description. We further discuss its nonperturbative realization as a unitary 1d CFT and conclude with a lightning review of the AYK model, as well as the 1d SRI model.

In Section 3, we present our weakly coupled field theory for the 1d LRI–SRI crossover. We discuss its symmetries and operator content.

Section 4 features perturbative RG analysis of the model introduced in Section 3. We calculate the relevant beta functions, identify the weakly coupled fixed point, and compute CFT data. In particular, we extend AYK results for the critical exponents to higher orders in perturbation theory, and provide new predictions for OPE coefficients of light operators.

The conformal data can also be computed using the conformal bootstrap, with very few assumptions. In Section 5, we demonstrate how analytic bootstrap methods – particularly analytic functional techniques for 1d CFTs – independently reproduce and extend the RG results of Section 4 to higher orders.

We conclude in Section 6 with a summary of our results and a discussion of future directions. Supplementary material can be found in the appendices, including the definition of a 1d compact GFF, further details on the relation between the 1d LRI and the AYK model, the formulation of 1d LRI as a defect CFT, alternative versions of the proposed model, technical aspects of the perturbative computations, and logarithmic corrections to the scaling behavior of 1d LRI at the crossover.

## 2 Knowns and unknowns about the 1d long-range Ising model

In this section, we review the main properties of the $\varphi^4$ formulation of the 1d LRI model, what it teaches us about the IR CFT, and what was previously known about the physics near the crossover to the short-range universality class.

## 2.1 The $\varphi^4$-formulation of the long-range Ising model

The $\varphi^4$ formulation of the 1d LRI in the continuum is given in terms of a GFF $\varphi$ of scaling dimension

$$\Delta_\varphi = (1-s)/2\,, \tag{2.1}$$

perturbed by quartic and quadratic self-interactions. The latter preserve the $\mathbb{Z}_2$ and parity symmetries of the GFF.[1] The action is [11]

$$S_{\text{LRI}}[\varphi] = \frac{c_s}{4} \int_{-\infty}^{+\infty} dx_1 dx_2 \frac{(\varphi(x_1) - \varphi(x_2))^2}{|x_1 - x_2|^{1+s}} + \int_{-\infty}^{+\infty} dx \left( \frac{\lambda_2}{2} \varphi(x)^2 + \frac{\lambda_4}{4} \varphi(x)^4 \right)\,. \tag{2.2}$$

We can think of the continuous field $\varphi$ as the order parameter (the spontaneous magnetization) in a Ginzburg-Landau description of the LRI. As usual, the coupling $\lambda_2$ is associated to the deviation from the critical temperature of the statistical model.

The canonical dimension of the field is not renormalized by the presence of local interactions [11, 24], hence it sticks to its GFF value of eq. (2.1). For $0 < s \le 1/2$, the quartic interaction is irrelevant and the IR theory is GFF, which explains why the critical exponents of 1d LRI are controlled by MFT in this region [18]. For $s = (1 + \epsilon)/2$ with $0 < \epsilon \ll 1$, the quartic interaction is weakly relevant. Setting $\lambda_2 = 0$ and using analytic $\epsilon$-regularization for the UV divergences, the one-loop beta function for the renormalized coupling is found to be [11] (see [25, 26] for the three-loop result)

$$\beta_{\lambda_4} = -\epsilon \lambda_4 + \frac{3}{2\pi} \lambda_4{}^2 + O(\lambda_4{}^3)\,. \tag{2.3}$$

The critical IR theory at $\lambda_4 = 2\pi\epsilon/3 + O(\epsilon^2)$ is weakly coupled for $\epsilon \ll 1$. In the opposite regime, when $\epsilon \sim O(1)$, perturbation theory is unreliable.[2] What can we say about the critical IR theory near the crossover transition to short-range, which in 1d happens at $s = \epsilon = 1$?

The expected scenario, dating back to Sak's work [12] (see also [28, 9, 10]), is that the short-range kinetic operator $\varphi \partial^2 \varphi$, becomes a *dangerously irrelevant* operator at the crossover, i.e. it is irrelevant in the UV theory, but becomes relevant in the IR, thus destabilizing the IR fixed point.[3] The scenario by Sak leaves open a number of puzzles, that have been resolved in higher dimensions by the crossover picture proposed in [9, 10], and briefly sketched in the introduction. In particular, in Sak's scenario there is a problem of missing operators after crossover to the short-range universality (e.g. the $\varphi^3$ operator, discussed below), and this has been solved in [9, 10] with the realization that beyond the crossover the theory is equivalent to SRI plus a decoupled GFF. Before presenting our proposal for a weakly coupled description of the crossover in 1d, we will review a few additional pieces of background material.

## 2.2 The 1d long-range Ising CFT

The critical 1d long-range Ising model is expected to develop symmetry under Möbius transformations of the line, and is thus described by a 1d CFT [33]. Informally, a 1d CFT is a theory living on a line, whose correlators transform covariantly under the group $\text{PSL}_2(\mathbb{R})$ of real fractional linear transformations $x \mapsto \frac{ax+b}{cx+d}$ with $ad - bc = 1$. 1d CFTs are inherently non-local. Indeed, a local 1d theory with $\text{PSL}_2(\mathbb{R})$ symmetry and invariant vacuum would necessarily be topological as the stress tensor vanishes. Examples

---

[1]The reader can consult appendix A regarding the definition of GFF on the line.

[2]See refs. [25, 26, 14, 27], for various resummation-based estimates of critical exponents when $\epsilon \sim O(1)$.

[3]We note that in such a scenario, when following the IR theory from $s = 1/2$ to $s = 1$, we should observe level crossing of the scaling dimensions of $\varphi^4$ and $\varphi \partial^2 \varphi$. Indeed they start as marginal and irrelevant, respectively, at $s = 1/2$, and should end up as irrelevant and marginal at $s = 1$. Such level crossing is observed in higher dimensions by means of perturbative series resummations [27], but it is expected that the true behavior of the operators would result in a level repulsion due to nontrivial mixing near the would-be level-crossing point, see e.g. [29, 30, 31, 32] for recent investigations on this phenomenon.

of 1d CFTs include conformal boundary conditions of 2d CFTs, conformal line defects in general-d CFTs, as well as the boundary duals of QFT in $AdS_2$, and fixed points of 1d long-range models. We refer the reader to [35, 36, 37, 38] for some previous literature on general aspects of 1d CFTs.

Besides the $PSL_2(\mathbb{R})$ and global $\mathbb{Z}_2$ symmetry, the 1d LRI also features parity symmetry, acting on the line as $x \mapsto -x$, which combines with $PSL_2(\mathbb{R})$ to form $PGL_2(\mathbb{R})$. Note that in the context of line defects, parity is usually called S-parity [39, 40]. Furthermore, 1d LRI is a reflection-positive theory, and therefore its analytic continuation to Lorentzian signature is unitary. We will now describe the bootstrap definition of a unitary 1d CFT with parity symmetry, and then discuss several nonperturbative features of the 1d LRI CFT, namely its protected operators and OPE ratios.

### 2.2.1 Unitary 1d CFTs with parity symmetry: the bootstrap definition

In a unitary (=reflection-positive) CFT, the space of local operators carries a positive norm, which makes it into a Hilbert space $V$.[4] Due to Möbius symmetry, $V$ is a unitary representation of the group $\widetilde{\mathrm{SL}}_2(\mathbb{R}) \rtimes \mathbb{Z}_2$, where the second factor represents parity. This group is the universal cover of $\mathrm{SL}_2(\mathbb{R}) \rtimes \mathbb{Z}_2$, itself arising as the real section of the complexification of $PGL_2(\mathbb{R})$ which preserves the inner product of radial quantization. $V$ decomposes as a discrete direct sum of unitary irreducible lowest-weight representations of $\widetilde{\mathrm{SL}}_2(\mathbb{R}) \rtimes \mathbb{Z}_2$

$$V = \bigoplus_{i=0}^{\infty} D_{\Delta_i, J_i} \,. \tag{2.4}$$

The lowest-weight state in $D_{\Delta_i, J_i}$ corresponds to a local primary operator $\phi_i$ at $x = 0$. $\Delta_i$ is the scaling dimension of $\phi_i$, and $J_i \in \{0, 1\}$ its parity, so that $x \mapsto -x$ acts as $\phi_i(0) \mapsto (-1)^{J_i}\phi_i(0)$. The sequence of ordered pairs $((\Delta_i, J_i))_{i=0}^{\infty}$ is called the *spectrum* of the theory. Reflection positivity implies $\Delta_i \geq 0$. We have $\Delta_0 = 0$, $J_0 = 0$, corresponding to the vacuum (in other words, $\phi_0$ is identity operator $\mathbb{1}$). Correlation functions of $\phi_i$ are invariant under $PGL_2(\mathbb{R})$, which maps point $x$ to $\frac{ax+b}{cx+d}$ and the field $\phi$ to $\phi'$, given by[5]

$$\phi'(x) = \mathrm{sign}(ad - bc)^{J_i}|-cx + a|^{-2\Delta_i}\phi_i\left(\frac{dx-b}{-cx+a}\right) \,. \tag{2.5}$$

Local operators satisfy the operator product expansion (OPE). The $PGL_2(\mathbb{R})$ symmetry constrains it to take the form

$$\phi_i(x_1)\phi_j(x_2) = \sum_{k=0}^{\infty} c_{ijk}(-1)^{J_k}|x_{12}|^{-\Delta_i - \Delta_j + \Delta_k} \sum_{n=0}^{\infty} \frac{(\Delta_k + \Delta_i - \Delta_j)_n}{n!(2\Delta_k)_n} x_{12}^n \partial_2^n \phi_k(x_2) \,, \tag{2.6}$$

where here and in the following we assume $x_{ij} \equiv x_i - x_j > 0$. $c_{ijk}$ are the OPE coefficients, also known as structure constants. Reflection positivity implies that we can take $c_{ijk} \in \mathbb{R}$, which we will assume from now on. The spectrum and the structure constants are together referred to as the *CFT data*.

The CFT data contains all the information needed for calculating any correlation functions of local operators. In particular, up to $n = 4$, the $n$-point functions of primary operators take the form

$$\langle \phi_i(x_1) \rangle = \delta_{i0} \,,$$
$$\langle \phi_i(x_1)\phi_j(x_2) \rangle = \frac{(-1)^{J_i}\delta_{ij}}{(x_{12})^{2\Delta_i}} \,,$$
$$\langle \phi_i(x_1)\phi_j(x_2)\phi_k(x_3) \rangle = \frac{c_{ijk}}{(x_{12})^{\Delta_{ijk}}(x_{13})^{\Delta_{ikj}}(x_{23})^{\Delta_{jki}}} \,, \tag{2.7}$$
$$\langle \phi_i(x_1)\phi_j(x_2)\phi_k(x_3)\phi_l(x_4) \rangle = \left(\frac{x_{14}}{x_{24}}\right)^{\Delta_{ji}} \left(\frac{x_{14}}{x_{13}}\right)^{\Delta_{kl}} \frac{\mathcal{G}_{ijkl}(z)}{(x_{12})^{\Delta_i + \Delta_j}(x_{34})^{\Delta_k + \Delta_l}} \,,$$

---

[4]The reflection operation is defined as the composition of parity with complex conjugation.

[5]Here $|-cx + a|$ is defined using a representative $g \in \mathrm{GL}_2(\mathbb{R})$ of $[g] \in PGL_2(\mathbb{R})$ with $|\det(g)| = 1$.

where $\Delta_{ij} \equiv \Delta_i - \Delta_j$, $\Delta_{ijk} \equiv \Delta_i + \Delta_j - \Delta_k$, and

$$z \equiv \frac{x_{12}x_{34}}{x_{13}x_{24}} \tag{2.8}$$

is the cross-ratio, satisfying $z \in (0,1)$ for $x_1 > x_2 > x_3 > x_4$. Since the three-point functions are proportional to the structure constants, the latter satisfy (anti-)symmetry under permutations of labels

$$c_{ijk} = c_{jki} = c_{kij} = (-1)^{J_i+J_j+J_k}c_{jik} = (-1)^{J_i+J_j+J_k}c_{ikj} = (-1)^{J_i+J_j+J_k}c_{kji}, \quad c_{0ij} = \delta_{ij}. \tag{2.9}$$

The CFT data must be compatible with associativity of the OPE, most tangibly expressed as crossing symmetry of all four-point functions. In practice, we use either the $\phi_i\phi_j$, or $\phi_j\phi_k$ OPE, in the form (2.6), inside the four-point function $\langle\phi_i\phi_j\phi_k\phi_\ell\rangle$. This leads to the crossing equations

$$\mathcal{G}_{ijk\ell}(z) = \sum_{m=0}^{\infty} c_{ijm}c_{k\ell m}(-1)^{J_m}G_{\Delta_m}^{\Delta_i,\Delta_j,\Delta_k,\Delta_\ell}(z) = \sum_{m=0}^{\infty} c_{jkm}c_{\ell im}(-1)^{J_m}G_{\Delta_m}^{\Delta_i,\Delta_\ell,\Delta_k,\Delta_j}(1-z), \tag{2.10}$$

holding for all $i,j,k,\ell \in \mathbb{Z}_{>0}$ and all $0 < z < 1$. Here $G_{\Delta_m}^{\Delta_i,\Delta_j,\Delta_k,\Delta_\ell}(z)$ are the *1d conformal blocks*, first computed in [41], taking the form

$$G_{\Delta_m}^{\Delta_i,\Delta_j,\Delta_k,\Delta_\ell}(z) = z^{\Delta_m-\Delta_k-\Delta_\ell}\,_2F_1(\Delta_m - \Delta_i + \Delta_j, \Delta_m + \Delta_k - \Delta_\ell; 2\Delta_m; z). \tag{2.11}$$

In summary, a unitary 1d CFT with parity symmetry can be defined as the collection of data $((\Delta_i, J_i))_{i=0}^{\infty}$, $c_{ijk}$, with $\Delta_i \geq 0$, $c_{ijk} \in \mathbb{R}$ and subject to (2.9) and (2.10).

### 2.2.2 The range $0 < s \leq 1/2$

In this region, the IR fixed point of (2.2) is at $\lambda_2 = \lambda_4 = 0$. The critical behavior is described by GFF of dimension $\Delta_\varphi = (1-s)/2$, which is a simple example of a 1d CFT, as defined above. The GFF is indeed invariant under the Möbius group including parity, as well $\mathbb{Z}_2$ global symmetry. $\varphi$ is even under parity and odd under the global $\mathbb{Z}_2$.

A basis of local operators of the GFF is obtained by forming normal-ordered words using letters $\partial^n\varphi$ with $n \in \mathbb{Z}_{\geq 0}$. The normal order eliminates any Wick contractions of $\varphi$ fields at the same point, hence the conformal covariance of the correlators of such operators is a trivial consequence of the conformal transformations of the covariance $C(x)$.

The spatial derivative $\partial$ is even under the global $\mathbb{Z}_2$ and odd under parity. Therefore, the states are counted by the generating function

$$Z_{\text{GFF}}(x,y,q) = \text{tr}_V\left(x^\alpha y^J q^D\right) = \frac{1}{\prod_{n=0}^{\infty}(1 - x\,y^n q^{n+\Delta_\varphi})}. \tag{2.12}$$

Here $D$ is the dilatation operator, $(-1)^\alpha$ is the generator of the global $\mathbb{Z}_2$, while $(-1)^J$ is the generator of parity. Note that we should impose $x^2 = y^2 = 1$ on the RHS. The partition function counting primaries only is

$$Z^*_{\text{GFF}}(x,y,q) = 1 + (1 - yq)[Z_{\text{GFF}}(x,y,q) - 1]. \tag{2.13}$$

Note that the lightest $\mathbb{Z}_2$-even and parity-odd primary, schematically $\varphi^3\partial^3\varphi$, has dimension $3 + 4\Delta_\varphi$.

### 2.2.3 The range $1/2 < s \leq 1$

In this region, the fixed point of (2.3) defines a one-parameter family of 1d CFTs that reduce to the GFF at $\epsilon = 0$.[6] At $\epsilon > 0$, the perturbative fixed point will in general give reliable results only at $\epsilon \ll 1$. Nevertheless, there are some facts about this family of 1d CFTs that remain valid also nonperturbatively at finite $\epsilon$.

Firstly, the interacting fixed point still possesses a $\mathbb{Z}_2$ global and parity symmetry. Secondly, the long-range nature of the fixed point implies that certain scaling dimensions are protected. Only the Gaussian part of the model is long-range, while the $\varphi^4$ interaction is fully local. It follows that (2.1) is valid also in the IR for all $0 < s \leq 1$.

Furthermore, as a consequence of the Schwinger-Dyson equations, we have the following identification (up to contact terms):

$$\left\langle \int dy\, C^{-1}(x-y)\varphi(y) \ldots \right\rangle = \left\langle \lambda_4 \varphi(x)^3 \ldots \right\rangle . \tag{2.14}$$

Since the inverse covariance is not a differential operator for $s < 2$, and its action coincides with a shadow transform [42], $\varphi$ and $\varphi^3$ are two distinct primaries that form a shadow pair [33]. It follows that the dimension of $\varphi^3$ is also protected

$$\Delta_{\varphi^3} = 1 - \Delta_\varphi = (1+s)/2 .$$

We will refer to these protected operators more abstractly as $\sigma$ and $\chi$, with the understanding that in the $\varphi^4$ description we have the identifications $\sigma \sim \varphi$ and $\chi \sim \varphi^3$. In particular, we have $\Delta_\sigma = (1-s)/2$ and $\Delta_\chi = (1+s)/2$ for all $s \in [1/2, 1]$.

Finally, since the Schwinger-Dyson equation effectively allows us to build $\chi$ as the shadow transform of $\sigma$, it implies nonperturbative relations between various OPE coefficients [33, 43, 14]. Specifically, for any four primaries $\phi_i$, $\phi_j$, $\phi_k$, $\phi_\ell$, we have

$$\frac{c_{\sigma ij}c_{\chi k\ell}}{c_{\chi ij}c_{\sigma k\ell}} = \frac{\Gamma\left(\frac{\Delta_\sigma + \Delta_i - \Delta_j + a_{ij}}{2}\right)\Gamma\left(\frac{\Delta_\sigma - \Delta_i + \Delta_j + a_{ij}}{2}\right)\Gamma\left(\frac{1-\Delta_\sigma + \Delta_k - \Delta_\ell + a_{k\ell}}{2}\right)\Gamma\left(\frac{1-\Delta_\sigma - \Delta_k + \Delta_\ell + a_{k\ell}}{2}\right)}{\Gamma\left(\frac{1-\Delta_\sigma + \Delta_i - \Delta_j + a_{ij}}{2}\right)\Gamma\left(\frac{1-\Delta_\sigma - \Delta_i + \Delta_j + a_{ij}}{2}\right)\Gamma\left(\frac{\Delta_\sigma + \Delta_k - \Delta_\ell + a_{k\ell}}{2}\right)\Gamma\left(\frac{\Delta_\sigma - \Delta_k + \Delta_\ell + a_{k\ell}}{2}\right)} , \tag{2.15}$$

where $a_{ij} = [1 - (-1)^{J_i + J_j}]/2 = J_i + J_j \mod 2$. A derivation appears in [43] and in our Appendix G.

In the particular case $\phi_i = \chi$, $\phi_k = \sigma$, $\phi_j = \phi_\ell = \mathcal{O}$, for a general primary $\mathcal{O}$, this relation becomes

$$(c_{\sigma \chi \mathcal{O}})^2 = \frac{\Gamma\left(\frac{1+J_\mathcal{O} - \Delta_\mathcal{O}}{2}\right)^2 \Gamma\left(\frac{1+\Delta_\mathcal{O} + J_\mathcal{O} - 2\Delta_\sigma}{2}\right)\Gamma\left(\frac{2\Delta_\sigma + \Delta_\mathcal{O} + J_\mathcal{O} - 1}{2}\right)}{\Gamma\left(\frac{\Delta_\mathcal{O} + J_\mathcal{O}}{2}\right)^2 \Gamma\left(\frac{2\Delta_\sigma - \Delta_\mathcal{O} + J_\mathcal{O}}{2}\right)\Gamma\left(\frac{2-2\Delta_\sigma - \Delta_\mathcal{O} + J_\mathcal{O}}{2}\right)}c_{\sigma \sigma \mathcal{O}}c_{\chi \chi \mathcal{O}} . \tag{2.16}$$

When $\mathcal{O}$ is odd under parity, Bose symmetry (2.9) implies $c_{\sigma \sigma \mathcal{O}} = c_{\chi \chi \mathcal{O}} = 0$. Equation (2.16) then implies that either $c_{\sigma \chi \mathcal{O}} = 0$ or $\Delta_\mathcal{O}$ is a positive even integer, so that a pole of the first gamma function in the numerator has a chance to cancel the vanishing factor $c_{\sigma \sigma \mathcal{O}}c_{\chi \chi \mathcal{O}}$. These protected operators can be thought of as double traces built out of $\sigma$ and $\chi$, together with an odd number of derivatives. As the mean field description near $s = 1/2$ shows, the case with one derivative is a descendant of $\varphi^4$, and the protected operators thus have dimensions $4, 6, 8, \ldots$. See also [44] for a recent discussion of these operators from a different point of view.

Since there are no local conserved currents in the long-range model, we expect all other operators to get nontrivial anomalous dimensions, and thus lift any degeneracies present in the GFF. We refer the reader to [43, 25, 26, 14, 27], for detailed perturbative results in the $\varphi^4$ formulation and up to three loops in $\epsilon$-expansion.

---

[6]For the LRI model in general dimension $d$, in [33] it has been proved to all orders of the $\epsilon$ expansion that the scale invariance of the fixed-point theory is enhanced to full conformal invariance.

## 2.3 Anderson-Yuval-Kosterlitz model

In this section, we review the relation between the partition function of the 1d LRI model for $s \lesssim 1$,

$$Z_{\text{LRI}} = \sum_{\{\sigma_i\}} e^{-\beta \mathcal{H}_{\text{LRI}}} \,, \tag{2.17}$$

and a Coulomb gas of kinks with alternating charges. The main idea, which goes back to the work of Anderson and Yuval in [22], is to view the locations of the domain walls, i.e. the sites $i$ where the spins $\sigma_i$ flip sign (see fig. 2) as the weakly coupled degrees of freedom for 1d LRI. We will refer to a domain wall configuration from 1 to $-1$ ($-1$ to 1) as a kink (anti-kink). For $s \lesssim 1$, such domain walls are indeed diluted at low temperature, where the model is in the ordered phase.

For $s = 1$, upon rewriting LRI in terms of kinks and anti-kinks, $Z_{\text{LRI}}$ is seen to be equivalent to the following Coulomb gas partition function [22] (see also the related previous work [45, 46, 47] and the review in [48]):[7]

$$Z_{\text{Coulomb}} = \sum_{n=0}^{+\infty} g^{2n} \int_{\text{I}_{2n}(a)} \Big( \prod_{i=1}^{2n} \frac{\mathrm{d}x_i}{a} \Big) e^{2\mathcal{J} \sum_{i<j} (-1)^{i-j} \log(|x_i - x_j|/a)} \,, \tag{2.18}$$

where positive and negative charges are associated to kinks and anti-kinks, respectively, and in accordance with the domain wall interpretation they need to alternate along the line. The domain of integration is:

$$\text{I}_m(a) = \{L/2 \geq x_1 \geq x_2 \geq \ldots \geq x_m \geq -L/2 \mid x_i - x_{i+1} \geq a\} \,. \tag{2.19}$$

The fugacity of the gas of kinks is $g = \exp(-\mathcal{K} - c\,\mathcal{J})$, for some constant $c$ and with $\mathcal{K}$ being the short-range coupling that is added to the LRI in order to have two independently-tunable couplings. The expansion in powers of the fugacity should be valid at low temperatures (i.e. large $\mathcal{J}$ and $\mathcal{K}$, or small $g$), where domains are large and the domain walls are thus very diluted.

In writing (2.18), a continuum limit has been implemented, while keeping a UV cutoff $a$ (inherited from the lattice cutoff) in the form of a hard core repulsion.[8] The IR regularization instead has been implemented by choosing a finite $L$. Physically, this can be seen as defining the LRI model on the infinite line, but with a strong magnetic field in $\mathbb{R} \backslash [-L/2, L/2]$, forcing all the spins to point in the same direction in the complement of the interval $[-L/2, L/2]$, and thus excluding kinks in that region.[9]

For $s \lesssim 1$, as found by Kosterlitz in [23], rewriting LRI in terms of kinks and anti-kinks leads to a modification of the Coulomb gas, where the logarithms of $Z_{\text{Coulomb}}$ are replaced by a power-law potential:

$$Z_{\text{AYK}} = \sum_{n=0}^{+\infty} g^{2n} \int_{\text{I}_{2n}(a)} \Big( \prod_{i=1}^{2n} \frac{\mathrm{d}x_i}{a} \Big) e^{2\mathcal{J} \sum_{i<j} (-1)^{i-j} (1-s)^{-1} ((|x_i - x_j|/a)^{1-s} - 1)} \,. \tag{2.20}$$

This modified Coulomb gas model, which reduces $Z_{\text{Coulomb}}$ for $s \to 1^-$, is what we call the AYK model.

From the AYK model, by means of a primordial version of the Wilsonian RG, AYK obtained the RG flow equations for the renormalized parameters $\mathcal{J}(a), g(a)$ with respect to the cutoff length $a$:

$$\begin{aligned} \frac{dg}{d\log(a^{-1})} &= (\mathcal{J} - 1)g + \mathcal{O}(g^3) \,, \\ \frac{d\mathcal{J}}{d\log(a^{-1})} &= \mathcal{J}(4g^2 + s - 1) + \mathcal{O}(g^4) \,. \end{aligned} \tag{2.21}$$

---

[7]In appendix B, we provide a simplified derivation of the this model from the $\varphi^4$ formulation of LRI.

[8]The UV cutoff can actually be removed for $\mathcal{J} < 1/2$, as the singularity at coinciding points is then integrable.

[9]Alternatively, one can define the model with periodic boundary conditions; however, one should keep in mind that in this case we should also change the argument of the logarithm as in $\log(\frac{\sin(\pi |r_i - r_j|/L)}{\pi a/L})$.

The flow diagram associated to such beta functions is displayed in fig. 1. At $s = 1$, the associated flow has a similar structure to the BKT transition [48]: a line of fixed points at $g = 0$, parametrized by $\mathcal{J}$, which are attractive or repulsive in the IR, depending on the sign of $\mathcal{J} - 1$; the line ending at $g = 0$ and $\mathcal{J} = 1$ in the IR corresponds to the phase transition between order ($g = 0$, i.e. no kinks, in the IR) and disorder ($g \sim O(1)$, i.e. proliferation of kinks, in the IR). At $s < 1$, the line of fixed points disappears, leaving only an isolated fixed point, which is the continuation of the $g = 0$ and $\mathcal{J} = 1$ fixed point of $s = 1$. One of our purposes will be to reproduce these RG equations from a modern perspective, in the language of CFT, and at the same time provide a framework allowing to systematically improve them and produce other results.

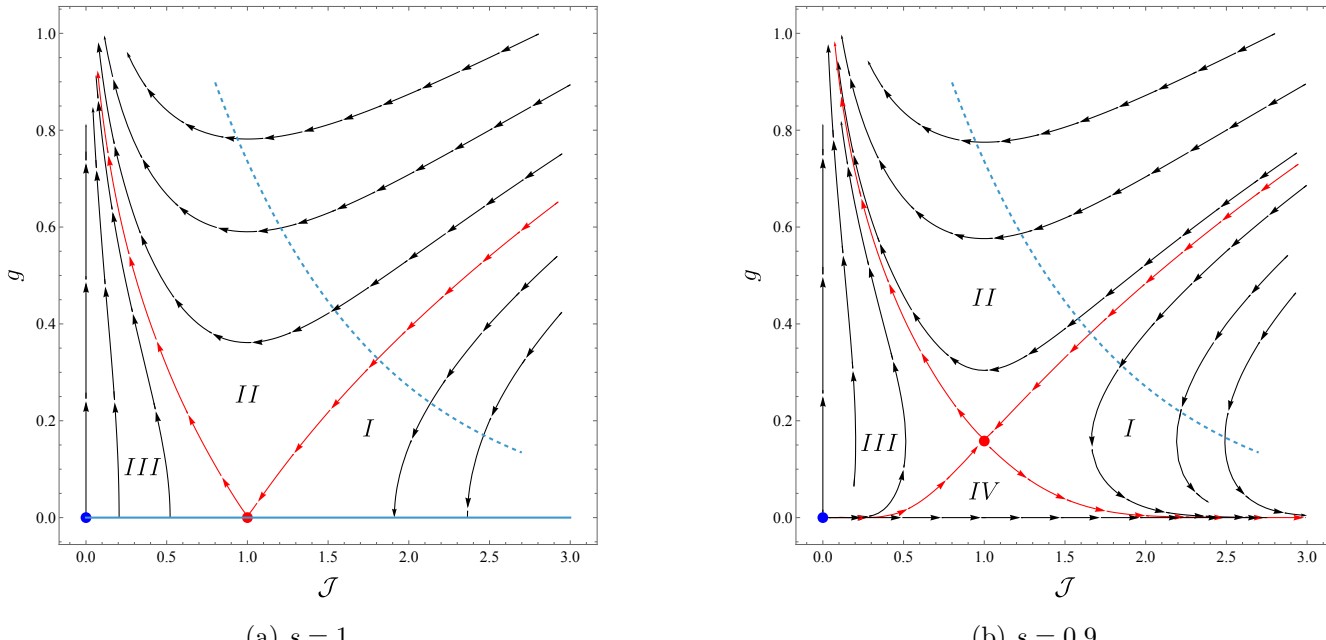

|  |  |
|:--:|:--:|
| (a) $s = 1$ | (b) $s = 0.9$ |

Figure 1: The RG flow for the beta functions in (2.21), at $s = 1$ and $s = 0.9$. Arrows point towards the IR. At $s = 0$, the $g = 0$ axis is a line of fixes points, but we have emphasized in blue and red the special points $\mathcal{J} = 0$ and $\mathcal{J} = 1$, respectively. The red lines separate the diagram into three or four regions: region $I$ is the low-temperature broken phase ($g = 0$ in the IR); region $II$ is the high-temperature symmetric phase ($g \sim O(1)$ in the IR); regions $III$ and $IV$ are inaccessible to the LRI. The latter indeed corresponds to a one-dimensional subspace such as the dashed blue line. The intersection of such line with the red line defines the critical coupling $\mathcal{J}_c$ (i.e. critical temperature at fixed bond strength $J$, if $\mathcal{J} = J/T$).

## 2.4 Insights from the 1d short-range Ising model

Another important insight for developing a theory of the LRI model near $s = 1$ comes from scrutinizing the possible physical content of the theory at $s > 1$. As reviewed above, for $d > 1$ and $s \geq s^*$, the LRI crosses to the SRI universality plus a decoupled GFF sector [10, 9]. What changes when $d = 1$? As we argue below, the physics is bound to be not so different from that of the classical 1d SRI model, indeed:

1. For 1d LRI, there is no phase transition at finite temperature for $s > 1$, just as for the 1d SRI. This was established rigorously in [19];

2. At $s = 1$, from eq. (2.1), the LRI field has a vanishing scaling dimension, thus matching that of the 1d SRI at zero temperature, e.g. [49]. We return to this point below.

At the same time, the physics cannot be exactly that of the classical 1d SRI model. The classical 1d SRI model in a magnetic field $h$ is given by the Hamiltonian

$$\beta \, \mathcal{H}_{\text{SRI}} = \frac{\mathcal{K}}{2} \sum_i (\sigma_{i+1} - \sigma_i)^2 - h \sum_i \sigma_i \,, \quad \mathcal{K} > 0 \,, \tag{2.22}$$

and it is easily solved, in particular by transfer matrix method. The latter underlines the standard classical-to-quantum mapping [50, 51, 52], and in the continuum limit we can think of the classical 1d SRI model as a quantum SRI in transverse field in zero dimensions, i.e. the quantum mechanics of a single Ising spin, with Hamiltonian

$$\hat{H}_Q = -\gamma \, \hat{\sigma}_1 - h \, \hat{\sigma}_3 \,, \tag{2.23}$$

where $\hat{\sigma}_a$ are the standard Pauli matrices and

$$2\gamma = \log \coth(\mathcal{K}) = \xi^{-1} \,, \tag{2.24}$$

with $\xi$ the correlation length of the classical Ising chain at $h = 0$. We see that the correlation length diverges only for $\gamma \to 0$, i.e. $\mathcal{K} \to +\infty$, corresponding to the zero temperature limit of the classical model. The critical SRI model is thus located at $\gamma = h = 0$, that is, it is described by a trivial quantum Hamiltonian.

Correlation functions of the classical Ising chain with periodic boundary conditions are mapped to thermal expectation values of time-ordered operators in the quantum theory, where time-dependent operators are defined by the imaginary-time evolution under $\hat{H}_Q$. The imaginary time and the inverse temperature of the quantum model are identified with the position $x$ and the extension $L$ of the classical model, respectively. At the critical point, the two-point function of the classical Ising field then simply becomes

$$\langle \sigma(x)\sigma(0) \rangle = \frac{1}{Z_Q} \, \text{tr} \left[ e^{-L\hat{H}_Q} \hat{\sigma}_3(x)\hat{\sigma}_3(0) \right] \xrightarrow{\gamma,h\to 0} \frac{1}{2} \, \text{tr} \left[ \hat{\sigma}_3^2 \right] = 1 \,. \tag{2.25}$$

Therefore, the Ising field becomes a field with vanishing scaling dimension, and operator representation given by the Pauli matrix $\hat{\sigma}_3$. Notice that when approaching the critical theory along the $h$-axis at $\gamma = 0$, we select (for $L \to \infty$) as ground state of $\hat{H}_Q$ the eigenvector of $\hat{\sigma}_3$ with eigenvalue $\pm 1$, depending on the sign of $h$, hence the spontaneous magnetization $\langle \sigma \rangle$ is discontinuous at zero temperature.

Given the facts we just recalled about the SRI, we would expect that at $s \geq 1$ the critical LRI, with classical Hamiltonian (1.1), would be described by the degrees of freedom of a $\mathbb{C}^2$ Hilbert space, with trivial correlators. However, as in $d > 1$, this cannot be the end of the story, because it would imply that at the crossover some operators would disappear from the spectrum. For example, by continuity of spectrum, at $s = 1$ we should find two $\mathbb{Z}_2$-odd operators with opposite parity and of dimension exactly equal to one, corresponding to $\varphi^3$ and $\partial\varphi$ in the $\varphi^4$ description (see Section 2.2), but clearly the SRI alone cannot provide such operators.[10]

The solution, to be presented in the following section, will come from matching the theory at $s = 1$ to the AYK model.

# 3 The weakly coupled model near crossover

As reviewed above, the dynamics of 1d LRI model at $s \lesssim 1$ is described by a gas of kinks-antikinks with alternating charges. For $s = 1$, the same Coulomb gas can also be obtained as a perturbative expansion

---

[10]Following [9, 10], we could try to account for the missing operators by adding to the 1d SRI a GFF $\chi$ of scaling dimension $\Delta_\chi = (1 + s)/2$, that could be identified with $\varphi^3$, and that should be decoupled from the spin degree of freedom at $s \geq 1$. However, we would still be missing the parity-odd operator corresponding to $\partial\varphi$, as it would be unclear how to introduce derivatives of $\hat{\sigma}_3$. Moreover, it would be unclear how to couple the two sectors at $s < 1$ in order to flow to the LRI CFT: if for example, combining insights from the above reminder of 1d SRI and from [9, 10], we introduce the defect operator $\text{tr}[\exp(h \int \text{d}x \, \hat{\sigma}_3 \, \chi(x))]$ in correlators over the $\mathbb{C}^2$+GFF theory, we obtain a trivial result, as the trace vanishes.

of the Kondo model [22]. In its bosonized version, the latter is very similar to the boundary sine-Gordon model, except for the presence of Pauli matrices in its action, enforcing the alternating order of positive and negative charges in (2.19).[11]

In this section, we introduce a model that, generalizing the bosonized Kondo model, provides a field theory whose perturbative expansion reproduces the AYK model also at $s < 1$, and that has the correct operator content to match the $\varphi^4$ description. We elucidate various aspects of such model, arguing that even at $s = 1$ the observation by Anderson and Yuval needs a refinement, as the Kondo model leads to a larger spectrum than the LRI.

## 3.1 Compact generalized free fields with negative scaling dimension

As a preliminary step towards introducing the full dual model for 1d LRI near $s = 1$, we first introduce a dual GFF, that will provide a basis around which to construct the perturbative treatment. From here on we set $s = 1 - \delta$, with $0 \leq \delta \ll 1$.

Consider the 1d GFF $\phi$ of negative dimension $\Delta_\phi = -\delta/2$, whose target space is a circle of circumference $2\pi/b_0$:

$$\phi \sim \phi + 2\pi n/b_0 \quad n \in \mathbb{Z}. \tag{3.1}$$

Note that the inverse radius has mass dimension $[b_0] = \delta/2$, and we thus write $b_0 = \kappa^{\delta/2}$, for some mass scale $\kappa$, such that $b_0 \to 1$ for $\delta \to 0$.

As we review in Appendix A, for $-1 < \Delta_\phi \leq 0$, the noncompact 1d GFF can be rigorously defined [57] as a probability measure on the space of distributions $\mathbb{R} \to \mathbb{R}$ defined modulo an additive constant. In fact, the resulting random distribution is almost surely a continuous function (modulo an additive constant). In other words, such GFF preserves shift symmetry. Hence, we can compactify the target space and define a compact version of such GFF on the space of continuous functions $\mathbb{R} \to S^1_{1/b_0}$.

We denote the expectation value in this compact GFF theory by $\langle \cdot \rangle_0$. The covariance kernel $C(x) = \langle \phi(x)\phi(0) \rangle_0$ is defined only up to an additive constant, as discussed in Appendix A.2. This simply means that $\phi$ is not a well-defined random variable. However, for practical calculations we still deal with its covariance in intermediate steps, hence it will be convenient to fix its form. We will choose the normalization and the additive constant so that the covariance has a well-defined limit for $\delta \to 0$:[12]

$$C(x) = -\frac{2}{\delta} \left( |x|^\delta - \kappa^{-\delta} \right) = -2 \log(\kappa|x|) + O(\delta) . \tag{3.2}$$

Due to the identification in field space, all well-defined observables must be built out of $\partial_x^n \phi(x)$ with $n \geq 1$, and vertex operators $e^{inb_0\phi(x)}$, with $n \in \mathbb{Z}$. The correlators of $\partial_x^n \phi(x)$ are obtained straightforwardly as derivatives of (products of) $C(x)$, and so they are clearly independent of the additive constant; those of the vertex operators will be discussed below. All such operators have positive dimension, and the resulting theory is unitary, as proven rigorously in [58].[13]

We point out that while this GFF might seem a bit exotic, its status is not much different from the standard massless scalar field in $d = 1$. The latter corresponds to taking $\delta = 1$, and thus has negative scaling dimension. Therefore, it also needs to be defined by modding out its zero mode, and thus also in this case the well-defined random variables are those that are independent of the zero mode. In particular, the variable $\phi(x) - \phi(0)$ corresponds to the standard Wiener process (Brownian motion), and the growth of the covariance reproduces the well-known linear growth of the Brownian motion: $\langle (\phi(x) - \phi(0))^2 \rangle_0 \sim |x|$. The generalization to $\delta \neq 1$ is known as *fractional Brownian motion*, and it was introduced long ago in [59]. Lastly, the compactification simply corresponds to the restriction of the random process to a circle.

---

[11]The relation between Kondo and boundary sine-Gordon models has been elucidated in [53]. We refer also to [54, 55, 56] for reviews of the CFT approach to Kondo and boundary sine-Gordon models.

[12]Notice that continuing to $\delta < 0$ this normalization would differ from that of the GFF in Section 2.1.

[13]More precisely, in [58] the authors proved reflection positivity of the fractional Brownian motion, that is obtained from the GFF $\phi$ by fixing its value to be zero at the origin.

**Correlation function of vertex operators.** At $\delta = 0$, the vertex operators $V_n(x) \equiv e^{inb_0\phi(x)}$ are defined with normal ordering[14] and have the following correlators:

$$\langle V_{n_1}(x_1)\cdots V_{n_m}(x_m)\rangle_0 = \delta_{0,\sum_i n_i} \prod_{i<j} |x_i - x_j|^{2b_0^2 n_i n_j} \,, \tag{3.3}$$

where we temporarily keep $b_0$ although it equals one at $\delta = 0$. This is derived precisely as in 2d [60], because as reviewed in Appendix A.2, at $\delta = 0$ the field $\phi(x)$ is just the boundary value of a free boson in 2d.

If we take $n_i = \pm 1$, and introduce the simplified notation $V_\pm \equiv V_{\pm 1}$, the only non-vanishing correlators have an equal number of $V_+$ and $V_-$, resulting in

$$\langle V_+(x_1)\cdots V_+(x_n)V_-(x_1')\cdots V_-(x_n')\rangle_0 = \frac{\prod_{j<k} |x_j - x_k|^{2b_0^2}|x_j' - x_k'|^{2b_0^2}}{\prod_{j,k} |x_j - x_k'|^{2b_0^2}} \tag{3.4}$$

$$= a^{-2nb_0^2} e^{2b_0^2\left(\sum_{j<k}(\log(|x_j - x_k|/a) + \log(|x_j' - x_k'|/a)) - \sum_{j,k} \log(|x_j - x_k'|/a)\right)}\,.$$

In the last expression, we recognize the structure of the Boltzman weight for the Coulomb gas in eq. (2.18), except for the ordering constraint. Fitting with the Coulomb gas interpretation of such correlators, it is common to refer to the parameters $n_i$ as "charges" and to the constraint on the total charge as "neutrality condition".

At $\delta > 0$, we can similarly obtain the correlators of vertex operators by using a standard trick (e.g. proposition 23.6.1 of [61]). For a GFF $\phi$ of covariance $C(x)$, we have the formula

$$\langle e^{i\phi[f]}\rangle_0 = e^{-\frac{1}{2}\langle\phi[f]\phi[f]\rangle_0}\,, \tag{3.5}$$

where $\phi[f] \equiv \int dx\, \phi(x) f(x)$. Therefore, choosing $f(x) = \sum_i \alpha_i \delta(x - x_i)$, where the test function constraint $\int dx\, f(x) = 0$ (see Appendix A.2) demands that the neutrality condition $\sum_i \alpha_i = 0$ be satisfied, we have

$$\langle e^{i\alpha_1\phi(x_1)}\cdots e^{i\alpha_n\phi(x_n)}\rangle_0 = e^{-\frac{1}{2}C(0)\sum_i \alpha_i^2} \prod_{i<j} e^{-\alpha_i\alpha_j C(x_i - x_j)}\,. \tag{3.6}$$

Since $C(x)$ is defined only up to a constant, we could set $C(0)$ to zero. However, for the $\delta \to 0$ limit we choose $C(x)$ as in (3.2), and thus absorb $C(0)$ by a normal ordering of the vertex operators, consistently with the $\delta \to 0$ limit, where such normal ordering is needed because of UV divergences (see footnote 14). We thus define the vertex operators at $\delta > 0$ as:

$$V_n(x) = \kappa^{n^2} : e^{inb_0\phi(x)} := \kappa^{n^2} e^{\frac{1}{2}C(0)n^2b_0^2} e^{inb_0\phi(x)}\,. \tag{3.7}$$

Notice that while (3.6) is independent of $\kappa$, the choice of normalization above will introduce a $\kappa$-dependence in correlators of vertex operators, due to shift symmetry. This choice, which appears to be slightly unnatural at fixed $\delta > 0$, is the correct one if we wish to recover the $\delta = 0$ case in a smooth way.

At this point it should be clear that, identifying $b_0^2 a^\delta = \mathcal{J}$, the Boltzmann weight of a configuration of $2n$ kinks and anti-kinks in the AYK model in eq. (2.20) is proportional to the correlation function of

---

[14]The standard definition of normal ordering gives:

$$V_n(x) =: e^{inb_0\phi(x)} :\equiv \exp\left\{-\frac{1}{2}\int dx_1 dx_2 \frac{\delta}{\delta\phi(x_1)} C(x_1 - x_2) \frac{\delta}{\delta\phi(x_2)}\right\} e^{inb_0\phi(x)} = \exp\left\{\frac{n^2b_0^2}{2}C(0)\right\} e^{inb_0\phi(x)}\,,$$

where we typically regularize $C(0)$ introducing a cutoff $a$ and replacing $C(0) \to C(a)$. We further multiply by $\kappa^{n^2b_0^2}$, so that by virtue of the neutrality condition $\sum_i n_i = 0$ the correlators are independent of $\kappa$.

$2n$ vertex operators, with alternating charges, for a 1d GFF $\phi$ of negative dimension $\Delta_\phi = -\delta/2$, i.e. with covariance (3.2):

$$\langle V_+(x_1)V_-(x_2)\dots V_+(x_{2n-1})V_-(x_{2n})\rangle_0 = \kappa^{2n}e^{2b_0^2 \sum_{i<j} \frac{(-1)^{i-j}}{\delta}(|x_i-x_j|^\delta - \kappa^{-\delta})} \,. \tag{3.8}$$

Therefore, in order to generate the terms appearing in the AYK partition function, we need to perturb the GFF by the vertex operators. However, in order to force the charges to alternate, we will need to form a product of the vertex operators with an appropriate algebra of noncommuting operators. Coincidentally, such algebra will be provided precisely by the Pauli matrices appearing in the 1d SRI.

## 3.2 Interacting model

We claim that the 1d LRI CFT can be identified with the IR fixed point of the following 1d continuum model:

$$Z_s(b,g) = \Big\langle \operatorname{tr} \operatorname{Pexp}\Big\{ \int_{-L/2}^{L/2} \mathrm{d}x \, [g\,\mathcal{O}_g(x) + h\,\mathcal{O}_h(x)] \Big\} \Big\rangle_0 , \tag{3.9}$$

where $b = b_0 - \sqrt{2}h$, and $b_0 = \kappa^{\delta/2}$, for some arbitrary mass scale $\kappa$. The operators in the exponent are

$$\mathcal{O}_g(x) \equiv \hat{\sigma}_+ V_+(x) + \hat{\sigma}_- V_-(x) , \qquad \mathcal{O}_h(x) \equiv \hat{\sigma}_3 \chi(x) , \qquad \chi(x) \equiv \frac{\mathrm{i}}{\sqrt{2}}\partial_x\phi(x) , \tag{3.10}$$

with $\hat{\sigma}_\pm = \frac{1}{2}(\hat{\sigma}_1 \pm \mathrm{i}\hat{\sigma}_2)$. Here $\hat{\sigma}_{i=1,2,3}$ are the Pauli matrices

$$\hat{\sigma}_1 = \begin{pmatrix} 0 & 1 \\ 1 & 0 \end{pmatrix} , \quad \hat{\sigma}_2 = \begin{pmatrix} 0 & -\mathrm{i} \\ \mathrm{i} & 0 \end{pmatrix} , \quad \hat{\sigma}_3 = \begin{pmatrix} 1 & 0 \\ 0 & -1 \end{pmatrix} . \tag{3.11}$$

Lastly, $\operatorname{tr} \operatorname{Pexp}$ in (3.9) is the trace of the path-ordered exponential of the $2 \times 2$ matrix in curly brackets. As a reminder, the path ordering means

$$\begin{aligned}
\operatorname{P}\exp\Big(g \int \mathrm{d}x\, A(x)\Big) &= \sum_{n\geq 0} \frac{g^n}{n!}\operatorname{P}\Big(\int \mathrm{d}x\, A(x)\Big)^n \\
&= \sum_{n\geq 0} g^n \int_{x_1 \geq x_2 \geq \dots \geq x_n} \Big(\prod_{i=1}^n \mathrm{d}x_i\Big) A(x_1)\cdots A(x_n) .
\end{aligned} \tag{3.12}$$

It is convenient to define the defect operator

$$\mathcal{D}(x_j, x_i) \equiv \operatorname{P}\exp\Big( \int_{x_i}^{x_j} \mathrm{d}x\, [g\,\mathcal{O}_g(x) + h\,\mathcal{O}_h(x)]\Big) , \tag{3.13}$$

so that the partition function (3.9) is[15]

$$Z_s(b,g) = \operatorname{tr} \big\langle \mathcal{D}(L/2, -L/2)\big\rangle_0 . \tag{3.14}$$

General correlators of operators $\mathcal{O}_1(x_1),\dots,\mathcal{O}_n(x_n)$, with $L/2 > x_1 > x_2 > \dots > x_n > -L/2$, have the form

$$\langle \mathcal{O}_1(x_1)\cdots \mathcal{O}_n(x_n)\rangle_\mathcal{D} \equiv \frac{1}{Z_s} \operatorname{tr} \big\langle \mathcal{D}(L/2, x_1)\mathcal{O}_1(x_1)\mathcal{D}(x_1, x_2)\mathcal{O}_2(x_2)\cdots \mathcal{O}_n(x_n)\mathcal{D}(x_n, -L/2)\big\rangle_0 , \tag{3.15}$$

---

[15]Notice that in the periodic case, at $s = 1$, $\mathcal{D}(L/2, -L/2)$ is a special case (spin $1/2$) of the "monodromy matrix" of [62], that was used in [53] to establish a relation between the partition functions of the Kondo and boundary sine-Gordon models.

where the operators $\mathcal{O}_i(x)$ are in general matrix valued.

We have restricted the interaction to a finite region, i.e. the interval $[-L/2, L/2]$, in order to regularize IR divergences. Any other interval of length $L$ would be equivalent, due to the translation invariance of the unperturbed theory. We stress that we do not assume periodicity. As explained in Section 2.3, this IR regularization should rather be thought as imposing that the LRI model has no kinks in the complement of the interval $[-L/2, L/2]$. It is also understood that besides tuning to an IR fixed point, the 1d LRI CFT should be obtained in the limit $L \to \infty$.

The first justification of our claim is that we reproduce the AYK partition function (2.20) by expanding (3.9) in $g$. In order to see that, let us first set $h = 0$, and expand $Z_s(b_0, g)$ in powers of $g$. Using (3.12), with $A = \mathcal{O}_g$, we find precisely (2.20), upon the identification $b_0^2 = \mathcal{J}$. Indeed, since $\hat{\sigma}_+^2 = \hat{\sigma}_-^2 = 0$, only alternating chains of kink and antikink operators $\hat{\sigma}_+ V_+(x)$ and $\hat{\sigma}_- V_-(x)$ survive. Turning on $h$, we will show below that it is equivalent to shifting $b_0$ to $b_0 - \sqrt{2}h$, hence the partition function only depends on the combination $b = b_0 - \sqrt{2}h$ and the correspondence with AYK still holds, with $\mathcal{J} = b^2$. However, keeping the $\mathcal{O}_h$ term will prove essential in order to consistently renormalize the model.

The partition function (3.9) is similar to those encountered in impurity models [63, 64], and the path ordering of $2 \times 2$ matrices can be traded for a path integral using a complex bosonic spinor $z(x) = \{z_1(x), z_2(x)\}$ subject to $\bar{z}(x)z(x) = 1$, similarly to [65, 63, 64]. We will review this formulation in Appendix C.1. See also Appendix C.3 for an alternative formulation in terms of a nonlinear sigma model.

Since the covariance of the GFF $\phi$ agrees with that of a free field in $s+1 = 2-\delta$ dimensions, our model can be formally interpreted as a one-dimensional defect for the $(s+1)$-dimensional free theory. At $s = 1$, the bulk is two-dimensional and the defect is a boundary condition, with the vertex operators becoming conformal primaries of the Gaussian theory. The resulting $s = 1$ model then becomes the bosonized version of the Kondo model [66, 53], already related to the LRI model by Anderson and Yuval [22]. However, already at $s = 1$, a number of puzzles had not been addressed before, namely the field theoretic derivation of the beta functions and the mismatch of operators between the Kondo fixed points and the LRI CFT. Therefore, our ensuing RG analysis and identification of the spectrum of operators are novel even at $s = 1$.

For $s = 1 - \delta < 1$, the model (3.9) is a genuinely new theory. It might seem rather unorthodox, since $\Delta_\phi < 0$ and since the vertex operators do not have a definite scaling dimension in the UV. Nevertheless, it is consistent, as it should be be clear from the previous discussion of the GFF with negative dimension, and from the analysis presented in the rest of the paper.[16]

### 3.2.1 Relation between $h$ and $b$

Given that $\mathcal{O}_h(x)$ is a total derivative, it would be tempting to conclude that its contribution vanishes in the case of cyclic boundary, and that it reduces to the insertion of new vertex operators at the boundaries in the open case. However, such a conclusion would be wrong, as it does not take into account the path ordering of $\mathcal{O}_h$ with respect to $\mathcal{O}_g$, and their noncommutativity. As we will now show, a proper treatment of this interaction leads to uncovering a relation between the coupling $h$ and the charge $b$ of the vertex operators.

In order to see that, we apply formula (3.15) to the perturbative expansion itself. Namely, if we expand only in $g$, introducing a UV cutoff $a$, we obtain

$$Z_s = 2 \sum_{n=0}^{+\infty} g^{2n} \int_{\mathrm{I}_{2n}(3a)} \mathrm{tr} \langle \mathcal{D}_h(L/2, x_1 + a)\hat{\sigma}_+ V_+(x_1)\mathcal{D}_h(x_1 - a, x_2 + a)\hat{\sigma}_- V_-(x_2) \cdots$$
$$\cdots \hat{\sigma}_- V_-(x_{2n})\mathcal{D}_h(x_{2n} - a, -L/2)\rangle_0 \,, \tag{3.16}$$

where the integration domain was defined in (2.19) (from now on we occasionally omit the Lebesgue

---

[16]We notice moreover that similar type of non-polynomial non-scaling operators appears also in other contexts, such as in Lagrangians of nonlinear sigma models and in low-energy effective actions of $QED_3$ [67, 68].

measure to slim down long expressions), and we introduced the partial defect operator

$$\mathcal{D}_h(x_j, x_i) \equiv \mathcal{D}(x_j, x_i)_{|g=0} = \mathrm{P} \exp\left( h\,\hat{\sigma}_3 \int_{x_i}^{x_j} \mathrm{d}x\,\chi(x) \right) . \tag{3.17}$$

At this stage, having decoupled the nonncommuting operators, we can perform the integration in the exponent, to get

$$\mathcal{D}_h(x_j, x_i) = \exp\left( \frac{\mathrm{i}h}{\sqrt{2}}\,\hat{\sigma}_3\,(\phi(x_j) - \phi(x_i)) \right) , \tag{3.18}$$

and then use the commutation relation

$$e^{-\mathrm{i}\alpha\hat{\sigma}_3}\,\hat{\sigma}_\pm\,e^{+\mathrm{i}\alpha\hat{\sigma}_3} = \hat{\sigma}_\pm\,e^{\mp 2\mathrm{i}\alpha} , \tag{3.19}$$

to write

$$e^{-\frac{\mathrm{i}h}{\sqrt{2}}\hat{\sigma}_3\phi(x+a)}\,\hat{\sigma}_\pm V_\pm(x)\,e^{+\frac{\mathrm{i}h}{\sqrt{2}}\hat{\sigma}_3\phi(x-a)} = \hat{\sigma}_\pm V_\pm(x)\,e^{\mp\sqrt{2}\mathrm{i}h\phi(x+a)}\,e^{+\frac{\mathrm{i}h}{\sqrt{2}}\hat{\sigma}_3(\phi(x-a)-\phi(x+a))} ,$$

$$\xrightarrow[a\to 0]{} \hat{\sigma}_\pm V_{\pm(1-\sqrt{2}h/b_0)}(x)\,e^{h(b_0\sqrt{2}-h)C(0)} , \tag{3.20}$$

where in the last step we took into account the definition of normal ordered vertex operator (3.7). Therefore, for the partition function we obtain

$$Z_s = 2\sum_{n=0}^{+\infty} g_h^{2n} \int_{\mathrm{I}_{2n}(a)} \mathrm{tr}\left\langle e^{\frac{\mathrm{i}h}{\sqrt{2}}\hat{\sigma}_3\,\phi(L/2)}\hat{\sigma}_+ V_{1-\sqrt{2}h/b_0}(x_1)\cdots\hat{\sigma}_- V_{-1+\sqrt{2}h/b_0}(x_{2n})e^{-\frac{\mathrm{i}h}{\sqrt{2}}\hat{\sigma}_3\,\phi(-L/2)} \right\rangle_0$$

$$= 2\sum_{n=0}^{+\infty} g_h^{2n} \int_{\mathrm{I}_{2n}(a)} \mathrm{tr}\left\langle \hat{\sigma}_+ V_{1-\sqrt{2}h/b_0}(x_1)\cdots\hat{\sigma}_- V_{-1+\sqrt{2}h/b_0}(x_{2n})\mathcal{D}_h(L/2, -L/2) \right\rangle_0 , \tag{3.21}$$

where we have defined the multiplicatively renormalized coupling

$$g_h = g\,e^{h(b_0\sqrt{2}-h)C(0)} . \tag{3.22}$$

At $\delta = 0$, choosing the regularization $C(0) \to C(a) = -2\log(\kappa a)$, inverting the relation between $g$ and $g_h$, and expanding in powers of $h$, we find

$$g = g_h \left( 1 + 2h(\sqrt{2} - h)\log(\kappa a) + 4h^2\log^2(\kappa a) + O(h^3) \right) . \tag{3.23}$$

Therefore, up to a boundary contribution,[17] we find that the effect of the new interaction term is equivalent to shifting $b_0 \to b \equiv b_0 - h\sqrt{2}$, and renormalizing the coupling $g$. This implies that we can trade a change in $b_0$ for a change in $h$, or viceversa. This fact is often used in the Kondo model (i.e. at $\delta = 0$) in order to set $h = 0$.[18] We will instead set $b_0$ to our favorite value (in particular, such that $b_0 = 1$ when $\delta = 0$) and keep $h \neq 0$ in our model. As we will see in Section 4, the choice of keeping a nonvaninshing $h$ is forced upon us by the renormalization flow: indeed the choice $h = 0$ is not stable under renormalization.

---

[17]If we had defined the model on a circle, i.e. with periodic boundary conditions at $x = -L/2$ and $x = L/2$, the factor $\mathcal{D}_h(L/2, -L/2)$ would be trivial if $\phi$ was non-compact, while it would give a nontrivial monodromy in the compact case.

[18]The fact that one can trade $b$ for $h$, or vice versa, is mentioned in various old papers about the Kondo model, but typically it is either stated without explanation (e.g. [22, 53]) or expressed as a canonical transformation on the quantum Hamiltonian (e.g. [66, 69, 70]). We have not found a proof in the literature that uses the path integral language as does the one provided here.

### 3.2.2 Covariant derivative and Schwinger-Dyson equation

The path ordering introduces some non-trivial dependence of correlators on the end-points of the partial defect operator $\mathcal{D}$ appearing in the right-hand side of (3.15). Following [64], it is convenient for this reason to define a defect covariant derivative $D_x$, by demanding that

$$\operatorname{tr} \langle \ldots D_x \mathcal{O}(x) \ldots \rangle_0 = \frac{d}{dx} \operatorname{tr} \langle \ldots \mathcal{O}(x) \ldots \rangle_0 \ , \tag{3.24}$$

which, because of (3.15), is equivalent to

$$\mathcal{D}(x', x) D_x \mathcal{O}(x) \mathcal{D}(x, x'') \equiv \frac{d}{dx} \left( \mathcal{D}(x', x) \mathcal{O}(x) \mathcal{D}(x, x'') \right) \ . \tag{3.25}$$

For the model at hand we easily find

$$D_x \mathcal{O}(x) = \partial_x \mathcal{O}(x) - [g \mathcal{O}_g + h \mathcal{O}_h, \mathcal{O}](x) \ , \tag{3.26}$$

where $[,]$ is the matrix commutator.

The covariant derivative allows us to understand if some operator is a primary or a descendant at the conformal fixed point. Interestingly, $D_x O$ can be a descendant of $O$ even if $O$ has no explicit dependence on the defect coordinate. An important example for us will be the case of $O = \hat{\sigma}_3$, for which we have

$$D_x \hat{\sigma}_3 = 2g(\hat{\sigma}_+ V_+ - \hat{\sigma}_- V_-) \ . \tag{3.27}$$

This also motivates the following notation. Since in the $\langle \ldots \rangle$ notation the insertion of operators without explicit $x$-dependence is ambiguous, we will sometime write $\hat{\sigma}_i(x)$ for the insertion of a Pauli matrix at point $x$. For example, for $L/2 > x_1 > x_2 > -L/2$ we have:

$$\langle \hat{\sigma}_3(x_1) \hat{\sigma}_3(x_2) \rangle_{\mathcal{D}} = \frac{1}{Z_s} \operatorname{tr} \langle \mathcal{D}(L/2, x_1) \hat{\sigma}_3 \mathcal{D}(x_1, x_2) \hat{\sigma}_3 \mathcal{D}(x_2, -L/2) \rangle_0 \ . \tag{3.28}$$

We can now derive Schwinger-Dyson equations for our model, assuming for simplicity periodic boundary conditions. As usual, we perform an arbitrary infinitesimal variation $\phi(x) \to \phi(x) + \varepsilon(x)$ in the functional integral, expand to linear order in $\varepsilon(x)$ and demand that the correlators are invariant, because it is just a change of variable in the functional integral, where the measure is translation invariant. Taking into account the GFF action (A.15), we obtain

$$0 = - \int_{-L/2}^{L/2} dx \, \varepsilon(x) \langle C^{-1} \cdot \phi(x) \ldots \rangle_{\mathcal{D}} + \int_{-L/2}^{L/2} dx \, \varepsilon(x) \, \mathrm{i}\, b_0 \, g \langle \ldots (\hat{\sigma}_+ V_+ - \hat{\sigma}_- V_-)(x) \ldots \rangle_{\mathcal{D}}$$
$$+ \int_{-L/2}^{L/2} dx \, \frac{\mathrm{i} h}{\sqrt{2}} \langle \ldots \hat{\sigma}_3 \, \partial_x \varepsilon(x) \ldots \rangle_{\mathcal{D}} \ , \tag{3.29}$$

up to contact terms involving other inserted operators. Integration by parts on the last term leads to

$$\langle C^{-1} \cdot \phi(x) \ldots \rangle_{\mathcal{D}} = \mathrm{i}\, b_0 \, g \langle \ldots (\hat{\sigma}_+ V_+ - \hat{\sigma}_- V_-)(x) \ldots \rangle_{\mathcal{D}} - \frac{\mathrm{i} h}{\sqrt{2}} \langle \ldots D_x \hat{\sigma}_3(x) \ldots \rangle_{\mathcal{D}}$$
$$= \mathrm{i} g (b_0 - \sqrt{2} h) \langle \ldots (\hat{\sigma}_+ V_+ - \hat{\sigma}_- V_-)(x) \ldots \rangle_{\mathcal{D}} \ , \tag{3.30}$$

where in the last step we used the covariant derivative introduced above. Notice that we find again the combination $b_0 - \sqrt{2} h$, confirming again that the role of $h$ is to shift $b_0$ by $-\sqrt{2} h$.

The Schwinger-Dyson equations (3.30), together with the fact that the inverse covariance $C^{-1}$ acts on $\phi$ as a shadow transform, imply that at the IR fixed point, if the latter is non-trivial, the spectrum of the model contains a protected operator of scaling dimension $1 - \Delta_\phi$, odd under both $\mathbb{Z}_2$ and parity. Notably, this cannot be a primary, as in virtue of (3.27) it must be a descendant of $\hat{\sigma}_3$. As we will see, for the $s = 1$ LRI the only IR fixed point is the trivial one, and the two primaries do not recombine. On the other hand, for the model at $s = 1 - \delta$, there is a non-trivial IR fixed point and the argument above implies the existence of a protected operator of dimension $1 + \delta/2$, which must be a descendant of $\hat{\sigma}_3$. Moreover, the latter must also have protected dimension $\Delta_\sigma = \delta/2$.

### 3.2.3 Symmetries

For $g = h = 0$, the UV theory (3.9) has O(2) global symmetry acting on the target space circle of $\phi$, and PU(2) global symmetry acting on the $2 \times 2$ matrix degrees of freedom $A$ by conjugation. By turning on $\mathcal{O}_g$ and $\mathcal{O}_h$, O(2) $\times$ PU(2) is broken to the diagonal O(2), generated by U(1) rotations and $\mathbb{Z}_2$ reflection

$$\text{U}(1): \qquad \phi(x) \mapsto \phi(x) + \alpha/b_0\,, \qquad\qquad A \mapsto e^{-\mathrm{i}\frac{\alpha}{2}\hat{\sigma}_3} A e^{\mathrm{i}\frac{\alpha}{2}\hat{\sigma}_3}\,, \qquad\qquad (3.31)$$

$$\mathbb{Z}_2: \qquad \phi(x) \mapsto -\phi(x)\,, \qquad\qquad A \mapsto \hat{\sigma}_1 A \hat{\sigma}_1\,, \qquad\qquad (3.32)$$

where $\alpha \in \mathbb{R}/2\pi\mathbb{Z}$. The model (3.9) is natural because $\mathcal{O}_g$ and $\mathcal{O}_h$ are the only relevant or marginal operators invariant under this O(2).

The model also respects parity $x \mapsto -x$. In order for it to be compatible with the path ordering, and to preserve $\mathcal{O}_g$ and $\mathcal{O}_h$, it must act as

$$\text{parity}: \quad \phi(x) \mapsto -\phi(-x)\,, \quad A \mapsto A^T\,. \qquad\qquad (3.33)$$

In the next section, we will confirm the existence of an IR fixed point of (3.9) at $b = 1 + O(\delta)$, $g = O(\sqrt{\delta})$. The fixed-point theory thus exhibits O(2) $\rtimes$ parity symmetry.

From the point of view of the LRI model, the $\mathbb{Z}_2$ symmetry can be understood as the usual $\mathbb{Z}_2$ symmetry for the original Ising spin in the absence of an external magnetic field. Indeed $\mathbb{Z}_2$ maps kinks to antikinks, and vice versa, hence it must swap up-spins with down-spins. Similarly, parity is in direct correspondence with parity on the Ising side.

On the other hand, the $U(1)$ symmetry has no counterpart in the LRI model. Therefore, we should assume that the latter corresponds to a superselection sector of the model (3.9) with only operators that are neutral under $U(1)$. This is also known as the singlet sector, and we can restrict to it by gauging the model, as discussed in Appendix C.

$U(1)$-invariance has the important consequence of excluding any vertex operators with charge $n_i$ different from $\pm 1$. Indeed, under the U(1) symmetry, operators have the following charges

$$
\begin{array}{c|c}
\text{operator} & \text{U}(1) \text{ charge} \\
\hline
\partial\phi & 0 \\
V_n & n \\
\hat{\sigma}_3 & 0 \\
\hat{\sigma}_\pm & \mp 1\,.
\end{array}
\qquad\qquad (3.34)
$$

This means that the space of U(1)-neutral local operators is generated by

$$A, \quad B\,\hat{\sigma}_3, \quad C\,V_+\hat{\sigma}_+, \quad D\,V_-\hat{\sigma}_-\,, \qquad\qquad (3.35)$$

where $A, B, C, D$ are words built out of the letters $\partial^n \phi$ with $n > 0$.

At first sight, such a restriction might seem inconsistent, because already at $g = h = 0$, i.e. in the GFF, the OPE of $V_+$ with itself contains $V_{-2}$. However, $V_+$ by itself is not in the spectrum of the singlet sector: neutrality under $U(1)$ requires that $V_+$ be always multiplied by $\hat{\sigma}_+$. And the product of $\hat{\sigma}_+ V_+$ with itself vanishes because $\hat{\sigma}_+^2 = 0$.

## 4 Renormalization group analysis

In this section, we discuss some perturbative computations for the model introduced in the previous section, which we rewrite here for convenience:

$$Z_s(b,g) = \left\langle \operatorname{tr} \operatorname{Pexp}\left\{ \int_{-L/2}^{L/2} \mathrm{d}x \, [g\,\mathcal{O}_g(x) + h\,\mathcal{O}_h(x)] \right\} \right\rangle_0\,. \qquad\qquad (4.1)$$

Expectation values $\langle\ldots\rangle_0$ are computed with respect to the GFF theory (A.15), with covariance (3.2). The operators $\mathcal{O}_g$, $\mathcal{O}_h$ are defined in equation (3.10). We recall that the partition function depends on $g$ and $b = b_0 - \sqrt{2}h$, of mass dimension 0 and $\delta/2$, respectively. At $\delta = 0$, we are free to choose $b_0 = 1$, so that in this case the UV scaling dimension of both $\mathcal{O}_g$ and $\mathcal{O}_h$ equals 1, but we should keep in mind that for $\delta > 0$, the coupling $b_0$ has positive mass dimension and thus it has a (purely classical) running.

The UV divergences of the perturbative expansion are regularized by point-splitting, i.e. by introducing hard core repulsions as in (2.19), with cutoff length $a$. The UV divergences in the $a \to 0$ limit are then removed by including the appropriate counterterms, so that correlation functions of renormalized operators are UV-finite when expressed in terms of the renormalized couplings. Beta functions and anomalous dimensions are then obtained from the Callan–Symanzik equation, which states that renormalized observables are cut-off independent.

We shall distinguish between the model with $s = 1$, in which case a free boundary CFT is perturbed by marginal primary operators, from the model with $s < 1$, in which case a GFF with a compact and dimensionful target space is perturbed by composite operators that are not scaling operators. In the latter case we do not have at the moment a proof that the theory is renormalizable to all orders of the perturbative expansion, but we also found no evidence to the contrary in our computations. Although we hope to come back to this question in the future, it should be said that the UV completeness of the model is not needed for our purposes, as we are only interested in its IR behavior.

## 4.1  Beta functions and fixed point at $s = 1$

We compute the beta functions for the model with $s = 1$ by employing conformal perturbation theory (see e.g. [71] for an extensive review). The basic idea is that it is sufficient to look at perturbative expansion of one point functions of the perturbing operators, inserted at infinity. For an operator $\mathcal{O}(x)$ with dimension $\Delta_{\mathcal{O}}$ in a given CFT, its insertion at infinity is defined as

$$\langle \mathcal{O}(\infty)\ldots\rangle_{\mathrm{CFT}} \equiv \lim_{x \to \infty} x^{2\Delta_{\mathcal{O}}}\langle \mathcal{O}(x)\ldots\rangle_{\mathrm{CFT}}, \tag{4.2}$$

and its finiteness is a simple consequence of the conformal properties of correlators.

In our case, we will be considering $\langle\mathcal{O}_g\rangle_{\mathcal{D}}$ and $\langle\mathcal{O}_h\rangle_{\mathcal{D}}$, where the expectation value $\langle\ldots\rangle_{\mathcal{D}}$ has been defined in (3.15), and it is here understood to be evaluated perturbatively in $g$ and $h$. The fact that correlators appearing in such expansion are the GFF correlators, and hence they are all conformal at $\delta = 0$, guarantees the existence of the limit.

In this setup, as long as the IR cutoff $L$ is kept finite, UV divergences in the $a \to 0$ limit can always be removed via coupling renormalization. Indeed we do not need to renormalize the operator inserted at $\infty$, as it will never be at coinciding point with an operator from the interaction (as long as this is restricted to a finite interval), and thus it will be sufficient to renormalize the coupling.

Expanding to quadratic order in the couplings, we have

$$
\begin{aligned}
\langle\hat{\sigma}_3\chi(\infty)\rangle_{\mathcal{D}} =& \frac{h}{2}\,\mathrm{tr}(\hat{\sigma}_3^2)\int_{-L/2}^{L/2}\mathrm{d}x\langle\chi(\infty)\chi(x)\rangle_0 \\
&+ \frac{g^2}{2}\int_{-L/2+a}^{L/2}\mathrm{d}x_1\int_{-L/2}^{x_1-a}\mathrm{d}x_2\,\mathrm{tr}[\langle\hat{\sigma}_3\chi(\infty)\hat{\sigma}_+V_+(x_1)\hat{\sigma}_-V_-(x_2)\rangle_0 + (12)] \\
&+ \ldots \\
=& \, h\,L + \frac{g^2}{2}I_1 + \ldots,
\end{aligned}
\tag{4.3}
$$

$$\langle \hat{\sigma}_+ V_+(\infty)\rangle_{\mathcal{D}} = \frac{g}{2}\,\mathrm{tr}(\hat{\sigma}_+\hat{\sigma}_-)\int_{-L/2}^{L/2} dx\,\langle V_+(\infty)V_-(x)\rangle_0$$

$$+ \frac{gh}{2}\int_{-L/2+a}^{L/2} dx_1 \int_{-L/2}^{x_1-a} dx_2\,\mathrm{tr}\left[\langle \hat{\sigma}_+ V_+(\infty)\hat{\sigma}_- V_-(x_1)\hat{\sigma}_3\chi(x_2)\rangle_0 + (12)\right]$$

$$+ \dots \tag{4.4}$$

$$= \frac{g}{2}\,L + \frac{gh}{2}\,I_2 + \dots \,,$$

where $(ij)$ means exchanging the operators located at $x_i$ and $x_j$ while preserving the path ordering. The factors of $1/2$ on the right-hand side are due to the partition function in the denominator appearing in the definition of the expectation value (3.15).

For the terms linear in the couplings, we used that $\hat{\sigma}_3^2 = \mathbb{1}$ (the $2\times 2$ identity matrix), $\mathrm{tr}(\hat{\sigma}_+\hat{\sigma}_-) = 1$, and that the two-point functions of $\chi$ and $V_\pm$ are unit normalized.

The three-point functions are given in appendix D, and their integrals evaluate to:

$$I_1 = I_2 = -2\sqrt{2}L(1 + \log(a/L)) + 2\sqrt{2}a\,. \tag{4.5}$$

Therefore, eq. (4.3) and (4.4) feature UV divergences as $a \to 0$, which we can remove by expressing the bare couplings in terms of renormalized couplings (denoted with subscript r) as follows:

$$g = g_{\mathrm{r}} + 2\sqrt{2}g_{\mathrm{r}}h_{\mathrm{r}}\log(a/L)\,, \quad h = h_{\mathrm{r}} + \sqrt{2}g_{\mathrm{r}}^2\log(a/L)\,. \tag{4.6}$$

At the next (i.e. cubic) order in the perturbative expansion, we have:

$$\langle \hat{\sigma}_3\chi(\infty)\rangle_{\mathcal{D}}\big|_{\text{cubic order}} = \frac{hg^2}{2}\int_{\mathrm{I3}(a)}\mathrm{tr}\left[\langle \hat{\sigma}_3\chi(\infty)\hat{\sigma}_+ V_+(x_1)\hat{\sigma}_- V_-(x_2)\hat{\sigma}_3\chi(x_3)\rangle_{0,c} + 5\ \text{perm.}\right]$$

$$+ \frac{h^3}{2}\int_{\mathrm{I3}(a)}\mathrm{tr}\,\langle \hat{\sigma}_3\chi(\infty)\hat{\sigma}_3\chi(x_1)\hat{\sigma}_3\chi(x_2)\hat{\sigma}_3\chi(x_3)\rangle_{0,c}$$

$$\equiv hg^2 L\,I_3 + h^3 L\,I_4\,,$$

$$\langle \hat{\sigma}_+ V_+(\infty)\rangle_{\mathcal{D}}\big|_{\text{cubic order}} = \frac{gh^2}{2}\int_{\mathrm{I3}(a)}\mathrm{tr}\left[\langle \hat{\sigma}_+ V_+(\infty)\hat{\sigma}_- V_-(x_1)\hat{\sigma}_3\chi(x_2)\hat{\sigma}_3\chi(x_3)\rangle_{0,c} + 2\ \text{perm.}\right]$$

$$+ \frac{g^3}{2}\int_{\mathrm{I3}(a)}\mathrm{tr}\,\langle \hat{\sigma}_+ V_+(\infty)\hat{\sigma}_- V_-(x_1)\hat{\sigma}_+ V_+(x_2)\hat{\sigma}_+ V_-(x_3)\rangle_{0,c}$$

$$\equiv gh^2 L\,I_5 + g^3 L\,I_6\,, \tag{4.7}$$

where the integration domain was defined in (2.19), and $\langle \dots \rangle_{0,c}$ means that we are taking the (order-respecting) connected correlator, obtained by taking into account the expansion of the partition function in the denominator of the expectation value (see again (3.15)). In the case of four-point functions for generic operators $O_i$, the connected correlator is:

$$\int_{\mathrm{I3}(a)}\mathrm{tr}\langle O_\infty(\infty)O_1(x_1)O_2(x_2)O_3(x_3)\rangle_{0,c} \equiv \int_{\mathrm{I3}(a)}\mathrm{tr}\langle O_\infty(\infty)O_1(x_1)O_2(x_2)O_3(x_3)\rangle_0 \tag{4.8}$$

$$- \int_{-L/2}^{L/2} dx_1\,\mathrm{tr}\langle O_\infty(\infty)O_1(x_1)\rangle_0 \int_{\mathrm{I2}(a)}\mathrm{tr}\langle O_2(x_2)O_3(x_3)\rangle_0\,,$$

where we used the fact that in the unperturbed CFT one-point functions vanish. Evaluating the integrals, we find:

$$I_3 = 2I_5 = 4\log^2(a/L) + 10\log(a/L) + 10 - \frac{2\pi^2}{3} + \log 2 + \dots\,,$$

$$I_4 = -2 - \log 2 + \dots$$

$$I_6 = \log^2(a/L) + 5\log(a/L)/2 - \left(\pi^2 - 12\right)/6 + \dots\,, \tag{4.9}$$

where we have implicitly subtracted power-law divergences, and the dots stand for regular terms as $a \to 0$.

Plugging these integrals into (4.7), the relation between bare and renormalized couplings at cubic order is found to be:

$$g = g_\mathrm{r} + g_\mathrm{r} \left(2\sqrt{2}h_\mathrm{r} - g_\mathrm{r}^2 - 2h_\mathrm{r}^2\right) \log(a/L) + 2g_\mathrm{r} \left(g_\mathrm{r}^2 + 2h_\mathrm{r}^2\right) \log^2(a/L),$$
$$h = h_\mathrm{r} + g_\mathrm{r}^2 \left(\sqrt{2} - 2h_\mathrm{r}\right) \log(a/L) + 4g_\mathrm{r}^2 h_\mathrm{r} \log^2(a/L). \tag{4.10}$$

Note that the terms linear in $g_\mathrm{r}$ in the first equation exactly reproduce (3.23), up to a finite redefinition of $g_r$ (unless $\kappa = 1/L$), thus providing a useful crosscheck. However, we should stress the importance of keeping $h$ in the action, rather than eliminating it in favor of a shifted $b$. Indeed we see that the $g^3$ term, i.e. $I_6$, has a double-log divergence, and as there is no $g^2$ term in the renormalization of $g$, the $gh$ term is crucial for the cancellation of such term, and thus for the consistency of beta functions.

The beta functions for the bare couplings are given by their derivative with respect to $\log(1/a)$ at fixed renormalized couplings:[19]

$$\beta_g = -a\frac{dg}{da}\,, \quad \beta_h = -a\frac{dh}{da}\,, \tag{4.11}$$

which, upon re-expressing everything in terms of the bare couplings, give

$$\beta_g = -2gh(\sqrt{2} - h) + g^3 + O(g^3 h, g^5)\,, \quad \beta_h = -g^2(\sqrt{2} - 2h) + O(g^2 h^2, g^4)\,. \tag{4.12}$$

Two comments are in order. First, we notice that the beta functions are invariant under the reflection $h \to \sqrt{2} - h$. Remembering that $b = 1 - \sqrt{2}h$ (cfr. Section 3.2.1), this is nothing but the invariance under $b \to -b$ of the model. Second, in terms of $b = 1 - \sqrt{2}h$, the beta functions translate to

$$\beta_g = (b^2 - 1)g + g^3, \quad \beta_{b^2} = 4b^2 g^2\,, \tag{4.13}$$

which at leading order agree with (2.21), at $s = 1$, with $\mathcal{J} = b^2$.

### Fixed points

From the computing the zeros of the beta functions above we find one line of RG fixed points at $g^\star = 0$, parametrized by $h$ (or $b$), with the BKT structure of the RG flow around the special point $g = h = 0$ (or $g = 0$ and $b = 1$), as shown in fig. 1(a).

Along the line of fixed points we find one marginal operator (i.e. $\chi$), and one operator with scaling dimension $\Delta_2 = 1 - 2h(\sqrt{2} - h) = b^2$. The latter is the dimension of the vertex operator (with $b_0 = 1$ replaced by $b$), which becomes relevant or irrelevant at $h > 0$ or $h < 0$, respectively. As explained before, the vertex operators changing from relevant to irrelevant corresponds to the transition from the disordered to ordered phase. Indeed, as found in [22], and reviewed in appendix B, the original LRI model corresponds to a line in the $\{g, h\}$ plane, parametrized by $\mathcal{J} = J/T$, where $J$ is the Ising coupling and $T$ is the temperature. The critical temperature corresponds to the point at which such a line intersects the RG trajectory that in the IR ends at $g = h = 0$, which therefore is the fixed point characterizing the universality class of the phase transition. However, the approach to such a fixed point is via a marginally irrelevant operator, because the operators $\mathcal{O}_g$ and $\mathcal{O}_h$ are marginal at the fixed point. Therefore, we expect to find logarithmic corrections to scaling in the IR. See Appendix E for more details on this.

---

[19]Alternatively, we could define beta functions for the renormalized couplings, by deriving them with respect to $L$, at fixed bare couplings. The two schemes are related by a coupling redefinition (e.g. [72]).

## 4.2 Spectrum at $s = 1$

At the fixed point $g = h = 0$, the theory at $s = 1$ consists of a compact GFF $\phi$ with $\Delta_\phi = 0$ (or equivalently, the Neumann boundary condition for the $d = 2$ free compact scalar $\Phi$, see Appendix A.2) and radius 1, together with an auxiliary $\mathbb{C}^2$ Hilbert space. As explained in Section 3.2.3, the LRI CFT is obtained by restricting to the $U(1)$-singlet sector.

In light of the list of possible $U(1)$-singlet operators in (3.35), we can write down the generating function for the full spectrum at $s = 1$

$$Z_1(x, y, q) = \text{tr}_{\mathcal{H}} \left( x^\alpha y^J q^D \right) = \frac{1 + x + q + xyq}{\prod\limits_{n=1}^{\infty} \left( 1 - x\, y^{n+1} q^n \right)} \, . \tag{4.14}$$

Here $D$ is the dilatation operator, $(-1)^\alpha$ is the generator of the global $\mathbb{Z}_2$, while $(-1)^J$ is the generator of parity. The construction of (4.14) is the following: the numerator provides the seeds for the parity even or odd operators with or without vertex operators, i.e. $\mathbb{1}$, $\hat{\sigma}_3$ and $V_+\hat{\sigma}_+ \pm V_-\hat{\sigma}_-$, while the denominator accounts for the possible words $A, B, C, D$ in (3.35), assigning them a factor $x$ for each power of $\phi$, a factor $y$ for each power of $\partial$ and $\phi$, and a factor $q$ for each power of $\partial$.

The generating function for primaries only is

$$Z_1^*(x, y, q) = 1 + (1 - yq)[Z_1(x, y, q) - 1] \, , \tag{4.15}$$

where, after having isolated the identity contribution, we subtract the operators that can be written as a total derivative.

We can use $Z_1^*$ to produce the following table of multiplicities of primaries, up to scaling dimension $\Delta = 10$:

| $\Delta$ | $d_{++}$ | $d_{+-}$ | $d_{-+}$ | $d_{--}$ |
|---|---|---|---|---|
| 0 | 1 | 0 | 1 | 0 |
| 1 | 2 | 0 | 1 | 0 |
| 2 | 1 | 0 | 1 | 0 |
| 3 | 2 | 0 | 1 | 1 |
| 4 | 2 | 1 | 3 | 0 |
| 5 | 4 | 0 | 2 | 2 |
| 6 | 3 | 3 | 5 | 1 |
| 7 | 7 | 1 | 3 | 5 |
| 8 | 5 | 6 | 9 | 2 |
| 9 | 13 | 2 | 6 | 9 |
| 10 | 8 | 12 | 16 | 4 |

$$\tag{4.16}$$

Here $d_{ab}$ stands for the dimension of the space of primaries with $\mathbb{Z}_2$ charge $a$ and parity charge $b$.

The first $\mathbb{Z}_2$-even, parity-odd primary appears at $\Delta = 4$, in agreement with what we saw at $s = 1/2$, in Section 2.2.2. This is reassuring since, as explained in Section 2.2.3, we expect a tower of protected $\mathbb{Z}_2$-even parity-odd primaries with $\Delta = 4, 6, \ldots$ to be present for all $s \in (1/2, 1)$.

There are two operators of $\Delta = 0$, namely the identity and $\hat{\sigma}_3$. The latter is $\mathbb{Z}_2$-odd and parity-even, so we can identify it with the $s \to 1$ limit of the operator $\sigma$. Moving on to $\Delta = 1$, we find the following operators

| operator | $(-1)^\alpha$ | $(-1)^J$ |
|---|---|---|
| $\partial\phi$ | $-1$ | $+1$ |
| $\partial\phi\hat{\sigma}_3$ | $+1$ | $+1$ |
| $V_+\hat{\sigma}_+ + V_-\hat{\sigma}_-$ | $+1$ | $+1$ |
| $V_+\hat{\sigma}_+ - V_-\hat{\sigma}_-$ | $-1$ | $-1$ . |

$$\tag{4.17}$$

We can identify the first line with $\chi \sim \varphi^3$ (in hindsight, this is why we defined $\chi = \frac{i}{\sqrt{2}}\partial\phi$). The second and third line are two linearly independent marginal operators which are uncharged under both $\mathbb{Z}_2$ and parity. Finally, the last line is needed to recombine with the trivial $\Delta = 0$ conformal multipliet of $\sigma$ to form a conformal multiplet with $\Delta_\sigma > 0$.

## 4.3 Beta functions and fixed point at $s < 1$

Next, we discuss beta functions for the model at $\delta > 0$, i.e. $s < 1$. While the coupling $g$ that multiplies the generalized vertex operators remains dimensionless, the coupling $h$ that multiplies $\chi$ has now (mass) dimension equal to $\delta/2$, and so we expect the beta function of $h$ to start linearly in $h$. More drastically, while the correlators of $\mathcal{O}_h$ remain conformal, $\mathcal{O}_g$ is no longer a scaling operator at $\delta > 0$ and its correlators are exponential functions. Nevertheless, the structure of perturbation theory remains similar to that of the $s = 1$ model. In particular, for the computation of beta functions, we will again restrict the interaction to a finite interval and renormalize one point functions of operators $\mathcal{O}_g$ and $\mathcal{O}_h$ inserted far away from it.

Generalizing (4.2), we define

$$\langle \mathcal{O}(\infty)\dots\rangle_{\mathcal{D}} \equiv \lim_{x\to\infty} \frac{\langle \mathcal{O}(x)\dots\rangle_{\mathcal{D}}}{\langle \mathcal{O}(x)\mathcal{O}(0)\rangle_0}. \tag{4.18}$$

Unlike for the conformal case (4.2), it is in general nontrivial to show that the above definition leads to a finite result. Fortunately, our UV theory is GFF and upon applying Wick's theorem, it is straightforward to see that the above limit is finite for any primary operator that is defined as a normal-ordered monomial in $\phi$ (with derivatives). The case of the vertex operators is less obvious, but if also all the other operators in the correlator are vertex operators, then the explicit formula (3.8), together with the fact that only an even number of alternating charges is allowed, leads to the same conclusion. The most intricate case is the one of mixed correlators, but again the finiteness of (4.18) can be proven with the help of Wick's theorem, in the same way as it is employed in Appendix D. Some examples of useful correlators with an operator inserted at infinity are provided in Appendix D.1.

As in the $\delta = 0$ case, we have, up to higher order perturbative corrections:

$$\langle \hat{\sigma}_3\chi(\infty)\rangle_{\mathcal{D}} = \frac{h}{2}\,\mathrm{tr}(\hat{\sigma}_3^2)\int_{-L/2}^{L/2} \mathrm{d}x\langle\chi(\infty)\chi(x)\rangle_0$$
$$+ \frac{g^2}{2}\int_{-L/2+a}^{L/2} \mathrm{d}x_1 \int_{-L/2}^{x_1-a}\mathrm{d}x_2\,\mathrm{tr}[\langle\hat{\sigma}_3\chi(\infty)\hat{\sigma}_+V_+(x_1)\hat{\sigma}_-V_-(x_2)\rangle_0 + (12)]$$
$$+ \frac{hg^2}{2}\int_{\mathrm{I}_3(a)}\mathrm{tr}\,[\langle\hat{\sigma}_3\chi(\infty)\hat{\sigma}_+V_+(x_1)\hat{\sigma}_-V_-(x_2)\hat{\sigma}_3\chi(x_3)\rangle_{0,c} + 5\ \mathrm{perm.}]$$
$$+ \frac{h^3}{2}\int_{\mathrm{I}_3(a)}\mathrm{tr}\,\langle\hat{\sigma}_3\chi(\infty)\hat{\sigma}_3\chi(x_1)\hat{\sigma}_3\chi(x_2)\hat{\sigma}_3\chi(x_3)\rangle_{0,c} + \dots,$$

$$\langle \hat{\sigma}_+V_+(\infty)\rangle_{\mathcal{D}} = \frac{g}{2}\,\mathrm{tr}(\hat{\sigma}_+\hat{\sigma}_-)\int_{-L/2}^{L/2} \mathrm{d}x\,\langle V_+(\infty)V_-(x)\rangle_0$$
$$+ \frac{gh}{2}\int_{-L/2+a}^{L/2} \mathrm{d}x_1 \int_{-L/2}^{x_1-a}\mathrm{d}x_2\,\mathrm{tr}\,[\langle\hat{\sigma}_+V_+(\infty)\hat{\sigma}_-V_-(x_1)\hat{\sigma}_3\chi(x_2)\rangle_0 + (12)] + \dots$$
$$+ \frac{gh^2}{2}\int_{\mathrm{I}_3(a)}\mathrm{tr}\,[\langle\hat{\sigma}_+V_+(\infty)\hat{\sigma}_-V_-(x_1)\hat{\sigma}_3\chi(x_2)\hat{\sigma}_3\chi(x_3)\rangle_{0,c} + 2\ \mathrm{perm.}]$$
$$+ \frac{g^3}{2}\int_{\mathrm{I}_3(a)}\mathrm{tr}\,\langle\hat{\sigma}_+V_+(\infty)\hat{\sigma}_-V_-(x_1)\hat{\sigma}_+V_+(x_2)\hat{\sigma}_-V_-(x_3)\rangle_{0,c} + \dots, \tag{4.19}$$

where the finite-$\delta$ correlation functions of $V_\pm$ and $\chi$ can be found in appendix D.

In practice, evaluating the divergent part of the integrals above for generic $\delta$ can be rather difficult, for those correlators whose exponential factors survive in the limit (4.18), as for the $\langle \chi VV \rangle$ or $\langle VVVV \rangle$ cases (see Appendix D.1). Luckily, as long as we are interested in the IR fixed point properties only at the leading nontrivial order in $\delta$, we can evaluate such integrals at $\delta = 0$ and thus recover (4.5) and (4.9). In order to justify the exchanging of the limit $\delta \to 0$ with the integration, it suffices to notice that at fixed $a$ and $L$, and for some finite $\delta^* > 0$, all the integrands are uniformly bounded functions in the integration range for $\delta \in [0, \delta^*]$; therefore, by the dominated convergence theorem we are allowed to exchange limit and integral.

Therefore, taking into account the dimensionality of $h$, we get

$$\beta_g = -2gh(\sqrt{2} - h) + g^3 + O(g^3 h, g^5)\,, \quad \beta_h = -\frac{\delta}{2}h - g^2(\sqrt{2} - 2h) + O(g^2 h^2, g^4)\,. \tag{4.20}$$

This is yet not the final answer for the *physical* RG. Indeed, as explained in Section 3.2.1, partition function and correlators only depend on the combination $b_0 - \sqrt{2}h$, hence the beta function entering the Callan-Symanzik equation is that for such a combination. In standard RG language, what this means is that while both $h$ and $b_0$ flow, one of them is a redundant coupling, only the combination $b_0 - \sqrt{2}h$ being an essential coupling. Therefore, we define the physical coupling in cutoff units as $b = (b_0 - \sqrt{2}h)a^{\delta/2}$, and obtain the beta functions

$$\beta_g = (b^2 - 1)g + g^3\,, \quad \beta_{b^2} = -\delta b^2 + 4b^2 g^2\,, \tag{4.21}$$

namely the same as (4.13) plus the classical contribution due to the dimension of $b^2$. We have thus shown how a perturbative computation in the model (3.9) fully reproduces the Kosterlitz beta functions (2.21), with $\mathcal{J} = b^2$, plus a next-to-leading correction of order $g^3$.

**Fixed points**

With the beta functions (4.21), we find a non-trivial fixed point at

$$g_* = \pm\sqrt{\frac{\delta}{4}} + O(\delta^{3/2})\,, \quad b_*^2 = 1 - \frac{\delta}{4} + O(\delta^2)\,. \tag{4.22}$$

We stress that although the fixed point of $b^2$ is of order one, the result is consistent with perturbation theory, because using that $b_*^2 = (b_0 - \sqrt{2}h_*)^2 a^\delta$, we find that

$$h_* = \frac{\delta}{\sqrt{2}}\left(\frac{1}{8} + \frac{1}{2}\log(\kappa a)\right) + O(\delta^2)\,, \tag{4.23}$$

so that both $g_*$ and $h_*$ are perturbative in $\delta$. The choice of sign for $g_*$ is irrelevant, as the full set of observables is invariant under $g \to -g$. However, intermediate calculations do depend on the sign of $g_*$, hence in the following we will choose $g_* = -\sqrt{\delta/4}$.

## 4.4 Anomalous dimensions

### 4.4.1 Leading near-marginal operators at the fixed point

By standard RG arguments, linearizing the beta functions around the IR fixed point we find the linear combination of $\mathcal{O}_h$ and $\mathcal{O}_g$ that behave as scaling operators at the fixed point, along with their IR scaling dimensions $\Delta_\pm = 1 + \omega_\pm$. The quantities $\omega_\pm$ are the eigenvalues of the stability matrix:

$$B_{ij} = \partial_i \beta_j|_{g_*, b_*^2} = \begin{pmatrix} -1 + b_*^2 + 3g_*^2 & 8b_*^2 g_* \\ g_* & 4g_*^2 - \delta \end{pmatrix}\,, \tag{4.24}$$

where indices $i, j$ run over the couplings $\{g, b^2\}$. Diagonalizing this matrix, we find $\omega_\pm = \pm\sqrt{2\delta} + \delta/4 + O(\delta^{3/2})$, and thus the IR scaling dimensions

$$\Delta_\pm = 1 \pm \sqrt{2\delta} + \delta/4 + O(\delta^{3/2}) \,. \tag{4.25}$$

The associated scaling operators $\mathcal{O}_\pm$ are the linear combinations

$$\mathcal{O}_\pm = a^{-\Delta_\pm} \sum_{i\in\{g,b^2\}} v_\pm^i \tilde{\mathcal{O}}_i \,, \tag{4.26}$$

where we introduced the notation $\tilde{\mathcal{O}}_i \equiv \mathcal{O}_i a^{\Delta_i^0}$, with $\Delta_i^0$ being the canonical (or engineering) dimension of $\mathcal{O}_i$, and where $v_\pm^i$ is the $i$-th component of a left-eigenvector of $B_{ij}$, corresponding to the eigenvalue $\omega_\pm$. We stress that the scaling operators are written directly in terms of bare operators, because the beta functions of bare couplings are associated to the Callan-Symanzik equations for bare correlators.

We find $v_\pm = (\pm\frac{1}{\sqrt{2}} + \frac{\sqrt{\delta}}{8} + O(\delta), -2)$, and thus:

$$a^{\Delta_\pm} \mathcal{O}_\pm = \frac{1}{\sqrt{2}}(\tilde{\mathcal{O}}_h \pm \tilde{\mathcal{O}}_g) + \frac{\sqrt{\delta}}{8}\tilde{\mathcal{O}}_g + O(\delta) \,, \tag{4.27}$$

where we used $\tilde{\mathcal{O}}_{b^2} = -\frac{1}{2\sqrt{2}}\tilde{\mathcal{O}}_h + O(\delta)$, that follows from the relation between $b$ and $h$.[20]

The signs in the definition of $\mathcal{O}_\pm$ depend on the sign of the fixed point $g_*$. If we had picked the other sign, $g_* = +\sqrt{\delta}/2$, we would have found the same expression up to the change $\mathcal{O}_g \to -\mathcal{O}_g$, which is obvious from the action. Notice that at leading order, this corresponds to exchanging $\mathcal{O}_+ \leftrightarrow \mathcal{O}_-$, but this property is broken by the term of order $\sqrt{\delta}$.

One might wonder how robust the $O(\delta)$ correction for $\Delta_\pm$ is, given the order at which we have computed the beta functions. The hypothetical higher-order corrections to the beta functions above are constrained as follows. First, both $\beta_g$ and $\beta_{b^2}$ are even functions of $b$. Second, due to the charge neutrality constraint, $\beta_g$ ($\beta_{b^2}$) is an odd (even) function of $g$. Third, the beta functions are computed as expansions in powers of $g$ and $h$, that can then be rearranged as series in $g$ and $b^2 - 1$. Lastly, we assume that the beta functions coefficients depend analytically on $\delta$. All in all, we have

$$\beta_g = \sum_{n=0}\sum_{m=0} a_{n,m}^{(1)}(\delta)g^{2n+1}(b^2-1)^m \,, \quad \beta_{b^2} = -\delta b^2 + \sum_{n=1}\sum_{m=0} a_{n,m}^{(2)}(\delta)g^{2n}(b^2-1)^m \,, \tag{4.28}$$

with

$$a_{n,m}^{(1)}(\delta) = \sum_{k=0} a_{n,m,k}^{(1)}\delta^k \,, \quad a_{n,m}^{(2)}(\delta) = \sum_{k=0} a_{n,m,k}^{(2)}\delta^k \,, \tag{4.29}$$

and, in order to match (4.21),

$$a_{0,0,0}^{(1)} = 0 \,, \quad a_{0,1,0}^{(1)} = a_{1,0,0}^{(1)} = 1 \,, \quad a_{1,0,0}^{(2)} = a_{1,1,0}^{(2)} = 4 \,. \tag{4.30}$$

Using this ansatz it is not difficult to verify that the corrections to (4.21) affect the fixed point location and critical exponents starting from $O(\delta^{3/2})$, and the eigenvectors from $O(\delta)$, hence the predictions (4.25) and (4.27) are robust.

---

[20] An integrated operator $\tilde{\mathcal{O}}_i$ is obtained by deriving the action with respect to the associated (dimensionless) coupling. Therefore, we have $\tilde{\mathcal{O}}_{b^2} = (\frac{\partial b^2}{\partial(ha^{\delta/2})})^{-1}\tilde{\mathcal{O}}_h$.

### 4.4.2 Anomalous dimensions of $\hat{\sigma}_3$ and $\chi$

As we discussed in Section 2, the spectrum of the 1d LRI CFT should contain two protected primary operators which are even under parity and odd under global $\mathbb{Z}_2$. In the near-crossover description, these two operators are $\hat{\sigma}_3$ and $\chi$, with protected dimensions

$$\Delta_\sigma = \frac{\delta}{2}\,, \quad \Delta_\chi = 1 - \frac{\delta}{2}\,. \tag{4.31}$$

For $\hat{\sigma}_3$, we have argued in Section 3.2.2 that its dimension is protected as a consequence of the Schwinger-Dyson equations; for $\chi$, its dimension must be $1 + \Delta_\phi$, and as usual in long-range models $\phi$ has no anomalous dimension. We now want to verify that these conclusions are corroborated by actual computations with our model.

The anomalous dimension of an operator $\mathcal{O}$ can be computed by treating $\mathcal{O}$ as a perturbation with coupling $\lambda$, and computing the beta function $\beta_\lambda$ for this new coupling. From there, the scaling dimension is obtained from $\partial_\lambda\beta_\lambda$ evaluated at the fixed point. Considering the expectation value of $\sigma \equiv a^{-\Delta_\sigma}\hat{\sigma}_3$, where the factor of $a$ ensures that the engineering and scaling dimension match, the first non-trivial contribution is at quadratic order in $g$ and $h$, for which we obtain:

$$
\begin{aligned}
a^{\Delta_\sigma}\langle\sigma(\infty)\rangle_\mathcal{D} = {}& \frac{\lambda}{2}\int \mathrm{d}y\, \mathrm{tr}\langle\hat{\sigma}_3(\infty)\hat{\sigma}_3(y)\rangle_0 \\
& + \frac{\lambda g^2}{2}\int \mathrm{d}y_1\,\mathrm{d}y_2\,\mathrm{d}y_3\,\mathrm{tr}\left[\langle\hat{\sigma}_3(\infty)\hat{\sigma}_3(y_1)\mathcal{O}_g(y_2)\mathcal{O}_g(y_3)\rangle_{0,c} + \text{ perms}\right] \\
& + \frac{\lambda h^2}{2}\int dy_1\,dy_2\,dy_3\,\mathrm{tr}\left[\langle\hat{\sigma}_3(\infty)\hat{\sigma}_3(y_1)\mathcal{O}_h(y_2)\mathcal{O}_h(y_3)\rangle_{0,c} + \text{ perms}\right] \\
= {}& \lambda I_0^\sigma + \lambda g^2 I_1^\sigma + \lambda h^2 I_2^\sigma\,,
\end{aligned}
\tag{4.32}
$$

with

$$I_0^\sigma = L\,, \quad I_1^\sigma = L\left(2 + \log 2 + 2\log\frac{a}{L} + \dots\right)\,, \quad I_2^\sigma = -L(2 + \log 2) + \dots\,, \tag{4.33}$$

where the dots represent higher-order corrections in $a$. Removing the logarithmic divergences requires the introduction of the renormalized coupling $\lambda_r$ via the relation

$$\lambda = \lambda_r - 2\lambda_r g_r^2 \log\frac{a}{L} + \dots\,, \tag{4.34}$$

from which we find the following beta function:

$$\beta_{\lambda_\sigma} = -\frac{d\lambda}{d\log a} = 2\lambda g^2 + \dots\,. \tag{4.35}$$

The corresponding anomalous dimension is

$$\gamma_\sigma = \partial_\lambda\beta_\lambda = 2g^2 \quad \to \quad \gamma_\sigma|_{g=g_*} = \frac{\delta}{2}\,. \tag{4.36}$$

That this dimension does not receive higher-order corrections in $\delta$ is guaranteed by the Schwinger-Dyson equation (3.30), which fixes $D_x\hat{\sigma}_3$ to be a descendant operator of protected scaling dimension $1 + \delta/2$ at the non-trivial IR fixed point.

Let us now turn to $\chi$, with tree-level dimension $1 - \delta/2$. Repeating the same calculations we find that all terms are zero except the linear term, which evaluates to $\lambda L$. Hence, $\chi$ does not get an anomalous dimension. It is not difficult to check that the same conclusion holds for higher orders in $\delta$.

All in all, we reproduce (4.31) and find that they are indeed protected and will not receive corrections at higher orders in $\delta$.

## 4.5 OPE coefficients

In this section, we compute the following correlation functions:

$$\langle \psi_i(x_1)\psi_j(x_2)\mathcal{O}_\pm(x_3)\rangle_\mathcal{D}\,, \quad \langle \mathcal{O}_\pm(x_1)\mathcal{O}_\pm(x_2)\mathcal{O}_\pm(x_3)\rangle_\mathcal{D}$$
$$\langle \mathcal{O}_+(x_1)\mathcal{O}_+(x_2)\mathcal{O}_-(x_3)\rangle_\mathcal{D}\,, \quad \langle \mathcal{O}_+(x_1)\mathcal{O}_-(x_2)\mathcal{O}_-(x_3)\rangle_\mathcal{D}\,, \tag{4.37}$$

where we denote $\psi_i = \{\sigma, \chi\}$ for brevity.

At the IR fixed point, where the LRI becomes a 1d CFT, we expect the form of these correlation functions to be consistent with conformal symmetry – see Section 2.2.1. In particular, assuming that all $\mathcal{O}_i = \{\psi_i, \mathcal{O}_\pm\}$ flow to (scalar) conformal primaries in the IR, we must have that

$$\langle \mathcal{O}_i(x_1)\rangle_\mathcal{D} = 0\,,$$
$$\langle \mathcal{O}_i(x_1)\mathcal{O}_j(x_2)\rangle_\mathcal{D} = \frac{\delta_{ij}\mathcal{N}_i\mathcal{N}_j}{x_{12}^{2\Delta_i}}\,,$$
$$\langle \mathcal{O}_i(x_1)\mathcal{O}_j(x_2)\mathcal{O}_k(x_3)\rangle_\mathcal{D} = \frac{c_{ijk}\mathcal{N}_i\mathcal{N}_j\mathcal{N}_k}{x_{12}^{\Delta_{ik}+\Delta_j}x_{13}^{\Delta_{ij}+\Delta_k}x_{23}^{\Delta_{ji}+\Delta_k}}\,, \tag{4.38}$$

where $c_{ijk}$ are the OPE coefficients, and $\mathcal{N}_i$ are the (scheme-dependent) normalization factors computed in appendix F.[21]

Our computation proceeds as follows: we compute three-point functions in a perturbative expansion in the couplings $g$ and $h$, and expand the correlators in $\delta$ before integrating them, as we did for the beta functions. Then, we tune the couplings to their fixed-point values, which are given as expansions in $\sqrt{\delta}$. The overall result is thus arranged as a power series in $\sqrt{\delta}$.

### 4.5.1 Tree level

At $O(\delta^0)$, there are no nonzero three-point functions involving only $\sigma, \chi$. The ones between $\sigma, \chi$ and $\mathcal{O}_\pm$, the leading marginal operators defined in (4.27), are nonzero:

$$\langle \sigma(x_1)\chi(x_2)\mathcal{O}_\pm(x_3)\rangle_\mathcal{D} = \frac{1/\sqrt{2}}{x_{23}^2}\,. \tag{4.39}$$

Thus the OPE coefficients with $\sigma, \chi$ at tree level are given by:

$$c_{\psi_i\psi_j\pm} = \frac{1-\delta_{ij}}{\sqrt{2}}\,, \quad c_{\psi_i\psi_j\psi_k} = 0\,. \tag{4.40}$$

The OPE coefficients involving only $\mathcal{O}_\pm$ are nonzero at tree level as well. Using the correlation functions between $\chi$ and $V_\pm$ given in appendix D, we immediately find

$$c_{\pm\pm\pm} = \frac{3}{2}\,, \quad c_{++-} = c_{--+} = -\frac{1}{2}\,. \tag{4.41}$$

### 4.5.2 Order $\sqrt{\delta}$

At order $\sqrt{\delta}$, it is sufficient to consider three-point functions perturbed by $g\,\mathcal{O}_g$. The OPE coefficients $c_{\psi_i\psi_j\psi_k}$ remain zero at this order, while $c_{\psi_i\psi_j\pm}$ get corrected. Starting with $c_{\sigma\sigma\pm}$ we find

$$\langle \sigma(x_1)\sigma(x_2)\mathcal{O}_\pm(x_3)\rangle_\mathcal{D} = \frac{g_*}{2}\int dy\, \mathrm{tr}\langle \mathrm{P}\hat{\sigma}_3(x_1)\hat{\sigma}_3(x_2)\frac{1}{\sqrt{2}}([\tilde{\mathcal{O}}_h]\pm[\tilde{\mathcal{O}}_g])(x_3)\mathcal{O}_g(y)\rangle_{0,c}\,, \tag{4.42}$$

---

[21]In order to identify the basis of conformal primaries at the IR fixed point, operators $\mathcal{O}_\pm$ must mix with the identity $\mathbb{1}$, so that (4.27) is replaced by (F.4), and $\chi$ must mix with $\sigma$, so that $\chi$ is replaced by $[\chi]$ in (F.12). In order to avoid notational overburden, in the following we omit the square braket on $\chi$. We refer the reader to appendix F for the detailed calculation.

where we emphasize that the correlator is evaluated at the fixed point $g_*$. The assumed ordering of the three external points is $\frac{L}{2} > x_1 > x_2 > x_3 > -\frac{L}{2}$, and we will keep this ordering throughout this section. The integrals can be straightforwardly computed and, taking the limit $L \to \infty$, give:

$$\frac{g_*}{2} \int dy \ \mathrm{tr}\langle \mathrm{P}\hat{\sigma}_3(x_1)\hat{\sigma}_3(x_2)\frac{1}{\sqrt{2}}([\tilde{\mathcal{O}}_h] \pm [\tilde{\mathcal{O}}_g])(x_3)\mathcal{O}_g(y)\rangle_{0,c} = \mp \frac{g_*\sqrt{2}x_{12}}{x_{13}x_{23}} \ , \tag{4.43}$$

which, comparing to the expressions of eq. (4.38) and using the normalization factors computed in appendix F gives:

$$c_{\sigma\sigma\pm} = \pm\frac{1}{\sqrt{2}}\sqrt{\delta} \ . \tag{4.44}$$

Repeating the computation for $\mathrm{tr}\langle \chi(x_1)\chi(x_2)\mathcal{O}_\pm(x_3)\rangle_\mathcal{D}$ gives the result $c_{\chi\chi\pm} = 0$.

The other OPE coefficients $c_{\sigma\chi\pm}, c_{\pm\pm\pm}, c_{\pm\pm\mp}$ can be computed in a similar way. Note however that they have a nonzero value already at tree level, and are therefore sensitive to the $O(\sqrt{\delta})$ corrections to $\mathcal{O}_\pm$, see (F.4). Taking into account this correction, we compute

$$\begin{aligned}
\langle \sigma(x_1)\chi(x_2)\mathcal{O}_\pm(x_3)\rangle_\mathcal{D} &= \mathrm{tr}\langle \sigma(x_1)\chi(x_2)\mathcal{O}_\pm(x_3)\rangle_0 \\
&+ \frac{g_*}{2}\int dy \ \mathrm{tr}\langle \mathrm{P}\sigma(x_1)\chi(x_2)\mathcal{O}_\pm(x_3)\mathcal{O}_g(y)\rangle_{0,c} + O(\delta) \\
&= a^{1-\Delta_\pm-\Delta_\sigma}\left[ I_0^{\sigma\chi\pm} + g_*I_1^{\sigma\chi\pm} + O(\delta)\right] \ ,
\end{aligned} \tag{4.45}$$

$$\begin{aligned}
\langle \mathcal{O}_\pm(x_1)\mathcal{O}_\pm(x_2)\mathcal{O}_\pm(x_3)\rangle_\mathcal{D} &= \mathrm{tr}\langle \mathcal{O}_\pm(x_1)\mathcal{O}_\pm(x_2)\mathcal{O}_\pm(x_3)\rangle_0 \\
&+ \frac{g_*}{2}\int dy \ \mathrm{tr}\langle \mathrm{P}\mathcal{O}_\pm(x_1)\mathcal{O}_\pm(x_2)\mathcal{O}_\pm(x_3)\mathcal{O}_g(y)\rangle_{0,c} + O(\delta) \\
&= a^{3-3\Delta_\pm}\left[ I_0^{\pm\pm\pm} + g_*I_1^{\pm\pm\pm} + O(\delta)\right] \ ,
\end{aligned} \tag{4.46}$$

$$\begin{aligned}
\langle \mathcal{O}_+(x_1)\mathcal{O}_+(x_2)\mathcal{O}_-(x_3)\rangle_\mathcal{D} &= \mathrm{tr}\langle \mathcal{O}_+(x_1)\mathcal{O}_+(x_2)\mathcal{O}_-(x_3)\rangle_0 \\
&+ \frac{g_*}{2}\int dy \ \mathrm{tr}\langle \mathrm{P}\mathcal{O}_+(x_1)\mathcal{O}_+(x_2)\mathcal{O}_-(x_3)\mathcal{O}_g(y)\rangle_{0,c} + O(\delta) \\
&= a^{3-2\Delta_+-\Delta_-}\left[ I_0^{++-} + g_*I_1^{++-} + O(\delta)\right] \ ,
\end{aligned} \tag{4.47}$$

$$\begin{aligned}
\langle \mathcal{O}_+(x_1)\mathcal{O}_-(x_2)\mathcal{O}_-(x_3)\rangle_\mathcal{D} &= \mathrm{tr}\langle \mathcal{O}_+(x_1)\mathcal{O}_-(x_2)\mathcal{O}_-(x_3)\rangle_0 \\
&+ \frac{g_*}{2}\int dy \ \mathrm{tr}\langle \mathrm{P}\mathcal{O}_+(x_1)\mathcal{O}_-(x_2)\mathcal{O}_-(x_3)\mathcal{O}_g(y)\rangle_{0,c} + O(\delta) \\
&= a^{3-2\Delta_--\Delta_+}\left[ I_0^{+--} + g_*I_1^{+--} + O(\delta)\right] \ ,
\end{aligned} \tag{4.48}$$

where

$$I_0^{\sigma\chi\pm} = \frac{\sqrt{2}}{2x_{23}^2} \ , \qquad I_1^{\sigma\chi\pm} = \pm\frac{2\log\left(\frac{x_{13}x_{23}}{a\,x_{12}}\right)}{x_{23}^2} \ ,$$

$$I_0^{\pm\pm\pm} = \frac{3\left(4 \pm \sqrt{2\delta}\right)}{8x_{12}x_{13}x_{23}} \pm \frac{\sqrt{\delta}}{a\sqrt{2}}\left(\frac{1}{x_{12}^2} + \frac{1}{x_{13}^2} + \frac{1}{x_{23}^2}\right) \ ,$$

$$I_1^{\pm\pm\pm} = \pm\frac{\sqrt{2}}{a}\left(\frac{1}{x_{12}^2} + \frac{1}{x_{13}^2} + \frac{1}{x_{23}^2}\right) \pm \frac{12\log\left(\frac{x_{12}x_{13}x_{23}}{a^3}\right) - 9}{2\sqrt{2}x_{12}x_{13}x_{23}} \ , \tag{4.49}$$

$$I_0^{++-} = -\frac{\sqrt{\delta}}{a\sqrt{2}x_{12}^2} + \frac{\sqrt{2\delta} - 4}{8x_{12}x_{13}x_{23}} \ , \qquad I_1^{++-} = -\frac{\sqrt{2}}{ax_{12}^2} + \frac{4\log\left(\frac{x_{13}x_{23}a}{x_{12}^3}\right) + 1}{2\sqrt{2}x_{12}x_{13}x_{23}} \ ,$$

$$I_0^{+--} = \frac{\sqrt{\delta}}{a\sqrt{2}x_{23}^2} - \frac{\sqrt{2\delta} + 4}{8x_{12}x_{13}x_{23}} \ , \qquad I_1^{+--} = \frac{\sqrt{2}}{ax_{23}^2} + \frac{4\log\left(\frac{x_{23}^3}{x_{12}x_{13}a}\right) - 1}{2\sqrt{2}x_{12}x_{13}x_{23}} \ .$$

This results in the following values for the OPE coefficients at the fixed point at $O(\sqrt{\delta})$:

$$c_{\sigma\chi\pm} = \frac{1}{\sqrt{2}} \mp \frac{\sqrt{\delta}}{16} \,, \quad c_{\pm\pm\pm} = \frac{3}{2} \pm \frac{39\sqrt{\delta}}{16\sqrt{2}} \,, \quad c_{++-} = -\frac{1}{2} + \frac{\sqrt{\delta}}{16\sqrt{2}} \,, \quad c_{+--} = -\frac{1}{2} - \frac{\sqrt{\delta}}{16\sqrt{2}} \,. \tag{4.50}$$

### 4.5.3   Order $\delta$

While computing the order $O(\delta)$ correction to $c_{\sigma\chi\pm}, c_{\pm\pm\pm}, c_{\pm\pm\mp}$ requires including the $O(\delta)$ corrections to $\mathcal{O}_\pm$, in eq. (F.19), we can compute the OPE coefficients $c_{\sigma\sigma\pm}$ to order $O(\delta)$ straight away.

The three-point function is given by

$$\begin{aligned}
\langle\sigma(x_1)\sigma(x_2)\mathcal{O}_\pm(x_3)\rangle_\mathcal{D} &= \mathrm{tr}\langle\sigma(x_1)\sigma(x_2)\mathcal{O}_\pm(x_3)\rangle_0 + \frac{g_*}{2}\int dy \, \mathrm{tr}\langle\mathrm{P}\sigma(x_1)\sigma(x_2)\mathcal{O}_\pm(x_3)\mathcal{O}_g(y)\rangle_{0,c} \\
&\quad + \frac{g_*^2}{4}\int dy_1\,dy_2 \, \mathrm{tr}\langle\mathrm{P}\sigma(x_1)\sigma(x_2)\mathcal{O}_\pm(x_3)\mathcal{O}_g(y_1)\mathcal{O}_g(y_2)\rangle_{0,c} \\
&\quad + \frac{h_*}{2}\int dy_1 \, \mathrm{tr}\langle\mathrm{P}\sigma(x_1)\sigma(x_2)\mathcal{O}_\pm(x_3)\mathcal{O}_h(y)\rangle_{0,c} + O(\delta^{3/2}) \\
&= a^{1-2\Delta_\sigma-\Delta_\pm}\left[I_0^{\sigma\sigma\pm} + g_* I_1^{\sigma\sigma\pm} + g_*^2 I_2^{\sigma\sigma\pm} + h_* I_3^{\sigma\sigma\pm} + O(\delta^{3/2})\right] \\
&= \frac{c_{\sigma\sigma\pm}\mathcal{N}_\sigma^2\mathcal{N}_\pm}{(x_{12})^{2\Delta_\sigma-\Delta_\pm}(x_{13})^{\Delta_\pm}(x_{23})^{\Delta_\pm}} \,,
\end{aligned} \tag{4.51}$$

where we have explicitly included the normalizations $\mathcal{N}_\sigma, \mathcal{N}_\pm$ that differ from 1 at this order. The expressions are given in (F.9) and (F.18). The perturbation $\mathcal{O}_h$ contributes as well. Evaluating the integrals, we find

$$\begin{aligned}
I_0^{\sigma\sigma\pm} &= \frac{1}{a}\left(\pm\sqrt{\frac{\delta}{2}} - \frac{1}{2}\delta\log(8a\kappa)\right) \,, \quad I_1^{\sigma\sigma\pm} = \pm\frac{\sqrt{2}}{a} \mp \frac{\sqrt{2}x_{12}}{x_{13}x_{23}} + \sqrt{\delta}\left(\frac{1}{4a} - \frac{x_{12}}{4x_{13}x_{23}}\right) \,, \\
I_2^{\sigma\sigma\pm} &= \frac{\log 64}{a} \mp \frac{4x_{12}(\log\frac{ax_{12}}{x_{13}x_{23}} - 1)}{x_{13}x_{23}} \,, \quad I_3^{\sigma\sigma\pm} = \frac{\sqrt{2}}{a} \,.
\end{aligned} \tag{4.52}$$

Adding all contributions and evaluating them at the fixed point, we obtain

$$c_{\sigma\sigma\pm} = \pm\frac{\sqrt{\delta}}{\sqrt{2}} - \frac{15}{16}\delta + O(\delta^{3/2}) \,. \tag{4.53}$$

What about $c_{\chi\chi\pm}$? We would not need to worry about higher-order corrections to $\mathcal{O}_\pm$, however this would require us to use the three-point function $\langle\chi(x_1)\chi(x_2)\chi(x_3)\rangle$ at arbitrary $s$. An easier way to obtain these OPE coefficients is by exploiting the OPE relations mentioned in Section 2.2 and given in (5.116), which relate $c_{\chi\chi\pm}$ to $c_{\sigma\sigma\pm}$ and $c_{\sigma\chi\pm}$. The result is given in (5.120).

## 5   Analytic conformal bootstrap analysis

In this section, we study the 1d LRI CFT using analytic conformal bootstrap and determine CFT data of light primary operators perturbatively in $\sqrt{\delta}$. The bootstrap results are in perfect agreement with the RG calculations of the previous section and extend them to higher orders and to other observables. This agreement provides strong evidence for the conformal invariance of the IR fixed point, as well as for validity of the proposed field-theoretic description.

| operator | form at $\delta = 0$ | $\Delta^{(0)}$ | $\mathbb{Z}_2$ | parity |
|----------|:-------------------:|:--------------:|:--------------:|:------:|
| identity | $\mathbb{1}$ | 0 | $+1$ | $+1$ |
| $\sigma$ | $\hat{\sigma}_3$ | 0 | $-1$ | $+1$ |
| $\mathcal{O}_g$ | $\hat{\sigma}_+ V_+ + \hat{\sigma}_- V_-$ | 1 | $+1$ | $+1$ |
| $\mathcal{O}_h$ | $i\hat{\sigma}_3 \partial\phi/\sqrt{2}$ | 1 | $+1$ | $+1$ |
| $\chi$ | $i\partial\phi/\sqrt{2}$ | 1 | $-1$ | $+1$ |
| $\rho$ | $-:(\partial\phi)^2:/(2\sqrt{2})$ | 2 | $+1$ | $+1$ |
| $\widetilde{\rho}$ | $\hat{\sigma}_3 :(\partial\phi)^2:/(2\sqrt{2})$ | 2 | $-1$ | $+1$ |

Table 1: Primary operators with $\Delta^{(0)} = 0, 1, 2$.

## 5.1 Setting up the problem

Our bootstrap analysis rests on the following assumptions:

1. The critical 1d LRI is described by a family of 1d reflection-positive CFTs, as defined in Section 2.2.1, parametrized by $\delta \in [0, 1/2)$, possessing a global $\mathbb{Z}_2$ and parity symmetry;

2. As $\delta \to 0^+$, the CFT data tend continuously to those of the exact solution at $s = 1$, identified in Section 4.2, and denoted by $\Delta_i^{(0)}$ and $c_{ijk}^{(0)}$;

3. The CFT data admit an asymptotic expansion in nonnegative powers of $\sqrt{\delta}$ as $\delta \to 0^+$, i.e.

$$\Delta_i(\delta) \sim \sum_{n=0}^{\infty} \Delta_i^{(n)} \delta^{\frac{n}{2}}, \qquad c_{ijk}(\delta) \sim \sum_{n=0}^{\infty} c_{ijk}^{(n)} \delta^{\frac{n}{2}}; \tag{5.1}$$

4. The CFT contains $\mathbb{Z}_2$-odd parity-even primaries $\sigma$, $\chi$ of exact scaling dimensions $\Delta_\sigma = \delta/2$ and $\Delta_\chi = 1 - \delta/2$.

Eventually, we will also make an additional small assumption regarding the behaviour of the CFT data under $\sqrt{\delta} \to -\sqrt{\delta}$. In the rest of this section, we will use assumptions 1 and 4 to constrain $\Delta_i^{(n)}$, $c_{ijk}^{(n)}$ for $n > 0$.

Let us start by summarizing the $\delta = 0$ CFT data of primary operators with $\Delta^{(0)} \leq 2$, shown in Table 1.[22] Since we normalize the two-point function of $\phi$ as $\langle \phi(x)\phi(y) \rangle = -2\log|x - y|$, all of the operators in the table have unit-normalized two-point functions.

The space of $\mathbb{Z}_2$-even primaries with $\Delta^{(0)} = 1$ is two-dimensional, spanned by $\mathcal{O}_g, \mathcal{O}_h$. For $\delta > 0$, we expect that the degeneracy is lifted, giving rise to a pair of primaries $\mathcal{O}_+, \mathcal{O}_-$. As $\delta \to 0^+$, we must have

$$\begin{pmatrix} \mathcal{O}_+ \\ \mathcal{O}_- \end{pmatrix} \to A \begin{pmatrix} \mathcal{O}_g \\ \mathcal{O}_h \end{pmatrix}, \tag{5.2}$$

where $A \in \mathrm{GL}_2(\mathbb{R})$. Since both sets $\{\mathcal{O}_+, \mathcal{O}_-\}, \{\mathcal{O}_g, \mathcal{O}_h\}$ are orthonormal, we have in fact $A \in \mathrm{O}(2)$. By possibly multiplying $\mathcal{O}_+$ by $-1$, we can arrange $A \in \mathrm{SO}(2)$. It follows that at $\delta = 0^+$ we must have

$$\begin{aligned} \mathcal{O}_+ &= \cos\theta\, \mathcal{O}_g + \sin\theta\, \mathcal{O}_h, \\ \mathcal{O}_- &= -\sin\theta\, \mathcal{O}_g + \cos\theta\, \mathcal{O}_h, \end{aligned} \tag{5.3}$$

---

[22] Note that all of the listed primaries are parity-even. Strictly at $\delta = 0$, there is also a $\Delta = 1$, $\mathbb{Z}_2$-odd, parity-odd primary $\hat{\sigma}_+ V_+ - \hat{\sigma}_- V_-$, but the latter becomes the first descendant of $\sigma$ at $\delta = 0^+$, as we know from Section 3.2.2, and as we recover below from the bootstrap perspective.

where $\theta \in [0, 2\pi)$. By possibly multiplying both $\mathcal{O}_+$ and $\mathcal{O}_-$ by $-1$, and switching $\mathcal{O}_+ \leftrightarrow \mathcal{O}_-$, we can arrange $\theta \in [0, \pi/2)$. We will see below that the conformal bootstrap fixes $\theta$ uniquely.

Next, let us discuss the OPE of light primaries at $\delta = 0^+$. We have $\sigma(x)\sigma(y) = \hat{\sigma}_3^2 = \mathbb{1}$, and so $c_{\sigma\sigma\mathcal{P}}^{(0)} = 0$ for all primaries $\mathcal{P} \neq \mathbb{1}$. Next, we have

$$\sigma(x)\chi(y) = i\hat{\sigma}_3\partial\phi(y)/\sqrt{2} = \mathcal{O}_h(y) = \cos\theta\,\mathcal{O}_-(y) + \sin\theta\,\mathcal{O}_+(y)\,. \tag{5.4}$$

It follows that

$$c_{\sigma\chi+}^{(0)} = \sin\theta\,, \quad c_{\sigma\chi-}^{(0)} = \cos\theta\,, \tag{5.5}$$

and $c_{\sigma\chi\mathcal{P}}^{(0)} = 0$ for all other primaries. Let us consider the $\delta = 0^+$ OPEs

$$\sigma(x)\mathcal{O}_+(y) = \sin\theta\,\hat{\sigma}_3\mathcal{O}_h(y) + \cos\theta\,\hat{\sigma}_3\mathcal{O}_g(y) = \sin\theta\,\chi(y) + \cos\theta\,[\hat{\sigma}_+V_+(y) - \hat{\sigma}_-V_-(y)]\,, \tag{5.6}$$
$$\sigma(x)\mathcal{O}_-(y) = \cos\theta\,\hat{\sigma}_3\mathcal{O}_h(y) - \sin\theta\,\hat{\sigma}_3\mathcal{O}_g(y) = \cos\theta\,\chi(y) - \sin\theta\,[\hat{\sigma}_+V_+(y) - \hat{\sigma}_-V_-(y)]\,.$$

We see that the OPEs contain the primary $\chi$, with the expected coefficients $c_{\sigma+\chi}^{(0)} = c_{\sigma\chi+}^{(0)} = \sin\theta$ and $c_{\sigma-\chi}^{(0)} = c_{\sigma\chi-}^{(0)} = \cos\theta$. However, there is also the operator in the square bracket, proportional to $\partial\sigma$ for $\delta > 0$. To understand its appearance, consider the OPE of general parity-even primaries

$$\mathcal{P}_i(x_1)\mathcal{P}_j(x_2) = \sum_k c_{ijk}(-1)^{J_k}|x_{12}|^{-\Delta_i-\Delta_j+\Delta_k}\left[\mathcal{P}_k(x_2) + \frac{\Delta_k+\Delta_i-\Delta_j}{2\Delta_k}x_{12}\partial\mathcal{P}_k(x_2) + \ldots\right]\,. \tag{5.7}$$

Let us substitute $\mathcal{P}_i = \sigma$, $\mathcal{P}_j = \mathcal{O}_\pm$ and focus on the contribution of $\mathcal{P}_k = \sigma$. Since $\Delta_\sigma \to 0$ as $\delta \to 0$, the pole $1/\Delta_k$ in the coefficient of $\partial\sigma$ can cancel with a zero of $c_{\sigma\pm\sigma}$ to yield a nonvanishing contribution of the first descendant in the limit, although the primary is not present. This is precisely what happens in (5.6), as we will verify in the next subsection.

In the following, we will also need the $\delta = 0$ OPE coefficients of the possible triples of $\mathcal{O}_\pm$, namely $c_{+++}^{(0)}$, $c_{++-}^{(0)}$, $c_{+--}^{(0)}$, and $c_{---}^{(0)}$. The only nonvanishing three-point function between $\mathcal{O}_g$ and $\mathcal{O}_h$ is

$$\langle\mathcal{O}_g(x_1)\mathcal{O}_g(x_2)\mathcal{O}_h(x_3)\rangle = \frac{\sqrt{2}}{|x_{12}x_{13}x_{23}|}\,. \tag{5.8}$$

It follows that

$$c_{+++}^{(0)} = 3\sqrt{2}(\sin\theta)(\cos\theta)^2\,, \quad c_{---}^{(0)} = 3\sqrt{2}(\sin\theta)^2(\cos\theta)\,, \tag{5.9}$$
$$c_{++-}^{(0)} = \sqrt{2}[(\cos\theta)^3 - 2(\sin\theta)^2(\cos\theta)]\,, \quad c_{+--}^{(0)} = \sqrt{2}[(\sin\theta)^3 - 2(\sin\theta)(\cos\theta)^2]\,.$$

## 5.2 The crossing equations of light primaries up to $O(\sqrt{\delta})$

We begin the task of constraining the CFT data by considering the crossing equations of the $\langle\sigma\sigma\mathcal{O}_a\mathcal{O}_b\rangle$ correlators (here and in the following $a, b = \pm$). The crossing equation (2.10) takes the form[23]

$$\sum_{m=0}^{\infty} c_{\sigma\sigma m}c_{abm}G_{\Delta_m}^{\Delta_\sigma,\Delta_\sigma,\Delta_a,\Delta_b}(z) = \sum_{m=0}^{\infty} c_{\sigma am}c_{\sigma bm}G_{\Delta_m}^{\Delta_\sigma,\Delta_b,\Delta_a,\Delta_\sigma}(1-z)\,, \tag{5.10}$$

where the sum on both sides runs over all primary operators. Due to the factor $c_{\sigma\sigma m}$, the LHS only receives nonzero contributions from $\mathbb{Z}_2$-even, parity-even primaries, while the RHS from $\mathbb{Z}_2$-odd primaries of either parity. Let us expand both sides around $\delta = 0$. At $O(\delta^0)$, the LHS only contains the identity operator and

---

[23]By a slight abuse of notation, the lower indices labeling CFT data $\Delta_i$, $c_{ijk}$ will take both numerical values $i \in \mathbb{Z}_{\geq 0}$ (when we sum over all primaries), as well as specific values $i = \sigma, \chi, +, -, \ldots$ (when we refer to a specific primary).

equals $z^{-2}\delta_{ab}$. At the same time, the RHS only contains $\sigma$ and $\chi$ at $O(\delta^0)$. Indeed, $\chi$ contributes because $c^{(0)}_{\sigma a\chi}c^{(0)}_{\sigma b\chi} \neq 0$. On the other hand, it may seem that $\sigma$ should not contribute since $c^{(0)}_{\sigma a\sigma}c^{(0)}_{\sigma b\sigma} = 0$. However, the conformal block $G^{\Delta_\sigma,\Delta_b,\Delta_a,\Delta_\sigma}_{\Delta_\sigma}$ goes like $\delta^{-1}$ as $\delta \to 0$. In agreement with the discussion after (5.7), the singular contribution comes from the first descendant of $\sigma$. More concretely, the $\delta \to 0$ expansion of the t-channel conformal blocks takes the form

$$
\begin{aligned}
G^{\Delta_\sigma,\Delta_b,\Delta_a,\Delta_\sigma}_{\Delta_\sigma}(1-z) &= z^{-2}\delta^{-1} + z^{-2}\left[(\Delta^{(1)}_a + \Delta^{(1)}_b)(1-\log z) - \Delta^{(1)}_a \log(1-z)\right]\delta^{-\frac{1}{2}} + O(\delta^0) \\
G^{\Delta_\sigma,\Delta_b,\Delta_a,\Delta_\sigma}_{\Delta_\chi}(1-z) &= z^{-2} + z^{-2}\left[-(\Delta^{(1)}_a + \Delta^{(1)}_b)\log z - \Delta^{(1)}_a \log(1-z)\right]\delta^{\frac{1}{2}} + O(\delta).
\end{aligned}
\tag{5.11}
$$

Equality of the s-channel and t-channel at $O(\delta^0)$ is thus equivalent to

$$
\delta_{ab} = c^{(1)}_{\sigma\sigma a}c^{(1)}_{\sigma\sigma b} + c^{(0)}_{\sigma\chi a}c^{(0)}_{\sigma\chi b}.
\tag{5.12}
$$

In other words $(c^{(1)}_{\sigma\sigma+})^2 = (\cos\theta)^2$, $(c^{(1)}_{\sigma\sigma-})^2 = (\sin\theta)^2$, $c^{(1)}_{\sigma\sigma+}c^{(1)}_{\sigma\sigma-} = -\sin\theta\cos\theta$. There are two solutions

$$
c^{(1)}_{\sigma\sigma+} = s\cos\theta, \quad c^{(1)}_{\sigma\sigma-} = -s\sin\theta,
\tag{5.13}
$$

where $s = \pm 1$. The reason for the existence of two solutions is an additional $\mathbb{Z}_2$ symmetry of the $\delta = 0$ theory. This symmetry acts on all operators by conjugation by $\hat{\sigma}_3$. It leaves $\sigma$, $\chi$ and $\mathcal{O}_h$ invariant, sends $\mathcal{O}_g \mapsto -\mathcal{O}_g$, and thus exchanges $\mathcal{O}_- \leftrightarrow \mathcal{O}_+$. It is not a symmetry of the $\delta > 0$ theory. Instead, it exchanges pairs of equivalent solutions of the bootstrap. It is equivalent to the $2\pi$ monodromy around $\delta = 0$, i.e. the mapping $\sqrt{\delta} \mapsto -\sqrt{\delta}$. Without loss of generality, we will restrict to one branch of solutions by setting $s = 1$.

Let us expand (5.10) to $O(\sqrt{\delta})$. On the RHS, the only contributions can arise from corrections to the exchange of $\sigma$ and $\chi$ since $c_{\sigma am} = O(\sqrt{\delta})$ for $m \neq \sigma,\chi$. On the LHS, the contributions arise from the $O(\sqrt{\delta})$ correction to the identity conformal block, as well as from the exchange of $\mathcal{O}_-$ and $\mathcal{O}_+$, which appear for the first time at this order. In principle, there could also be exchanges of other primaries $\mathcal{P}$ for which $c^{(1)}_{\sigma\sigma\mathcal{P}} \neq 0$ and $c^{(0)}_{ab\mathcal{P}} \neq 0$. By using (5.9) and (5.11), we find that the crossing equations for $a,b = \pm$ are then equivalent to the finite number of constraints

$$
\theta = \frac{\pi}{4}, \quad \Delta^{(1)}_+ = \sqrt{2}, \quad \Delta^{(1)}_- = -\sqrt{2}, \quad c^{(1)}_{\sigma\chi+} = -c^{(1)}_{\sigma\chi-}, \quad c^{(2)}_{\sigma\sigma+} = c^{(2)}_{\sigma\sigma-} = c^{(1)}_{\sigma\chi-} - 1,
\tag{5.14}
$$

as well as the statement that $c^{(1)}_{\sigma\sigma\mathcal{P}} = 0$ unless $\mathcal{P} = \mathcal{O}_\pm$. We can now also update (5.5), (5.9), and (5.13) using the correct value of $\theta$

$$
c^{(0)}_{\sigma\chi+} = c^{(0)}_{\sigma\chi-} = \frac{1}{\sqrt{2}}, \quad c^{(1)}_{\sigma\sigma+} = -c^{(1)}_{\sigma\sigma-} = \frac{1}{\sqrt{2}}, \quad c^{(0)}_{+++} = c^{(0)}_{---} = \frac{3}{2}, \quad c^{(0)}_{++-} = c^{(0)}_{+--} = -\frac{1}{2}.
\tag{5.15}
$$

Note also that $c^{(0)}_{\chi\chi+} = c^{(0)}_{\chi\chi-} = 0$. Indeed, at $\delta = 0$, $\chi$ is a generalized free field with $\Delta_\chi = 1$. The $\chi \times \chi$ OPE contains only $\mathbb{1}$ and double trace operators with $\Delta = 2, 4, \ldots$.

### 5.2.1 Other crossing equations

Let us analyze other four-point functions of $\sigma$, $\chi$, $\mathcal{O}_+$, $\mathcal{O}_-$. We will study the complete set of such correlators for which $\sigma$ appears at least once among the external operators. Consider a four-point function $\mathcal{G}_{ijk\ell}(z)$ where $\sigma$ appears $N$ times among the external operators, where $N = 1, 2, 3, 4$. It turns out that the perturbative expansion of $\mathcal{G}_{ijk\ell}(z)$ up to and including $O(\delta^{\frac{N-1}{2}})$ involves only a finite number of conformal blocks in both the s- and the t-channel. This is because at $\delta = 0$, the $\sigma \times \mathcal{P}$ OPE contains a finite number of primaries, for any primary $\mathcal{P}$. Indeed, we saw in the previous subsection that $\mathcal{G}_{\sigma\sigma\pm\pm}$ (for which $N = 2$) involves only exchanges of $\mathbb{1}$ and $\mathcal{O}_\pm$ in the s-channel and only $\sigma$ and $\chi$ in the t-channel, up to $O(\sqrt{\delta})$.

Independent crossing equations in 1d CFTs are labeled by cyclic orderings of quadruples of primaries $(i, j, k, \ell)$. Indeed, the crossing equation (2.10) for $\mathcal{G}_{ijk\ell}$ is invariant under the cyclic shift $(i, j, k, \ell) \to (j, k, \ell, i)$. This follows immediately from identities satisfied by conformal blocks

$$G_{\Delta_m}^{\Delta_j, \Delta_i, \Delta_\ell, \Delta_k}(z) = (1-z)^{-\Delta_i + \Delta_j + \Delta_k - \Delta_\ell} \, G_{\Delta_m}^{\Delta_i, \Delta_j, \Delta_k, \Delta_\ell}(z) \,,$$
$$G_{\Delta_m}^{\Delta_j, \Delta_k, \Delta_\ell, \Delta_i}(z) = z^{-\Delta_i + \Delta_j + \Delta_k - \Delta_\ell} \, G_{\Delta_m}^{\Delta_i, \Delta_\ell, \Delta_k, \Delta_j}(z) \,. \tag{5.16}$$

If the 1d CFT is also invariant under parity, the crossing equation for $\mathcal{G}_{ijk\ell}$ is equivalent to that for $\mathcal{G}_{\ell k j i}$.

It follows that the full set of crossing equations with at least one external $\sigma$ and only $\sigma$, $\chi$, $\mathcal{O}_+$, $\mathcal{O}_-$ as external states arises by equating the s- and t-channel in the correlators

$$\begin{aligned} N = 4: \quad & \langle \sigma\sigma\sigma\sigma \rangle \\ N = 3: \quad & \langle \sigma\sigma\sigma\chi \rangle \\ N = 2: \quad & \langle \sigma\sigma\chi\chi \rangle, \ \langle \sigma\chi\sigma\chi \rangle, \ \langle \sigma\sigma\mathcal{O}_a\mathcal{O}_b \rangle, \ \langle \sigma\mathcal{O}_a\sigma\mathcal{O}_b \rangle \\ N = 1: \quad & \langle \sigma\chi\chi\chi \rangle, \ \langle \sigma\chi\mathcal{O}_a\mathcal{O}_b \rangle, \ \langle \sigma\mathcal{O}_a\chi\mathcal{O}_b \rangle \,. \end{aligned} \tag{5.17}$$

It turns out that if we work to $O(\delta^{\frac{N-1}{2}})$, then the crossing equations for each of these correlators are automatically satisfied by the CFT data (5.14), (5.15). Explicit formulas for the correlators are summarized later in Section 5.7.

## 5.3 Higher-order analysis

In order to bootstrap the CFT data to higher orders in $\sqrt{\delta}$, we have to contend with the fact that infinitely many primary operators appear in the OPE. For a four-point function with $N = 1, 2, 3, 4$ external $\sigma$ insertions, we will consider the crossing equation at $O(\delta^{\frac{N}{2}})$. This is the first order at which an infinite sum over conformal blocks occurs. Let us write the asymptotic expansion of a four-point function $\mathcal{G}_{ijk\ell}(z)$ as follows:

$$\mathcal{G}_{ijk\ell}(z) \sim \sum_{n=0}^{\infty} \mathcal{G}_{ijk\ell}^{(n)}(z) \delta^{\frac{n}{2}} \,. \tag{5.18}$$

Consider the s- and t-channel OPE of $\mathcal{G}_{ijk\ell}^{(N)}(z)$. The only operators with $\Delta^{(0)} = 0, 1$ are $\mathbb{1}$, $\sigma$, $\chi$, $\mathcal{O}_+$, and $\mathcal{O}_-$. Correspondingly, we separate the contribution of these operators to the two OPEs as follows

$$\mathcal{G}_{ijk\ell}^{(N)}(z) = \mathcal{G}_{ijk\ell}^{s,L}(z) + \mathcal{G}_{ijk\ell}^{s,H}(z) = \mathcal{G}_{ijk\ell}^{t,L}(z) + \mathcal{G}_{ijk\ell}^{t,H}(z) \,. \tag{5.19}$$

Here $\mathcal{G}_{ijk\ell}^{s,L}(z)$ and $\mathcal{G}_{ijk\ell}^{t,L}(z)$ denote the total contribution of the "light" operators $\mathbb{1}$, $\sigma$, $\chi$, $\mathcal{O}_+$, and $\mathcal{O}_-$ to the s-channel and t-channel conformal block expansion. Similarly, $\mathcal{G}_{ijk\ell}^{s,H}(z)$ and $\mathcal{G}_{ijk\ell}^{t,H}(z)$ denote the total contribution of the "heavy" operators, i.e. those with $\Delta^{(0)} \geq 2$. Since primary operators with $\Delta^{(0)} \geq 2$ appear for the first time at this order, $\mathcal{G}_{ijk\ell}^{s,H}(z)$ and $\mathcal{G}_{ijk\ell}^{t,H}(z)$ admit an expansion in conformal blocks all of whose scaling dimensions are evaluated at $\delta = 0$

$$\mathcal{G}_{ijk\ell}^{s,H}(z) = \sum_{n=2}^{\infty} A_n^{ijk\ell} \, G_n^{\Delta_i^{(0)}, \Delta_j^{(0)}, \Delta_k^{(0)}, \Delta_\ell^{(0)}}(z) \,,$$
$$\mathcal{G}_{ijk\ell}^{t,H}(z) = \sum_{n=2}^{\infty} A_n^{jk\ell i} \, G_n^{\Delta_i^{(0)}, \Delta_\ell^{(0)}, \Delta_k^{(0)}, \Delta_j^{(0)}}(1-z) \,. \tag{5.20}$$

Here $A_n^{ijk\ell}$ is the coefficient of the leading order in $\delta$ in the sum of the expression $(-1)^{J_m} c_{ijm} c_{k\ell m}$ over all primaries $\mathcal{P}_m$ such that $\Delta_m^{(0)} = n$:

$$A_n^{ijk\ell} = \sum_{m: \Delta_m^{(0)} = n} (-1)^{J_m} c_{ijm} c_{k\ell m} \big|_{\delta^{\frac{N}{2}}} \,. \tag{5.21}$$

### 5.3.1 Analytic functionals

The crossing equation (5.19) allows us to solve for $A_n^{ijk\ell}$ and $A_n^{jk\ell i}$ in terms of the CFT data appearing in $\mathcal{G}_{ijk\ell}^{s,L}(z)$ and $\mathcal{G}_{ijk\ell}^{t,L}(z)$. To solve for $A_n^{ijk\ell}$, we will apply a version of analytic functionals [73]. In particular, we will make use of bases of analytic functionals [74] dual to conformal blocks with integer scaling dimensions.[24]

Let $m_1, m_2, m_3, m_4 \in \{0, 1\}$ and $n \in \{2, 3, \ldots\}$. Let us define a family of linear functionals $\omega_n^{m_1, m_2, m_3, m_4}$ acting on functions $f(z)$ holomorphic in $\mathbb{C} \backslash (-\infty, 0] \cup [1, \infty)$ by the formula

$$\omega_n^{m_1, m_2, m_3, m_4}[f] = \int\limits_{\frac{1}{2}-i\infty}^{\frac{1}{2}-i\infty} G_{1-n}^{1-m_1, 1-m_2, 1-m_3, 1-m_4}(z) f(z) \frac{dz}{2\pi i}. \tag{5.22}$$

Note that the conformal block $G_{1-n}^{1-m_1, 1-m_2, 1-m_3, 1-m_4}(z)$ is well defined for the specified range of $m_{1,2,3,4}$ and $n$. It is a rational function of $z$ taking the form

$$G_{1-n}^{1-m_1, 1-m_2, 1-m_3, 1-m_4}(z) = z^{m_3+m_4-n-1} {}_2F_1(1 + m_1 - m_2 - n, 1 - m_3 + m_4 - n; 2 - 2n; z), \tag{5.23}$$

where the hypergeometric ${}_2F_1$ is a polynomial in $z$ of degree $n - 1 + \min(m_2 - m_1, m_3 - m_4)$. We claim that this set of functionals is dual to the s-channel conformal blocks of integer dimensions, in the sense

$$\omega_n^{m_1, m_2, m_3, m_4}[G_{n'}^{m_1, m_2, m_3, m_4}(\cdot)] = \delta_{nn'}, \tag{5.24}$$

for all integer $n, n' \geq 2$. Furthermore, all of the functionals annihilate the t-channel conformal blocks of integer dimensions

$$\omega_n^{m_1, m_2, m_3, m_4}[G_{n'}^{m_1, m_4, m_3, m_2}(1 - \cdot)] = 0, \tag{5.25}$$

for all $n, n' \in \mathbb{Z}$, $n, n' \geq 2$. To understand these statements, first note that the actions on s-channel and t-channel conformal blocks are finite. Indeed,

$$G_{1-n}^{1-m_1, 1-m_2, 1-m_3, 1-m_4}(z) G_{n'}^{m_1, m_2, m_3, m_4}(z) = O(z^{-2} \log |z|),$$
$$G_{1-n}^{1-m_1, 1-m_2, 1-m_3, 1-m_4}(z) G_{n'}^{m_1, m_4, m_3, m_2}(1 - z) = O(z^{-2} \log |z|) \tag{5.26}$$

as $z \to \pm i\infty$, and thus the integral in (5.22) converges. To evaluate (5.24), we close the contour to the left to pick up the residue at $z = 0$ and use

$$\text{Res}_{z=0}\left[ G_{1-n}^{1-m_1, 1-m_2, 1-m_3, 1-m_4}(z) G_{n'}^{m_1, m_2, m_3, m_4}(z) \right] = \delta_{nn'}, \tag{5.27}$$

which we state without proof.[25] To evaluate (5.25), we close the contour to the right. We encounter no pole at $z = 1$ since $n' \geq m_2 + m_3$, and thus the integral vanishes.

To solve for $A_n^{ijk\ell}$, we write the crossing equation (5.19) as

$$\sum_{n=2}^{\infty} A_n^{ijk\ell} \, G_n^{\Delta_i^{(0)}, \Delta_j^{(0)}, \Delta_k^{(0)}, \Delta_\ell^{(0)}}(z) = \mathcal{G}_{ijk\ell}^{t,L}(z) - \mathcal{G}_{ijk\ell}^{s,L}(z) + \sum_{n=2}^{\infty} A_n^{jk\ell i} \, G_n^{\Delta_i^{(0)}, \Delta_\ell^{(0)}, \Delta_k^{(0)}, \Delta_j^{(0)}}(1 - z). \tag{5.28}$$

Let us apply the analytic functional $\omega_n^{m_1, m_2, m_3, m_4}$ to this equation. If we can swap the functional action with the infinite sums on both sides, it follows from (5.24), (5.25) that

$$A_n^{ijk\ell} = \omega_n^{\Delta_i^{(0)}, \Delta_j^{(0)}, \Delta_k^{(0)}, \Delta_\ell^{(0)}}[\mathcal{G}_{ijk\ell}^{t,L}(z) - \mathcal{G}_{ijk\ell}^{s,L}(z)]. \tag{5.29}$$

---

[24]In fact the functionals used here are simpler than those of [74] since they only have simple, rather than double zeros at integer $\Delta$.

[25]The claim is a version of orthogonality of different solutions of the same Sturm-Liouville type ODE.

A sufficient condition for the functional to be swappable with the infinite sums is that the integral (5.22) is finite for $f(z) = \mathcal{G}^{s,H}_{ijk\ell}(z)$ and $f(z) = \mathcal{G}^{t,H}_{ijk\ell}(z)$ [75]. Whether this holds or not is directly determined by the asymptotics of $A^{ijk\ell}_n$ as $n \to \infty$. In the ensuing analysis, we will be able to check it directly. In fact, we will encounter situations where the swappability does not hold unless we use subtracted functionals, which possess a larger domain of swappability.

### 5.3.2 Crossing of $\langle \sigma\sigma\sigma\sigma \rangle$ at $O(\delta^2)$

Let us consider the case of $\langle \sigma\sigma\sigma\sigma \rangle$. The crossing equation (5.28) becomes

$$\sum_{n=2}^{\infty} A^{\sigma\sigma\sigma\sigma}_n G^{0,0,0,0}_n(z) = \mathcal{G}^{t,L}_{\sigma\sigma\sigma\sigma}(z) - \mathcal{G}^{s,L}_{\sigma\sigma\sigma\sigma}(z) + \sum_{n=2}^{\infty} A^{\sigma\sigma\sigma\sigma}_n G^{0,0,0,0}_n(1-z) \,. \tag{5.30}$$

Here

$$A^{\sigma\sigma\sigma\sigma}_n = \sum_{m:\Delta^{(0)}_m = n} (c^{(2)}_{\sigma\sigma m})^2 \geq 0 \,, \tag{5.31}$$

where we used $(-1)^{J_m} = 1$ for all $m$ such that $c_{\sigma\sigma m} \neq 0$. Let us recall the definitions

$$\begin{aligned}
\mathcal{G}^{s,L}_{\sigma\sigma\sigma\sigma}(z) &= \sum_{m\in\{0,+,-\}} \left[ (c_{\sigma\sigma m})^2 \, G^{\Delta_\sigma,\Delta_\sigma,\Delta_\sigma,\Delta_\sigma}_{\Delta_m}(z) \right]\Big|_{\delta^2} \,, \\
\mathcal{G}^{t,L}_{\sigma\sigma\sigma\sigma}(z) &= \mathcal{G}^{s,L}_{\sigma\sigma\sigma\sigma}(1-z) \,.
\end{aligned} \tag{5.32}$$

In order to extract $A^{\sigma\sigma\sigma\sigma}_n$ using functionals $\omega^{0,0,0,0}_n$, we need to check swappability. The idea is to use $A^{\sigma\sigma\sigma\sigma}_n \geq 0$ to relate the behaviour of

$$\mathcal{G}^{s,H}_{\sigma\sigma\sigma\sigma}(z) = \sum_{n=2}^{\infty} A^{\sigma\sigma\sigma\sigma}_n G^{0,0,0,0}_n(z) \tag{5.33}$$

in the limit $z \to i\infty$ to its behaviour as $z \to 1$. The latter can be estimated from the RHS of (5.30)

$$\sum_{n=2}^{\infty} A^{\sigma\sigma\sigma\sigma}_n G^{0,0,0,0}_n(z) \stackrel{z \to 1}{=} O((\log(1-z))^2) \,. \tag{5.34}$$

A standard argument using the radial coordinate [76] then implies

$$\sum_{n=2}^{\infty} A^{\sigma\sigma\sigma\sigma}_n G^{0,0,0,0}_n(z) \stackrel{z \to i\infty}{=} O((\log|z|)^2) \,. \tag{5.35}$$

Since $G^{1,1,1,1}_{1-n}(z) \stackrel{z \to i\infty}{=} O(z^{-2})$, it follows that the functionals are swappable and we get

$$A^{\sigma\sigma\sigma\sigma}_n = \omega^{0,0,0,0}_n[\mathcal{G}^{s,L}_{\sigma\sigma\sigma\sigma}(1-z) - \mathcal{G}^{s,L}_{\sigma\sigma\sigma\sigma}(z)] \,. \tag{5.36}$$

The integrals on the RHS can be done in a closed form

$$A^{\sigma\sigma\sigma\sigma}_n = \left[ (-1)^n n(n-1)(1 - 8c^{(1)}_{\sigma\chi-} - \Delta^{(2)}_+ - \Delta^{(2)}_-) + n(n-1) + 2 \right] \frac{((n-2)!)^2}{n^2(2n-2)!} \,. \tag{5.37}$$

Since $A^{\sigma\sigma\sigma\sigma}_n$ must be nonnegative, it follows that

$$0 \leq 8c^{(1)}_{\sigma\chi-} + \Delta^{(2)}_+ + \Delta^{(2)}_- \leq 2 \,. \tag{5.38}$$

The sum over conformal blocks (5.33) defining $\mathcal{G}^{s,H}_{\sigma\sigma\sigma\sigma}(z)$ can now also be done in a closed form. The resulting correlator $\mathcal{G}^{s,L}_{\sigma\sigma\sigma\sigma}(z) + \mathcal{G}^{s,H}_{\sigma\sigma\sigma\sigma}(z)$ is crossing symmetric if and only if the following additional constraint holds:

$$\frac{c^{(3)}_{\sigma\sigma-} - c^{(3)}_{\sigma\sigma+}}{\sqrt{2}} = c^{(1)}_{\sigma\chi-}(c^{(1)}_{\sigma\chi-} - 2) + \frac{\pi^2}{6} - 2. \tag{5.39}$$

This identity is also equivalent to $\omega^{0,0,0,0}_1[\mathcal{G}^{s,L}_{\sigma\sigma\sigma\sigma}(1-z) - \mathcal{G}^{s,L}_{\sigma\sigma\sigma\sigma}(z)] = 0$, which holds because in this case the functional $\omega^{0,0,0,0}_1$ is well-defined and swappable. We will present the answer for the full correlator in (5.121), once all the unknowns have been fixed.

### 5.3.3 Crossing of $\langle\sigma\sigma\chi\chi\rangle$ at $O(\delta)$

Let us proceed by studying the correlator $\langle\sigma\sigma\chi\chi\rangle$. The crossing equation at $O(\delta)$ reads

$$\sum_{n=2}^{\infty} A^{\sigma\sigma\chi\chi}_n G^{0,0,1,1}_n(z) = \mathcal{G}^{t,L}_{\sigma\sigma\chi\chi}(z) - \mathcal{G}^{s,L}_{\sigma\sigma\chi\chi}(z) + \sum_{n=2}^{\infty} A^{\sigma\chi\chi\sigma}_n G^{0,1,1,0}_n(1-z). \tag{5.40}$$

Here

$$A^{\sigma\sigma\chi\chi}_n = \sum_{m:\Delta^{(0)}_m=n} c^{(2)}_{\sigma\sigma m} c^{(0)}_{\chi\chi m}, \quad A^{\sigma\chi\chi\sigma}_n = \sum_{m:\Delta^{(0)}_m=n} (c^{(1)}_{\sigma\chi m})^2 \geq 0, \tag{5.41}$$

where we used $c^{(1)}_{\chi\sigma m} = (-1)^{J_m} c^{(1)}_{\sigma\chi m}$. Recall that the $\chi \times \chi$ OPE at $\delta = 0$ only contains double-trace operators, all of which have even scaling dimensions. It follows that $A^{\sigma\sigma\chi\chi}_n = 0$ for $n$ odd. We have

$$\mathcal{G}^{s,L}_{\sigma\sigma\chi\chi}(z) = \sum_{m\in\{0,+,-\}} \left[c_{\sigma\sigma m} c_{\chi\chi m} G^{\Delta_\sigma,\Delta_\sigma,\Delta_\chi,\Delta_\chi}_{\Delta_m}(z)\right]\Big|_\delta,$$
$$\mathcal{G}^{t,L}_{\sigma\sigma\chi\chi}(z) = \sum_{m\in\{\sigma,\chi\}} \left[(c_{\sigma\chi m})^2 G^{\Delta_\sigma,\Delta_\chi,\Delta_\chi,\Delta_\sigma}_{\Delta_m}(1-z)\right]\Big|_\delta. \tag{5.42}$$

To apply the analytic functionals and solve for $A^{\sigma\sigma\chi\chi}_n$, we need to analyze the $z \to i\infty$ limit of $\mathcal{G}^{s,H}_{\sigma\sigma\chi\chi}(z)$. We have for all $z \in \mathbb{C}\backslash(-\infty,0] \cup [1,\infty)$

$$\left|\sum_{n=2}^{\infty} A^{\sigma\sigma\chi\chi}_n G^{0,0,1,1}_n(z)\right| = \left|\sum_{m:\Delta^{(0)}_m\geq 2}^{\infty} c^{(2)}_{\sigma\sigma m} c^{(0)}_{\chi\chi m} G^{0,0,1,1}_n(z)\right|$$

$$\leq \sum_{m:\Delta^{(0)}_m\geq 2}^{\infty} |c^{(2)}_{\sigma\sigma m}||c^{(0)}_{\chi\chi m}|\,|G^{0,0,1,1}_n(z)|$$

$$= |z|^{-1} \sum_{m:\Delta^{(0)}_m\geq 2}^{\infty} |c^{(2)}_{\sigma\sigma m}|\sqrt{|G^{0,0,0,0}_n(z)|}|c^{(0)}_{\chi\chi m}|\sqrt{|G^{1,1,1,1}_n(z)|}$$

$$\leq |z|^{-1}\sqrt{\sum_{m:\Delta^{(0)}_m\geq 2}^{\infty} (c^{(2)}_{\sigma\sigma m})^2\,|G^{0,0,0,0}_n(z)|}\sqrt{\sum_{m:\Delta^{(0)}_m\geq 2}^{\infty} (c^{(0)}_{\chi\chi m})^2\,|G^{1,1,1,1}_n(z)|}. \tag{5.43}$$

The first equality follows by definition of $A^{\sigma\sigma\chi\chi}_n$, and the last inequality is Cauchy-Schwartz. We have already estimated the argument of the first square root in (5.35). The argument of the second square root is essentially $\langle\chi\chi\chi\chi\rangle$. The standard radial-coordinate argument relating $z \to i\infty$ to $z \to 1$ implies it is bounded in the limit $z \to i\infty$. It follows

$$\left|\sum_{n=2}^{\infty} A^{\sigma\sigma\chi\chi}_n G^{0,0,1,1}_n(z)\right| \overset{z\to i\infty}{=} O(|z^{-1}\log(z)|). \tag{5.44}$$

The kernels defining the functionals $\omega_n^{0,0,1,1}$ approach a constant at infinity. It follows that they do not necessarily commute with the sum over conformal blocks. To remedy the situation, we define the subtracted functionals

$$\widetilde{\omega}_n^{0,0,1,1} = \omega_n^{0,0,1,1} + (-1)^n \frac{((n-1)!)^2}{(2n-2)!} \omega_1^{0,0,1,1} \tag{5.45}$$

for $n \geq 2$. The kernel of $\widetilde{\omega}_n^{0,0,1,1}$ is $O(z^{-1})$ and therefore commutes with the sums over $n$ in (5.40). We get

$$A_n^{\sigma\sigma\chi\chi} = \widetilde{\omega}_n^{0,0,1,1}[\mathcal{G}_{\sigma\sigma\chi\chi}^{t,L}(z) - \mathcal{G}_{\sigma\sigma\chi\chi}^{s,L}(z)]. \tag{5.46}$$

The integrals defining the functional action can be done in a closed form. The condition that $A_n^{\sigma\sigma\chi\chi} = 0$ for $n$ odd is equivalent with the constraint

$$\sqrt{2}(c_{\chi\chi+}^{(1)} - c_{\chi\chi-}^{(1)}) = 8c_{\sigma\chi-}^{(1)} - \Delta_+^{(2)} - \Delta_-^{(2)}. \tag{5.47}$$

The final answer is

$$A_n^{\sigma\sigma\chi\chi} = [1 + (-1)^n] \frac{2(n-1)((n-2)!)^2}{n(2n-2)!}. \tag{5.48}$$

$\mathcal{G}_{\sigma,\sigma,\chi,\chi}^{s,H}(z)$ can now be evaluated in a closed form

$$\mathcal{G}_{\sigma\sigma\chi\chi}^{s,H}(z) = \sum_{n=2}^{\infty} A_n^{\sigma\sigma\chi\chi} G_n^{0,0,1,1}(z) = \frac{[\log(1-z)]^2}{z^2}. \tag{5.49}$$

Let us now rewrite (5.40) as

$$\sum_{n=2}^{\infty} A_n^{\sigma\chi\chi\sigma} G_n^{0,1,1,0}(1-z) = \mathcal{G}_{\sigma\sigma\chi\chi}^{s,L}(z) + \mathcal{G}_{\sigma\sigma\chi\chi}^{s,H}(z) - \mathcal{G}_{\sigma\sigma\chi\chi}^{t,L}(z) \tag{5.50}$$

The RHS admits the expansion on the LHS if and only if

$$c_{\sigma\chi+}^{(2)} + c_{\sigma\chi-}^{(2)} + \sqrt{2}(c_{\sigma\chi-}^{(1)})^2 = 0. \tag{5.51}$$

The formula for $A_n^{\sigma\chi\chi\sigma}$ then reads

$$A_n^{\sigma\chi\chi\sigma} = [n(n-1) - 2(-1)^n] \frac{((n-2)!)^2}{(2n-2)!}. \tag{5.52}$$

Note that this formula passes the nontrivial consistency check $A_n^{\sigma\chi\chi\sigma} \geq 0$ for all $n \geq 2$.

At this point, we can obtain another constraint on the CFT data by recalling that the space of $\mathbb{Z}_2$-even primaries with $\Delta^{(0)} = 2$ is one-dimensional, spanned by $\rho =: \chi^2: /\sqrt{2}$. It follows that

$$A_2^{\sigma\sigma\sigma\sigma} = (c_{\sigma\sigma\rho}^{(2)})^2, \qquad A_2^{\sigma\sigma\chi\chi} = c_{\sigma\sigma\rho}^{(2)} c_{\chi\chi\rho}^{(0)}. \tag{5.53}$$

It follows directly from the definition of $\rho$ that $c_{\chi\chi\rho}^{(0)} = \sqrt{2}$. Equation (5.48) then predicts

$$c_{\sigma\sigma\rho}^{(2)} = \frac{1}{\sqrt{2}}. \tag{5.54}$$

Hence $A_2^{\sigma\sigma\sigma\sigma} = 1/2$. By comparing this prediction to (5.37), we obtain the constraint

$$8c_{\sigma\chi-}^{(1)} + \Delta_+^{(2)} + \Delta_-^{(2)} = 1, \tag{5.55}$$

which in particular satisfies (5.38) and which allows us to simplify the formula for $A_n^{\sigma\sigma\sigma\sigma}$

$$A_n^{\sigma\sigma\sigma\sigma} = [n(n-1) + 2] \frac{((n-2)!)^2}{n^2(2n-2)!}. \tag{5.56}$$

### 5.3.4 Crossing of $\langle\sigma\chi\sigma\chi\rangle$ at $O(\delta)$

Our next step is to consider the crossing equation of $\langle\sigma\chi\sigma\chi\rangle$ at $O(\delta)$

$$\sum_{n=2}^{\infty} A_n^{\sigma\chi\sigma\chi} G_n^{0,1,0,1}(z) = \mathcal{G}_{\sigma\chi\sigma\chi}^{s,L}(1-z) - \mathcal{G}_{\sigma\chi\sigma\chi}^{s,L}(z) + \sum_{n=2}^{\infty} A_n^{\sigma\chi\sigma\chi} G_n^{0,1,0,1}(1-z). \tag{5.57}$$

Here

$$A_n^{\sigma\chi\sigma\chi} = \sum_{m:\Delta_m^{(0)}=n} (-1)^{J_m} (c_{\sigma\chi m}^{(1)})^2, \tag{5.58}$$

and

$$\mathcal{G}_{\sigma\chi\sigma\chi}^{s,L}(z) = \sum_{m\in\{+,-\}} \left[ (c_{\sigma\chi m})^2 \, G_{\Delta_m}^{\Delta_\sigma,\Delta_\chi,\Delta_\sigma,\Delta_\chi}(z) \right]\Big|_\delta. \tag{5.59}$$

To study the swappability of functionals, we need to estimate

$$\mathcal{G}_{\sigma\chi\sigma\chi}^{s,H}(z) = \sum_{n=2}^{\infty} A_n^{\sigma\chi\sigma\chi} G_n^{0,1,0,1}(z) \tag{5.60}$$

as $z \to i\infty$. Note that we have $|A_n^{\sigma\chi\sigma\chi}| \leq A_n^{\sigma\chi\chi\sigma}$. It follows

$$
\begin{aligned}
|\mathcal{G}_{\sigma\chi\sigma\chi}^{s,H}(z)| &\leq \sum_{n=2}^{\infty} A_n^{\sigma\chi\chi\sigma} \, |G_n^{0,1,0,1}(z)| \\
&= \sum_{n=2}^{\infty} A_n^{\sigma\chi\chi\sigma} \, |\tfrac{z}{z-1}|^{n-1} |_2F_1(n-1,n-1;2n;\tfrac{z}{z-1})| \\
&\leq \sum_{n=2}^{\infty} [n(n-1) - 2(-1)^n] \frac{((n-2)!)^2}{(2n-2)!} \, |\tfrac{z}{z-1}|^{n-1} {}_2F_1(n-1,n-1;2n;|\tfrac{z}{z-1}|) \\
&= O(\log(z)),
\end{aligned}
\tag{5.61}
$$

To go to the second line, we used a standard transformation of $_2F_1$ hypergeometric functions. To go to the third line, we used (5.52) and the positivity of coefficients of the $y$ expansion in the $_2F_1$. To go to the last line, we explicitly evaluated the sum on the third line and took the limit $|y| \to 1$.

Since the kernels of the functionals $\omega_n^{0,1,0,1}$ go like $z^{-2}$ for $n \geq 2$, these can be safely swapped with the sum over conformal blocks. We find

$$A_n^{\sigma\chi\sigma\chi} = \omega_n^{0,1,0,1}[\mathcal{G}_{\sigma\chi\sigma\chi}^{s,L}(1-z) - \mathcal{G}_{\sigma\chi\sigma\chi}^{s,L}(z)] = [(-1)^{n-1}n(n-1)(\Delta_+^{(2)} + \Delta_-^{(2)}) + 1] \frac{2((n-2)!)^2}{(2n-2)!}. \tag{5.62}$$

Let us recall again that due to the factor $(-1)^{J_m}$, we must have $|A_n^{\sigma\chi\sigma\chi}| \leq A_n^{\sigma\chi\chi\sigma}$ and note that $A_2^{\sigma\chi\chi\sigma} = 0$. It follows $A_2^{\sigma\chi\sigma\chi} = 0$, which gives the constraint

$$\Delta_+^{(2)} + \Delta_-^{(2)} = \frac{1}{2}. \tag{5.63}$$

Hence

$$A_n^{\sigma\chi\sigma\chi} = [(-1)^{n-1}n(n-1) + 2] \frac{((n-2)!)^2}{(2n-2)!} = (-1)^{n-1} A_n^{\sigma\chi\chi\sigma}. \tag{5.64}$$

The last equality implies that, for all primary operators $\mathcal{P}_m$, $c_{\sigma\chi m}^{(1)} \neq 0$ only if $(-1)^{J_m} = (-1)^{\Delta_m^{(0)}-1}$. Since there are no parity-odd primaries with $\Delta_m^{(0)} = 2$, this fact explains why we found $A_2^{\sigma\chi\chi\sigma} = 0$ in the first place.

### 5.3.5 Summary of results so far

So far, we have implemented crossing of $\langle\sigma\sigma\mathcal{O}_a\mathcal{O}_b\rangle$ up to $O(\sqrt{\delta})$, $\langle\sigma\sigma\chi\chi\rangle$ and $\langle\sigma\chi\sigma\chi\rangle$ up to $O(\delta)$, and $\langle\sigma\sigma\sigma\sigma\rangle$ up to $O(\delta^2)$. This has lead to the following constraints on the CFT data:

$$
\begin{aligned}
\Delta_{\pm}^{(1)} &= \pm\sqrt{2}, \quad \Delta_{+}^{(2)} + \Delta_{-}^{(2)} = \frac{1}{2}, \\
c_{\sigma\chi\pm}^{(0)} &= \frac{1}{\sqrt{2}}, \quad c_{\sigma\chi\pm}^{(1)} = \mp\frac{1}{16}, \quad c_{\sigma\chi+}^{(2)} + c_{\sigma\chi-}^{(2)} = -\frac{1}{128\sqrt{2}}, \\
c_{\sigma\sigma\pm}^{(1)} &= \pm\frac{1}{\sqrt{2}}, \quad c_{\sigma\sigma\pm}^{(2)} = -\frac{15}{16}, \quad c_{\sigma\sigma+}^{(3)} - c_{\sigma\sigma-}^{(3)} = \sqrt{2}\left(\frac{543}{256} - \frac{\pi^2}{6}\right), \\
c_{\chi\chi+}^{(1)} &- c_{\chi\chi-}^{(1)} = 0, \\
c_{+++}^{(0)} &= c_{---}^{(0)} = \frac{3}{2}, \quad c_{++-}^{(0)} = c_{+--}^{(0)} = -\frac{1}{2}.
\end{aligned}
\tag{5.65}
$$

We will now consider crossing symmetry of the remaining correlators in (5.17).

### 5.3.6 Crossing of $\langle\sigma\sigma\sigma\chi\rangle$ at $O(\delta^{3/2})$

The crossing equation takes the form

$$
\sum_{n=2}^{\infty} A_n^{\sigma\sigma\sigma\chi} G_n^{0,0,0,1}(z) = \mathcal{G}_{\sigma\sigma\sigma\chi}^{t,L}(z) - \mathcal{G}_{\sigma\sigma\sigma\chi}^{s,L}(z) + \sum_{n=2}^{\infty} A_n^{\sigma\sigma\sigma\chi} G_n^{0,1,0,0}(1-z).
\tag{5.66}
$$

Here

$$
A_n^{\sigma\sigma\sigma\chi} = \sum_{m:\Delta_m^{(0)}=n} c_{\sigma\sigma m}^{(2)} c_{\sigma\chi m}^{(1)},
\tag{5.67}
$$

and

$$
\begin{aligned}
\mathcal{G}_{\sigma\sigma\sigma\chi}^{s,L}(z) &= \sum_{m\in\{+,-\}} \left[c_{\sigma\sigma m} c_{\sigma\chi m} G_{\Delta_m}^{\Delta_\sigma,\Delta_\sigma,\Delta_\sigma,\Delta_\chi}(z)\right]\Big|_{\delta^{\frac{3}{2}}} \\
\mathcal{G}_{\sigma\sigma\sigma\chi}^{t,L}(z) &= \sum_{m\in\{+,-\}} \left[c_{\sigma\sigma m} c_{\sigma\chi m} G_{\Delta_m}^{\Delta_\sigma,\Delta_\chi,\Delta_\sigma,\Delta_\sigma}(1-z)\right]\Big|_{\delta^{\frac{3}{2}}}.
\end{aligned}
\tag{5.68}
$$

To check for the swappability of functionals, we estimate

$$
\begin{aligned}
|\sum_{n=2}^{\infty} A_n^{\sigma\sigma\sigma\chi} G_n^{0,0,0,1}(z)| &\leq \sum_{n=2}^{\infty} |A_n^{\sigma\sigma\sigma\chi}| \, |G_n^{0,0,0,1}(z)| \\
&\leq \sum_{n=2}^{\infty} \sqrt{A_n^{\sigma\sigma\sigma\sigma}}\sqrt{A_n^{\sigma\chi\chi\sigma}} \, |\tfrac{z}{z-1}|^{n-1} |{}_2F_1(n-1,n;2n;\tfrac{z}{z-1})| \\
&\leq \sum_{n=2}^{\infty} \frac{[(n-1)n+2]((n-2)!)^2}{n(2n-2)!} |\tfrac{z}{z-1}|^{n-1} {}_2F_1(n-1,n;2n;|\tfrac{z}{z-1}|) \\
&= O(\log z).
\end{aligned}
\tag{5.69}
$$

Since the kernels of the $\omega_n^{0,0,0,1}$ functionals are $O(z^{-2})$ for all $n \geq 2$, they can be swapped with the sum over conformal blocks. We find

$$
A_n^{\sigma\sigma\sigma\chi} = \omega_n^{0,0,0,1}[\mathcal{G}_{\sigma\sigma\sigma\chi}^{t,L}(z) - \mathcal{G}_{\sigma\sigma\sigma\chi}^{s,L}(z)] = 0,
\tag{5.70}
$$

for all $n \geq 0$. Note that based on previous information, we could have predicted that $A_n^{\sigma\sigma\chi} = 0$ for $n$ even. This is because $c_{\sigma\sigma m}^{(2)} \neq 0$ only if $(-1)^{J_m} = 1$, and $c_{\sigma\chi m}^{(1)} \neq 0$ only if $(-1)^{J_m} = (-1)^{\Delta_m^{(0)}-1}$. For $n$ odd, (5.70) is a new fact.

We can now go back to the crossing equation (5.66), which becomes $\mathcal{G}_{\sigma\sigma\sigma\chi}^{s,L}(z) = \mathcal{G}_{\sigma\sigma\sigma\chi}^{t,L}(z)$. This is in turn equivalent to the new constraint

$$\Delta_-^{(2)} + \sqrt{2}c_{\sigma\chi-}^{(2)} - \sqrt{2}c_{\sigma\sigma-}^{(3)} = \frac{303}{128} - \frac{\pi^2}{6}. \tag{5.71}$$

### 5.3.7 Crossing of $\langle\sigma\chi\chi\chi\rangle$ at $O(\sqrt{\delta})$

The crossing equation takes the form

$$\mathcal{G}_{\sigma\chi\chi\chi}^{s,L}(z) + \sum_{n=2}^{\infty} A_n^{\sigma\chi\chi\chi} G_n^{0,1,1,1}(z) = \mathcal{G}_{\sigma\chi\chi\chi}^{s,L}(1-z) + \sum_{n=2}^{\infty} A_n^{\sigma\chi\chi\chi} G_n^{0,1,1,1}(1-z), \tag{5.72}$$

where

$$\mathcal{G}_{\sigma\chi\chi\chi}^{s,L}(z) = \sum_{m\in\{+,-\}}\left[c_{\sigma\chi m}c_{\chi\chi m}\,G_{\Delta_m}^{\Delta_\sigma,\Delta_\chi,\Delta_\chi,\Delta_\chi}(z)\right]\Big|_{\delta^{\frac{1}{2}}}\,, \quad A_n^{\sigma\chi\chi\chi} = \sum_{m:\Delta_m^{(0)}=n} c_{\sigma\chi m}^{(1)}c_{\chi\chi m}^{(0)}\,. \tag{5.73}$$

Based on previous findings, we can conclude that in fact $A_n^{\sigma\chi\chi\chi} = 0$ for all $n$. Indeed, since the $\chi \times \chi$ OPE at $\delta = 0$ contains only the identity and double traces, we have $c_{\chi\chi m}^{(0)} \neq 0$ only if $(-1)^{J_m} = (-1)^{\Delta_m^{(0)}} = 1$. On the other hand, we have seen that $c_{\sigma\chi m}^{(1)} \neq 0$ only if $(-1)^{J_m} = (-1)^{\Delta_m^{(0)}-1}$. Thus $c_{\sigma\chi m}^{(1)}c_{\chi\chi m}^{(0)} = 0$ for all primaries $\mathcal{P}_m$.

The crossing equation (5.72) does not impose any additional constraints on the CFT data.

### 5.3.8 Crossing of $\langle\sigma\sigma\mathcal{O}_\pm\mathcal{O}_\pm\rangle$ at $O(\delta)$

The crossing equations take the form

$$\sum_{n=2}^{\infty} A_n^{\sigma\sigma ab} G_n^{0,0,1,1}(z) = \mathcal{G}_{\sigma\sigma ab}^{t,L}(z) - \mathcal{G}_{\sigma\sigma ab}^{s,L}(z) + \sum_{n=2}^{\infty} A_n^{\sigma ab\sigma} G_n^{0,1,1,0}(1-z), \tag{5.74}$$

with $a, b \in \{+, -\}$. Here

$$A_n^{\sigma\sigma ab} = \sum_{m:\Delta_m^{(0)}=n} c_{\sigma\sigma m}^{(2)}c_{abm}^{(0)}\,, \quad A_n^{\sigma ab\sigma} = \sum_{m:\Delta_m^{(0)}=n} c_{\sigma am}^{(1)}c_{\sigma bm}^{(1)}\,, \tag{5.75}$$

and

$$\mathcal{G}_{\sigma\sigma ab}^{s,L}(z) = \sum_{m\in\{0,+,-\}}\left[c_{\sigma\sigma m}c_{abm}\,G_{\Delta_m}^{\Delta_\sigma,\Delta_\sigma,\Delta_a,\Delta_b}(z)\right]\Big|_{\delta}$$

$$\mathcal{G}_{\sigma\sigma ab}^{t,L}(z) = \sum_{m\in\{\sigma,\chi\}}\left[c_{\sigma ma}c_{\sigma mb}\,G_{\Delta_m}^{\Delta_\sigma,\Delta_b,\Delta_a,\Delta_\sigma}(1-z)\right]\Big|_{\delta}\,. \tag{5.76}$$

The situation with swappability of functionals is identical to the correlator $\langle\sigma\sigma\chi\chi\rangle$, discussed in Section 5.3.3. In particular, $\omega_n^{0,0,1,1}$ are not swappable, but the subtracted version $\widetilde{\omega}_n^{0,0,1,1}$, defined in (5.45), are swappable. We have

$$A_n^{\sigma\sigma ab} = \widetilde{\omega}_n^{0,0,1,1}[\mathcal{G}_{\sigma\sigma ab}^{t,L}(z) - \mathcal{G}_{\sigma\sigma ab}^{s,L}(z)]\,. \tag{5.77}$$

We then perform the sums over conformal blocks to obtain

$$\mathcal{G}^{s,H}_{\sigma\sigma ab}(z) = \sum_{n=2}^{\infty} A^{\sigma\sigma ab}_n\, G^{0,0,1,1}_n(z)\,. \tag{5.78}$$

Then we extract $A^{\sigma ab\sigma}_n$ from

$$\sum_{n=2}^{\infty} A^{\sigma ab\sigma}_n\, G^{0,1,1,0}_n(z) = \mathcal{G}^{s,L}_{\sigma\sigma ab}(1-z) + \mathcal{G}^{s,H}_{\sigma\sigma ab}(1-z) - \mathcal{G}^{t,L}_{\sigma\sigma ab}(1-z)\,. \tag{5.79}$$

It turns out that the existence of the expansion on the LHS imposes additional constraints on the CFT data. If these constraints are not satisfied, the RHS contains a spurious conformal block $G^{0,1,1,0}_1(z)$. The constraints are

$$2\Delta^{(2)}_- + \sqrt{2}(c^{(1)}_{+++} - c^{(1)}_{++-}) = \frac{23}{8}\,, \quad 2\Delta^{(2)}_- + \sqrt{2}(c^{(1)}_{---} - c^{(1)}_{+--}) = -\frac{15}{8}\,, \quad \sqrt{2}(c^{(1)}_{++-} - c^{(1)}_{+--}) = \frac{1}{8}\,. \tag{5.80}$$

We can use these to solve for $c^{(1)}_{+++}$, $c^{(1)}_{++-}$, and $c^{(1)}_{+--}$ in terms of $\Delta^{(2)}_-$ and $c^{(1)}_{---}$. Once we impose the constraints, we obtain

$$A^{\sigma\sigma++}_n = A^{\sigma\sigma--}_n = \frac{(n-1)\,[n(n-1)+4+2(-1)^n]}{2n}\frac{((n-2)!)^2}{(2n-2)!}\,,$$
$$A^{\sigma\sigma+-}_n = A^{\sigma\sigma-+}_n = -\frac{(n-1)\,[n(n-1)-2(-1)^n]}{2n}\frac{((n-2)!)^2}{(2n-2)!}\,, \tag{5.81}$$

and

$$A^{\sigma++\sigma}_n = A^{\sigma--\sigma}_n = [n(n-1)+1-(-1)^n]\frac{((n-2)!)^2}{(2n-2)!}\,,$$
$$A^{\sigma+-\sigma}_n = A^{\sigma-+\sigma}_n = -[1+(-1)^n]\frac{((n-2)!)^2}{(2n-2)!}\,. \tag{5.82}$$

The above formulas for $A^{\sigma\sigma ab}_n$ and $A^{\sigma ab\sigma}_n$ pass several nontrivial consistency checks. Firstly, recall that the spaces of both $\mathbb{Z}_2$-even and $\mathbb{Z}_2$-odd primaries with $\Delta^{(0)} = 2$ are one-dimensional, spanned by $\rho$, $\widetilde{\rho}$. As $\delta \to 0$, we have $\rho \to :\chi^2: /\sqrt{2}$ and $\widetilde{\rho} \to :\sigma\chi^2: /\sqrt{2}$. It follows that

$$A^{\sigma\sigma ab}_2 = c^{(2)}_{\sigma\sigma\rho}c^{(0)}_{ab\rho}\,, \quad A^{\sigma ab\sigma}_2 = c^{(1)}_{\sigma a\widetilde{\rho}}c^{(1)}_{\sigma b\widetilde{\rho}}\,. \tag{5.83}$$

Recall from (5.54) that $c^{(2)}_{\sigma\sigma\rho} = 1/\sqrt{2}$. Equations (5.81) then predict

$$c^{(0)}_{++\rho} = c^{(0)}_{--\rho} = \sqrt{2}\,, \quad c^{(0)}_{+-\rho} = 0\,. \tag{5.84}$$

These results can be verified explicitly in the free theory at $\delta = 0$. Indeed, they are equivalent to the following three-point functions between $\rho$ and the operators $\mathcal{O}_g$, $\mathcal{O}_h$

$$c_{gg\rho} = c_{hh\rho} = \sqrt{2}\,, \quad c_{gh\rho} = 0\,. \tag{5.85}$$

$c_{gh\rho} = 0$ is immediate from the matrix structure, $c_{hh\rho} = \sqrt{2}$ is a simple exercise with Wick contractions. $c_{gg\rho} = \sqrt{2}$ requires more work but holds too.

For another consistency check, involving $\widetilde{\rho}$, note that (5.83) implies

$$A^{\sigma++\sigma}_2 A^{\sigma--\sigma}_2 = (A^{\sigma+-\sigma}_2)^2\,. \tag{5.86}$$

Indeed, (5.82) predicts that both sides equal 1.

### 5.3.9 Crossing of $\langle \sigma\mathcal{O}_\pm\sigma\mathcal{O}_\pm\rangle$ at $O(\delta)$

The crossing equations takes the form

$$\sum_{n=2}^{\infty} A_n^{\sigma a\sigma b}\, G_n^{0,1,0,1}(z) = \mathcal{G}_{\sigma a\sigma b}^{t,L}(z) - \mathcal{G}_{\sigma a\sigma b}^{s,L}(z) + \sum_{n=2}^{\infty} A_n^{\sigma a\sigma b}\, G_n^{0,1,0,1}(1-z)\,, \tag{5.87}$$

with $a, b \in \{+,-\}$. Here

$$A_n^{\sigma a\sigma b} = \sum_{m:\Delta_m^{(0)}=n} (-1)^{J_m} c_{\sigma am}^{(1)} c_{\sigma bm}^{(1)}\,, \tag{5.88}$$

and

$$\mathcal{G}_{\sigma a\sigma b}^{s,L}(z) = \sum_{m\in\{\sigma,\chi\}} \left[ c_{\sigma am} c_{\sigma bm}\, G_{\Delta_m}^{\Delta_\sigma,\Delta_a,\Delta_\sigma,\Delta_b}(z) \right]\Big|_\delta$$

$$\mathcal{G}_{\sigma a\sigma b}^{t,L}(z) = \sum_{m\in\{\sigma,\chi\}} \left[ c_{\sigma am} c_{\sigma bm}\, G_{\Delta_m}^{\Delta_\sigma,\Delta_b,\Delta_\sigma,\Delta_a}(1-z) \right]\Big|_\delta = \mathcal{G}_{\sigma b\sigma a}^{s,L}(1-z)\,. \tag{5.89}$$

To study the swappability of functionals, note that

$$|A_n^{\sigma+\sigma+}| \le A_n^{\sigma++\sigma}\,, \qquad |A_n^{\sigma-\sigma-}| \le A_n^{\sigma--\sigma} = A_n^{\sigma++\sigma}\,, \qquad |A_n^{\sigma-+\sigma}| \le \sqrt{A_n^{\sigma++\sigma} A_n^{\sigma--\sigma}} = A_n^{\sigma++\sigma}\,. \tag{5.90}$$

Using the same reasoning as in Section 5.3.4, we conclude that the functionals $\omega_n^{0,1,0,1}$ are swappable. Their application produces

$$A_n^{\sigma+\sigma+} = A_n^{\sigma-\sigma-} = [(-1)^n + 1]\frac{n!(n-2)!}{2(2n-2)!}\,,$$

$$A_n^{\sigma+\sigma-} = A_n^{\sigma-\sigma+} = \{[(-1)^n - 1]n(n-1) - 4\}\frac{((n-2)!)^2}{2(2n-2)!}\,. \tag{5.91}$$

### 5.3.10 Crossing of $\langle \sigma\chi\mathcal{O}_\pm\mathcal{O}_\pm\rangle$ at $O(\sqrt{\delta})$

The crossing equations take the form

$$\sum_{n=2}^{\infty} A_n^{\sigma\chi ab}\, G_n^{0,1,1,1}(z) = \mathcal{G}_{\sigma\chi ab}^{t,L}(z) - \mathcal{G}_{\sigma\chi ab}^{s,L}(z) + \sum_{n=2}^{\infty} A_n^{\sigma ba\chi}\, G_n^{0,1,1,1}(1-z)\,, \tag{5.92}$$

with $a, b \in \{+,-\}$. Here

$$A_n^{\sigma\chi ab} = \sum_{m:\Delta_m^{(0)}=n} (-1)^{J_m} c_{\sigma\chi m}^{(1)} c_{abm}^{(0)}\,,$$

$$A_n^{\sigma ab\chi} = \sum_{m:\Delta_m^{(0)}=n} (-1)^{J_m} c_{\sigma am}^{(1)} c_{b\chi m}^{(0)} = \sum_{m:\Delta_m^{(0)}=n} c_{\sigma am}^{(1)} c_{\chi bm}^{(0)}\,, \tag{5.93}$$

and

$$\mathcal{G}_{\sigma\chi ab}^{s,L}(z) = \sum_{m\in\{+,-\}} \left[ c_{\sigma\chi m} c_{abm}\, G_{\Delta_m}^{\Delta_\sigma,\Delta_\chi,\Delta_a,\Delta_b}(z) \right]\Big|_{\sqrt{\delta}}$$

$$\mathcal{G}_{\sigma\chi ab}^{t,L}(z) = \sum_{m\in\{\sigma,\chi\}} \left[ c_{\chi am} c_{\sigma bm}\, G_{\Delta_m}^{\Delta_\sigma,\Delta_b,\Delta_a,\Delta_\chi}(1-z) \right]\Big|_{\sqrt{\delta}}\,. \tag{5.94}$$

Following the same logic as in the previous subsections, we find that the functionals $\omega_n^{0,1,1,1}$ are not swappable with the sum over conformal blocks on the LHS of (5.92). We can define subtracted functionals

$$\widetilde{\omega}_n^{0,1,1,1} = \omega_n^{0,1,1,1} + (-1)^n \frac{n((n-1)!)^2}{(2n-2)!}\omega_1^{0,1,1,1}\,, \tag{5.95}$$

which are swappable. We obtain

$$A_n^{\sigma\chi ab} = \widetilde{\omega}_n^{0,1,1,1}[\mathcal{G}_{\sigma\chi ab}^{t,L}(z) - \mathcal{G}_{\sigma\chi ab}^{s,L}(z)] \,. \tag{5.96}$$

Since the space of $\mathbb{Z}_2$-even primaries with $\Delta^{(0)} = 2$ is spanned by $\rho$, and since we know from (5.52) that $c_{\sigma\chi\rho}^{(1)} = 0$, we conclude $A_2^{\sigma\chi++} = A_2^{\sigma\chi--} = A_2^{\sigma\chi+-} = 0$. This condition imposes the following constraint on the CFT data

$$24\Delta_-^{(2)} + 16\sqrt{2}c_{---}^{(1)} - 8\sqrt{2}c_{\chi\chi-}^{(1)} + 33 = 0 \,. \tag{5.97}$$

The functionals (5.96) and the crossing equation (5.92) then lead to a unique answer for the coefficients

$$A_n^{\sigma\chi++} = -A_n^{\sigma\chi--} = [1 - (-1)^n][n(n-1) + 2]\frac{(n-1)!(n-2)!}{2(2n-2)!} \,,$$
$$A_n^{\sigma\chi+-} = -A_n^{\sigma\chi-+} = -[1 + (-1)^n][n(n-1) - 2]\frac{(n-1)!(n-2)!}{2(2n-2)!} \,, \tag{5.98}$$

and

$$A_n^{\sigma++\chi} = -A_n^{\sigma--\chi} = [(-1)^n(n-2)(n+1) - 4]\frac{(n-1)!(n-2)!}{2(2n-2)!} \,,$$
$$A_n^{\sigma+-\chi} = -A_n^{\sigma-+\chi} = (-1)^{n+1}[n(n-1) + 2]\frac{(n-1)!(n-2)!}{2(2n-2)!} \,. \tag{5.99}$$

### 5.3.11 Crossing of $\langle\sigma\mathcal{O}_\pm\chi\mathcal{O}_\pm\rangle$ at $O(\sqrt{\delta})$

The last crossing equation that we will consider takes the form

$$\sum_{n=2}^{\infty} A_n^{\sigma a\chi b} G_n^{0,1,1,1}(z) = \mathcal{G}_{\sigma a\chi b}^{t,L}(z) - \mathcal{G}_{\sigma a\chi b}^{s,L}(z) + \sum_{n=2}^{\infty} A_n^{\sigma b\chi a} G_n^{0,1,1,1}(1-z) \,, \tag{5.100}$$

with $a, b \in \{+, -\}$. Here

$$A_n^{\sigma a\chi b} = \sum_{m:\Delta_m^{(0)}=n} (-1)^{J_m} c_{\sigma am}^{(1)} c_{\chi bm}^{(0)} \,, \tag{5.101}$$

and

$$\mathcal{G}_{\sigma a\chi b}^{s,L}(z) = \sum_{m\in\{\sigma,\chi\}} \left[c_{\sigma am}c_{\chi bm}\, G_{\Delta_m}^{\Delta_\sigma,\Delta_a,\Delta_\chi,\Delta_b}(z)\right]\Big|_{\sqrt{\delta}}$$
$$\mathcal{G}_{\sigma a\chi b}^{t,L}(z) = \sum_{m\in\{\sigma,\chi\}} \left[c_{\sigma bm}c_{\chi am}\, G_{\Delta_m}^{\Delta_\sigma,\Delta_b,\Delta_\chi,\Delta_a}(1-z)\right]\Big|_{\sqrt{\delta}} = \mathcal{G}_{\sigma b\chi a}^{s,L}(1-z) \,. \tag{5.102}$$

Again, we can use the subtracted functionals $\widetilde{\omega}_n^{0,1,1,1}$ of (5.95) to extract $A_n^{\sigma a\chi b}$ by applying them to (5.100)

$$A_n^{\sigma a\chi b} = \widetilde{\omega}_n^{0,1,1,1}[\mathcal{G}_{\sigma a\chi b}^{t,L}(z) - \mathcal{G}_{\sigma a\chi b}^{s,L}(z)] \,. \tag{5.103}$$

The answer is

$$A_n^{\sigma+\chi+} = -A_n^{\sigma-\chi-} = [(n-2)(n+1) - 4(-1)^n]\frac{(n-1)!(n-2)!}{2(2n-2)!} = (-1)^n A_n^{\sigma++\chi} \,,$$
$$A_n^{\sigma+\chi-} = -A_n^{\sigma-\chi+} = -[n(n-1) + 2]\frac{(n-1)!(n-2)!}{2(2n-2)!} = (-1)^n A_n^{\sigma+-\chi} \,. \tag{5.104}$$

The relation $A_n^{\sigma a\chi b} = (-1)^n A_n^{\sigma ab\chi}$ satisfied by these formulas is explained by observing that at $\delta = 0$ the $\chi \times \mathcal{O}_\pm$ OPE contains only double trace operators $\chi\overset{\leftrightarrow}{\partial}^n\mathcal{O}_\pm$ with $\Delta = 2 + n$ and parity $(-1)^n$.

## 5.4 Consistency checks

We can combine the previous results (5.82), (5.91) to obtain sums of $c^{(1)}_{\sigma am}c^{(1)}_{\sigma bm}$ over $\mathcal{P}_m$ with fixed $\Delta^{(0)}_m$ and $(-1)^{J_m}$. Indeed, we have

$$
\begin{aligned}
A^{\sigma ab\sigma}_{n,\text{even}} &:= \sum_{\substack{m:\Delta^{(0)}_m=n \\ (-1)^{J_m}=+1}} c^{(1)}_{\sigma am}c^{(1)}_{\sigma bm} = \frac{A^{\sigma ab\sigma}_n + A^{\sigma a\sigma b}_n}{2} \\
A^{\sigma ab\sigma}_{n,\text{odd}} &:= \sum_{\substack{m:\Delta^{(0)}_m=n \\ (-1)^{J_m}=-1}} c^{(1)}_{\sigma am}c^{(1)}_{\sigma bm} = \frac{A^{\sigma ab\sigma}_n - A^{\sigma a\sigma b}_n}{2} \,.
\end{aligned}
\tag{5.105}
$$

The resulting expressions need to pass a number of consistency checks. The first consistency check arises from the observation that $A^{\sigma++\sigma}_{n,\text{even}}$, $A^{\sigma--\sigma}_{n,\text{even}}$, $A^{\sigma++\sigma}_{n,\text{odd}}$ and $A^{\sigma--\sigma}_{n,\text{odd}}$ are sums of squares. Therefore, they must be non-negative. We can easily check that this is indeed the case for the above formulas.

The second consistency check is to note that for any $n \geq 2$ such that the space of $\mathbb{Z}_2$-odd and respectively parity even, odd is zero-dimensional, we must have respectively $A^{\sigma++\sigma}_{n,\text{even}} = A^{\sigma--\sigma}_{n,\text{even}} = A^{\sigma+-\sigma}_{n,\text{even}} = 0$, $A^{\sigma++\sigma}_{n,\text{odd}} = A^{\sigma--\sigma}_{n,\text{odd}} = A^{\sigma+-\sigma}_{n,\text{odd}} = 0$. We see from (4.16) that this is the case for odd parity with $n = 2, 4$ and never the case for even parity. Again, this agrees with the above formulas.

The third consistency check are the Cauchy-Schwartz inequalities

$$
(A^{\sigma+-\sigma}_{n,\text{even}})^2 \leq A^{\sigma++\sigma}_{n,\text{even}} A^{\sigma--\sigma}_{n,\text{even}} \,, \qquad (A^{\sigma+-\sigma}_{n,\text{odd}})^2 \leq A^{\sigma++\sigma}_{n,\text{odd}} A^{\sigma--\sigma}_{n,\text{odd}} \,.
\tag{5.106}
$$

Furthermore, whenever the relevant space of primaries is one-dimensional, the inequality must be saturated. From (4.16), we must have saturation for even parity with $n = 2, 3$ and odd parity with $n = 3, 6$. Again, it is not hard to check that all of these properties are satisfied by the above formulas.

We can carry out another consistency check by noting that the space of $\mathbb{Z}_2$-odd, parity-even primaries with $\Delta^{(0)} = 2$ is one-dimensional. At $\delta = 0$, it is spanned by $\widetilde{\rho} = \chi^2\hat{\sigma}_3/\sqrt{2}$. It follows from the field theory description at $\delta = 0$ that

$$
c^{(0)}_{\chi+\widetilde{\rho}} = c^{(0)}_{\chi-\widetilde{\rho}} = 1 \,.
\tag{5.107}
$$

This is consistent with (5.99), which furthermore predicts

$$
c^{(1)}_{\sigma+\widetilde{\rho}} = -1 \,, \quad c^{(1)}_{\sigma-\widetilde{\rho}} = 1 \,.
\tag{5.108}
$$

We must therefore have

$$
A^{\sigma++\sigma}_2 = A^{\sigma--\sigma}_2 = -A^{\sigma+-\sigma}_2 = 1 \,,
\tag{5.109}
$$

which indeed agrees with (5.82).

One more check follows from considering the space of $\mathbb{Z}_2$-odd primaries with $\Delta^{(0)} = 3$. For each parity, this space is one-dimensional (4.16). Let us denote the corresponding parity-even primary $\mu$, and parity-odd primary $\mu'$. From (5.82) and (5.91), we find

$$
c^{(1)}_{\sigma+\mu} = -c^{(1)}_{\sigma-\mu} = \pm\frac{1}{\sqrt{6}} \,, \quad c^{(1)}_{\sigma+\mu'} = c^{(1)}_{\sigma-\mu'} = \pm\frac{1}{\sqrt{6}} \,,
\tag{5.110}
$$

where the choice of sign on the RHS corresponds to the ambiguity in the definition of $\mu$, $\mu'$. It follows that we must have

$$
A^{\sigma++\chi}_3 = A^{\sigma-+\chi}_3 = \pm\frac{1}{\sqrt{6}}c^{(0)}_{\chi+\mu'} \,, \quad A^{\sigma+-\chi}_3 = A^{\sigma--\chi}_3 = \pm\frac{1}{\sqrt{6}}c^{(0)}_{\chi-\mu'} \,,
\tag{5.111}
$$

where the two signs on the RHS must be the same. This is consistent with (5.99), which furthermore predicts

$$
c^{(0)}_{\chi+\mu'} = -c^{(0)}_{\chi-\mu'} = \pm\sqrt{\frac{2}{3}} \,.
\tag{5.112}
$$

This result can be checked by a calculation in the free theory at $\delta = 0$.

## 5.5 Solution for the CFT data

Let us combine the constraints we have derived to determine the CFT data. The constraints consist of (5.65) together with (5.71), (5.80), and (5.97). They constitute 9 linear equations for the 12 unknowns

$$\Delta_\pm^{(2)}, \ c_{\sigma\chi\pm}^{(2)}, \ c_{\sigma\sigma\pm}^{(3)}, \ c_{\chi\chi\pm}^{(1)}, \ c_{+++}^{(1)}, \ c_{+--}^{(1)}, \ c_{+--}^{(1)}, \ c_{---}^{(1)}. \tag{5.113}$$

In order to find a unique solution, we will make use of the expected symmetry of the CFT data under the switch $\mathcal{O}_+ \leftrightarrow \mathcal{O}_-$, accompanied by $\sqrt{\delta} \mapsto -\sqrt{\delta}$. Note that this symmetry does leave the lower-order results in (5.65) invariant. To implement the symmetry at higher orders, let us impose the following 3 equations

$$c_{\chi\chi+}^{(1)} = -c_{\chi\chi-}^{(1)}, \quad c_{\sigma\chi+}^{(2)} = c_{\sigma\chi-}^{(2)}, \quad c_{\sigma\sigma+}^{(3)} = -c_{\sigma\sigma-}^{(3)}. \tag{5.114}$$

We then find a unique solution

$$\begin{aligned}
\Delta_\pm &= 1 \pm \sqrt{2}\sqrt{\delta} + \frac{\delta}{4} + O(\delta^{\frac{3}{2}}), \\
c_{\sigma\chi\pm} &= \frac{1}{\sqrt{2}} \mp \frac{\sqrt{\delta}}{16} - \frac{\delta}{256\sqrt{2}} + O(\delta^{\frac{3}{2}}), \\
c_{\sigma\sigma\pm} &= \pm \frac{\sqrt{\delta}}{\sqrt{2}} - \frac{15}{16}\delta \pm \left(\frac{543}{256} - \frac{\pi^2}{6}\right)\frac{\delta^{\frac{3}{2}}}{\sqrt{2}} + O(\delta^2), \\
c_{\chi\chi\pm} &= O(\delta), \\
c_{+++} &= \frac{3}{2} + \frac{39}{16\sqrt{2}}\sqrt{\delta} + O(\delta), \qquad c_{++-} = -\frac{1}{2} + \frac{\sqrt{\delta}}{16\sqrt{2}} + O(\delta), \\
c_{---} &= \frac{3}{2} - \frac{39}{16\sqrt{2}}\sqrt{\delta} + O(\delta), \qquad c_{+--} = -\frac{1}{2} - \frac{\sqrt{\delta}}{16\sqrt{2}} + O(\delta).
\end{aligned} \tag{5.115}$$

Reassuringly, the solution exhibits the symmetry also for $\Delta_\pm$ and $c_{\pm\pm\pm}$, although we only imposed it for $c_{\chi\chi\pm}$, $c_{\sigma\chi\pm}$, and $c_{\sigma\sigma\pm}$. As promised, the solution agrees with predictions of the RG method of Section 4.

## 5.6 Checking the OPE relation

As we discussed in Section 2.2, the OPE coefficients involving $\sigma$ and $\chi$ in the critical 1d LRI must satisfy certain relations. Specifically, for any four primaries $\phi_i$, $\phi_j$, $\phi_k$, $\phi_\ell$, we have [43] (see also appendix G)

$$\frac{c_{\sigma ij}c_{\chi k\ell}}{c_{\chi ij}c_{\sigma k\ell}} = \frac{\Gamma\left(\frac{\Delta_\sigma+\Delta_i-\Delta_j+a_{ij}}{2}\right)\Gamma\left(\frac{\Delta_\sigma-\Delta_i+\Delta_j+a_{ij}}{2}\right)\Gamma\left(\frac{1-\Delta_\sigma+\Delta_k-\Delta_\ell+a_{k\ell}}{2}\right)\Gamma\left(\frac{1-\Delta_\sigma-\Delta_k+\Delta_\ell+a_{k\ell}}{2}\right)}{\Gamma\left(\frac{1-\Delta_\sigma+\Delta_i-\Delta_j+a_{ij}}{2}\right)\Gamma\left(\frac{1-\Delta_\sigma-\Delta_i+\Delta_j+a_{ij}}{2}\right)\Gamma\left(\frac{\Delta_\sigma+\Delta_k-\Delta_\ell+a_{k\ell}}{2}\right)\Gamma\left(\frac{\Delta_\sigma-\Delta_k+\Delta_\ell+a_{k\ell}}{2}\right)}, \tag{5.116}$$

where $a_{ij} = [1 - (-1)^{J_i+J_j}]/2 = J_i + J_j \mod 2$.

Let us test whether these relations are satisfied by the CFT data (5.115) derived using the bootstrap. There are three choices for $(i, j, k, \ell) \in \{\sigma, \chi, \mathcal{O}_+, \mathcal{O}_-\}^4$ giving rise to independent constraints:

$$(i, j, k, \ell) = (\sigma, \mathcal{O}_+, \sigma, \mathcal{O}_-), \quad (i, j, k, \ell) = (\sigma, \mathcal{O}_+, \chi, \mathcal{O}_+), \quad (i, j, k, \ell) = (\sigma, \mathcal{O}_-, \chi, \mathcal{O}_-). \tag{5.117}$$

Considering first the case $(i, j, k, \ell) = (\sigma, \mathcal{O}_+, \sigma, \mathcal{O}_-)$, the bootstrap solution (5.115) predicts

$$\frac{c_{\sigma\sigma+}c_{\sigma\chi-}}{c_{\sigma\chi+}c_{\sigma\sigma-}} = -1 + \frac{7\sqrt{\delta}}{2\sqrt{2}} - \frac{49}{16}\delta + O(\delta^{\frac{3}{2}}). \tag{5.118}$$

At the same time, the RHS of (5.116) evaluates to

$$-1 + \frac{7\sqrt{\delta}}{2\sqrt{2}} - \left[\frac{49}{16} + \frac{\Delta_+^{(3)} + \Delta_-^{(3)}}{\sqrt{2}}\right]\delta + O(\delta^{\frac{3}{2}}). \tag{5.119}$$

We see that (5.116) holds automatically up to $O(\delta)$ since $\Delta_+^{(3)} + \Delta_-^{(3)} = 0$ by virtue of the $\sqrt{\delta} \mapsto -\sqrt{\delta}$ symmetry. This agreement is a highly nontrivial consistency check of our results.

Moving on to the cases $(i, j, k, \ell) = (\sigma, \mathcal{O}_\pm, \chi, \mathcal{O}_\pm)$, we find that the corresponding relation is satisfied by the bootstrap solution (5.115) up to $O(\delta)$. When we impose the OPE relation up to $O(\delta^2)$, we obtain the new result

$$c_{\chi\chi\pm} = -\frac{\pi^2}{2}\delta \mp \frac{11\pi^2}{16\sqrt{2}}\delta^{\frac{3}{2}} + O(\delta^2). \tag{5.120}$$

## 5.7 Four-point functions

Let us conclude this section by collecting the analytic bootstrap results for the four-point functions that we studied. We obtained them by combining the OPE contribution of 'light' operators $\mathbb{1}, \sigma, \chi, \mathcal{O}_+, \mathcal{O}_-$ with that of 'heavy' operators, using the closed formulas for $A_n^{ijk\ell}$. We begin with all four-point functions involving only $\sigma$ and $\chi$:

$$\begin{aligned}
\mathcal{G}_{\sigma\sigma\sigma\sigma}(z) = {}& 1 - [\log z + \log(1-z)]\delta \\
&+ \Big[-6\mathrm{Li}_3\left(\tfrac{z}{z-1}\right) + 4\log(\tfrac{1-z}{z})\mathrm{Li}_2(z) + \log^3(1-z) + \tfrac{1}{2}\log^2(z) + \tfrac{1}{2}\log^2(1-z) \\
&\quad - \log^2(z)\log(1-z) + \tfrac{1}{3}\pi^2\log(1-z) + \tfrac{1}{2}\log(z)\log(1-z)\Big]\delta^2 + O(\delta^3),
\end{aligned} \tag{5.121}$$

$$\mathcal{G}_{\sigma\sigma\sigma\chi}(z) = \sqrt{2}z^{-1}\left[z\log z + (1-z)\log(1-z)\right]\delta + O(\delta^2),$$
$$\mathcal{G}_{\sigma\sigma\chi\chi}(z) = z^{-2} + z^{-2}[\log z + (\log(1-z))^2]\delta + O(\delta^2),$$
$$\mathcal{G}_{\sigma\chi\sigma\chi}(z) = 1 + [\log z - \log(1-z)]^2\delta + O(\delta^2),$$
$$\mathcal{G}_{\sigma\chi\chi\chi}(z) = O(\delta).$$

Next, we list the four-point functions involving only $\sigma$, $\mathcal{O}_+$, and $\mathcal{O}_-$:

$$\begin{aligned}
\mathcal{G}_{\sigma\sigma\pm\pm}(z) = {}& z^{-2} \mp \sqrt{2}z^{-2}\left[2\log z + \log(1-z)\right]\sqrt{\delta} \\
&+ z^{-2}\Big[-\mathrm{Li}_2(z) + \log^2(1-z) + 4\log^2(z) + 3\log(z)\log(1-z) \\
&\quad - \tfrac{1}{2}\log(z) - \tfrac{3}{4}\log(1-z) + \tfrac{z}{2(1-z)}\Big]\delta + O(\delta^{\frac{3}{2}}),
\end{aligned}$$

$$\mathcal{G}_{\sigma\sigma\pm\mp}(z) = z^{-2}\left[\mathrm{Li}_2(z) + \log(1-z)\log(z) - \tfrac{1}{2}\log(1-z) + \tfrac{z}{2(z-1)}\right]\delta + O(\delta^{\frac{3}{2}}),$$

$$\mathcal{G}_{\sigma\pm\sigma\pm}(z) = \mp\tfrac{1}{4\sqrt{2}}\sqrt{\delta} + \left[-\tfrac{1}{4}\log(z(1-z)) + \tfrac{z}{2(1-z)} + \tfrac{1}{2z} + \tfrac{\pi^2}{6}\right]\delta + O(\delta^{\frac{3}{2}}),$$

$$\begin{aligned}
\mathcal{G}_{\sigma\pm\sigma\mp}(z) = {}& 1 \pm \sqrt{2}[\log z - \log(1-z)]\sqrt{\delta} \\
&+ \left[\left(\log\left(\tfrac{z}{1-z}\right)\right)^2 - \tfrac{3}{4}\log(z(1-z)) + \tfrac{1}{2(z-1)z} - \tfrac{\pi^2}{6} + \tfrac{31}{64}\right]\delta + O(\delta^{\frac{3}{2}}).
\end{aligned} \tag{5.122}$$

We conclude with all four-point functions involving $\sigma$, $\chi$, $\mathcal{O}_+$ and $\mathcal{O}_-$:

$$\mathcal{G}_{\sigma\chi\pm\pm}(z) = \frac{1}{\sqrt{2}z(1-z)} \pm \frac{4z^2 - 9z + 8(z-1)\log(1-z) + 9}{8(z-1)^2 z}\sqrt{\delta} + O(\delta)\,,$$

$$\mathcal{G}_{\sigma\chi\pm\mp}(z) = \frac{1}{\sqrt{2}z(z-1)} \pm \frac{(z-2)z - 2(z-1)\log(1-z)}{2(z-1)^2 z}\sqrt{\delta} + O(\delta)\,,$$

$$\mathcal{G}_{\sigma\pm\chi\pm}(z) = \frac{1}{\sqrt{2}z(z-1)} \pm \frac{9(z-1)z - 8(z-1)z\log[z(1-z)] + 4}{8(z-1)^2 z^2}\sqrt{\delta} + O(\delta)\,,$$

$$\mathcal{G}_{\sigma\pm\chi\mp}(z) = \frac{1}{\sqrt{2}z(1-z)} \pm \frac{2z(1-z)\log\left(\frac{z}{1-z}\right) - 2z + 1}{2(z-1)^2 z^2}\sqrt{\delta} + O(\delta)\,.$$

(5.123)

# 6 Conclusions and future directions

In this paper, we elaborated on the weakly coupled description of the short-range crossover for the 1d long-range Ising model universality class, presented in [21].

For all $s \in (1/2, 1)$, the 1d LRI has an interacting IR fixed point. While a weakly coupled description near $s = 1/2$ is provided by a GFF with $\varphi^4$ interaction, this model becomes strongly coupled near $s = 1$ and an alternative, weakly coupled field-theoretic description remained unknown for a long time. We have filled this gap, introducing the model in equation (3.9), which generalizes the Kondo model by giving a non-vanishing canonical dimension $-\delta/2$, with $0 \le \delta < 1$, to the scalar field $\phi$. We claim that the $U(1)$-singlet sector of its weakly interacting fixed point describes the 1d LRI CFT at $s = 1 - \delta$.

The idea behind the model is based on the physical description of the 1d LRI provided by Anderson and Yuval [45, 46, 47, 22] for $s = 1$, and extended by Kosterlitz [23] to $s < 1$. They recognized that at low temperatures, the 1d LRI close to $s = 1$ could be written as a dilute gas of kinks and antikinks, with the (anti)kinks describing domain walls where the spins of the LRI flip. At $s = 1$, this Coulomb gas admits various other descriptions, such as a perturbative expansion of the Kondo model, that in its bosonized version is similar to the boundary sine-Gordon model, but with an extra single spin-1/2 degree of freedom implementing the constraint of alternating kinks and antikinks. The latter description is the starting point for our model (3.9), that extends it to $s < 1$.

There is a loose analogy between our construction and the corresponding higher-dimensional model of [9, 10]. In 1d, the zero-temperature SRI model has a single qubit degree of freedom, with the spin field represented by the Pauli matrix $\hat{\sigma}_3$, see Section 2.4. We can thus view the $\mathcal{O}_h$ interaction in our model (3.9) as the analog of the $\sigma\chi$ interaction of [9, 10]. Indeed our notation for the operator $\chi$ was not accidental, as also in our case it represents the shadow of the spin field in the LRI CFT. The main difference in 1d is the need to introduce operators that create kinks and antikinks, otherwise the model would be trivial, and in particular it would not lead to the LRI CFT at $s < 1$, or to the logarithmic corrections at $s = 1$. As the (anti)kink-generating operators cannot be written in a local way in terms of $\chi$, we are led to introducing the ancestor field $\phi$, such that $\chi \sim \partial\phi$. In terms of $\phi$ and of the $\mathfrak{sl}_2$-triplet $\{\hat{\sigma}_3, \hat{\sigma}_+, \hat{\sigma}_-\}$, the (anti)kink-generating operators are the vertex operators $\hat{\sigma}_\pm e^{\pm ib_0\phi}$, which we combined in the $\mathcal{O}_g$ interaction in (3.9).

Our model (3.9) can appear to be nonstandard for at least three reasons: $(i)$ the unperturbed theory is a compact GFF with negative scaling dimension; $(ii)$ the perturbation $\mathcal{O}_g$ is not a scaling operator of the unperturbed theory; $(iii)$ the perturbations involve matrix degrees of freedom and a path-ordering. However, as we have argued in the main text, and as corroborated by the overall self-consistency of our results, none of these aspects should cause headaches.

Concerning the first point, 1d GFF with negative scaling dimension is standard in the mathematical literature, corresponding to the fractional Brownian motion [59]. Indeed, as reviewed in Appendix A, this class of theories arises from a probability measure on the space of continuous functions, defined modulo a

global additive constant. In other words, these are shift-symmetric Gaussian theories, their well-defined local observables have positive scaling dimension, and their target space can be consistently compactified.

Regarding the fact that $\mathcal{O}_g$ is not a scaling operator of the unperturbed theory, we make the following remarks. Firstly, even in standard massive perturbation theory, operators such as $\varphi^4$ are not scaling operators, they only become such in the UV limit, when the mass is neglected. Similarly, one could consider the UV limit of our model, in which case $b_0 \to 0$. In this limit, $\mathcal{O}_g$ reduces to a scaling operator of vanishing dimension. Secondly, in that same limit, one could expand $\mathcal{O}_g$ in an infinite series of relevant scaling operators. Since the $U(1)$ symmetry protects their relative coefficients, they still only give rise to a single coupling. Unfortunately, we have not obtained a proof of renormalizability to all orders in this framework. Having said that, we found no evidence that counterterms of irrelevant operators should be added, up to second order of the perturbative expansion. This also leads to our third remark: even if the model (3.9) turned out to not be UV complete (which we doubt), this would not affect its usefulness for the construction of the IR CFT. In fact, we are not proposing this model as a new fundamental theory, but rather as a tool to construct an IR CFT via standard RG computations. In the spirit of Wilsonian RG, the presence of irrelevant operators in the bare action has no effect on low-energy physics: the modern view is actually that all QFTs are only effective field theories. Lastly, the RG computations of Section 4, and their agreement with the bootstrap computations of Section 5, corroborate our point of view, showing that the model provides a perturbative computational framework for extracting CFT data of the 1d LRI near $s = 1$.

As for the fact that the interactions involve matrix degrees of freedom, we notice that similar models appear naturally in condensed-matter systems, like the Kondo model itself, and more generally for systems with localized impurities, see for example [63, 64]. Moreover, the $2 \times 2$ matrices and the associated path ordering could be traded for a path integral using a complex bosonic spinor [65, 63, 64], as reviewed in Appendix C.1.

An advantage of our description is the possibility to easily compute, in a systematic way, a host of CFT data, consisting of operator scaling dimensions and OPE coefficients, perturbatively in the couplings $g, h$. In Section 4 we have done this for the first few low-lying operators, including the two marginally relevant operators $\mathcal{O}_\pm$, which are linear combinations of $\mathcal{O}_g, \mathcal{O}_h$ appearing in (3.9). Since our model describes a family of 1d CFTs, we compared the perturbative results against results obtained by the analytic conformal bootstrap in Section 5. Here, the only input are the symmetries of the theory, and the spectrum at $s = 1$. We found complete agreement with the perturbative RG computations, and extended them to higher orders and other observables. This includes a large class of four-point correlation functions of light fields, which themselves encode an infinite set of CFT data.

The agreement solidifies our claim that the fixed point of (3.9) describes the 1d LRI near $s = 1$. The results for scaling dimensions and OPE coefficients are summarized in Table 2, and those for four-point functions in Section 5.7.

The nonlocality of the 1d LRI implies several nonperturbative properties of the CFT data. In particular, certain ratios of OPE coefficients involving $\sigma$ and $\chi$ are given by ratios of gamma functions of the scaling dimensions, see Section 2.2.3. These OPE relations have been used as important constraints in bootstrap studies of the long-range Ising model [43, 14][26]. However, we did not use them as an input in our analytic bootstrap analysis. Instead, the OPE relations are satisfied automatically (and in a rather nontrivial manner) by the bootstrap solution, which serves as another important consistency check. See Section 5.6 for more details.

We conclude with several open questions, left for future work. It would be interesting to explore other possible formulations of our model. For example, it might be useful to find a formulation reinterpreting it as a boundary theory of a massive scalar in $AdS_2$. By a standard AdS/CFT construction [79], the GFF

---

[26]Similar relations can be derived for other models describing interacting defects, such as boundaries, in a free bulk CFT [77, 78]

| Observable | Value |
|---|---|
| $\Delta_\sigma$ | $\frac{\delta}{2}$ (exact) |
| $\Delta_\chi$ | $1 - \frac{\delta}{2}$ (exact) |
| $\Delta_\pm$ | $1 \pm \sqrt{2\delta} + \frac{\delta}{4} + O(\delta^{3/2})$ |
| | |
| $c_{\sigma\sigma\pm}$ | $\pm\sqrt{\frac{\delta}{2}} - \frac{15\delta}{16} \pm \left(\frac{543}{256} - \frac{\pi^2}{6}\right)\frac{\delta^{3/2}}{\sqrt{2}} + O(\delta^2)$ |
| $c_{\sigma\chi\pm}$ | $\frac{1}{\sqrt{2}} \mp \frac{\sqrt{\delta}}{16} - \frac{\delta}{256\sqrt{2}} + O(\delta^{3/2})$ |
| $c_{\chi\chi\pm}$ | $-\frac{\pi^2\delta}{2} \mp \frac{11\pi^2\delta^{3/2}}{16\sqrt{2}} + O(\delta^2)$ |
| $c_{\pm\pm\pm}$ | $\frac{3}{2} \pm \frac{39\sqrt{\delta}}{16\sqrt{2}} + O(\delta)$ |
| $c_{+\pm-}$ | $-\frac{1}{2} \pm \frac{\sqrt{\delta}}{16\sqrt{2}} + O(\delta)$ |

Table 2: Summary of results for scaling dimensions and OPE coefficients from perturbative computations and analytic bootstrap.

$\phi$ could be viewed as the boundary theory of a bulk Klein-Gordon field with squared mass $m^2 = \frac{\delta}{2}(1 + \frac{\delta}{2})$ and negative-branch boundary condition $\Delta_- = \frac{1}{2} - \sqrt{\frac{1}{4} + m^2}$, which is admissible [80] for $m^2 < 3/4$, i.e. $\delta < 1$, the upper bound corresponding to the value at which the GFF reduces to the standard 1d free scalar. It would be nice to construct the 1d LRI as a suitable boundary condition for this AdS theory.

It would also be interesting to study the interpolation of CFT data in the range $1/2 < s < 1$, with both $s - 1/2$ and $1 - s$ not infinitesimal. The perturbative results that we obtained here near $s = 1$, together with perturbative results for the $\varphi^4$ description near $s = 1/2$ (derived at three loop for any dimension in [26, 14]), provide useful benchmark points for any nonperturbative study aiming at addressing such question. They could be also used to attempt an interpolation by means of resummations, but this approach is hampered by the small order at which the perturbative series are currently computed (see nevertheless [27] for some progress in three dimensions). Monte Carlo simulations offer another approach to study the nonperturbative interpolation between the $\varphi^4$ and the near short-range regimes. Only a few results for the 1d LRI are available [81, 82, 83, 84], and it would be desirable to improve them to enable a comparison with our results. Another approach that could be adapted to this problem is provided by the functional renormalization group (see e.g. [85]), already applied to long-range models for example in [86, 87].

Since the critical 1d LRI corresponds to a family of 1d CFTs, one could use the numerical conformal bootstrap [88] (see [89, 90] for reviews) to identify the 1d LRI and bound its CFT data, as was done for the higher-dimensional LRI in [43, 14]. It is also an interesting target for the multipoint numerical bootstrap, recently developed for one-dimensional CFTs [91, 92].

The Ising model, either in its short-range or long-range version, is a cornerstone of statistical physics, and thus the problem solved in [21] and this paper was a particularly pressing one. Similar questions can be asked for a variety of other 1d long-range models. For the long-range $O(N)$ model, a perturbative description near the short-range end was already provided by Kosterlitz in [23], in the form of a long-range nonlinear $O(N)$ sigma model, see also [93]. For other models, such as the long-range Blume-Capel (or tricritical Ising) model or the long-range Potts model [94, 95, 96], the situation is much more open. Same holds true for their particularly important limits, such as the self-avoiding walks [97] and percolation [98], or for disordered versions of these models [99, 100].

We hope to come back to these questions in the near future.

**Acknowledgments.** We thank C. Behan, N. Bobev, A. Gimenez-Grau, K. Ghosh, M. Paulos, L. Rastelli, S. Rychkov, J. Vošmera and Z. Zheng for discussions. EL is supported by the European Union (ERC,

QFT.zip project, Grant Agreement no. 101040260). PvV is funded by the European Union (ERC, FUN-BOOTS, project number 101043588). Views and opinions expressed are however those of the authors only and do not necessarily reflect those of the European Union or the European Research Council Executive Agency. Neither the European Union nor the granting authority can be held responsible for them.

# A   Generalized free field on the line

A $d$-dimensional generalized free field theory (GFF) is the quantum field theory of a single scalar $\varphi$, whose only nonvanishing connected $n$-point function is the two-point function $\langle\varphi(x)\varphi(y)\rangle \equiv C(x - y)$, assumed to be invariant under rotations and translations. All other (not connected) $n$-point functions are obtained as sums over Wick contractions. In other words, a GFF is a centered Gaussian measure, and as such it can be defined without introducing any action functional, see e.g. [61, 101].

Assuming also reflection positivity, GFFs provide the simplest examples of QFTs satisfying the standard axioms [102, 61]. If furthermore we take $C(x) \propto 1/|x|^{2\Delta_\varphi}$, with $\Delta_\varphi$ being the scaling dimension of $\varphi$, then the GFF is also conformally invariant, it is known in the mathematical literature as *fractional Gaussian field* [57], and for $\Delta_\varphi = (d-2)/2$ it reduces to the standard massless free scalar.

We restrict now to a conformal GFF in $d = 1$. For $\Delta_\varphi > 0$, it provides the simplest example of a 1d CFT as defined in Section 2.2.1. For $\Delta_\varphi < 0$, its definition and conformal properties are slightly more subtle, but it is also a well studied random field, associated to the *fractional Brownian motion* with so-called Hurst parameter $H = -\Delta_\varphi$, see for example [57, 58, 103]. The intermediate case $\Delta_\varphi = 0$ corresponds to the log-correlated Gaussian field [104], which shares many technical aspects with the $\Delta_\varphi < 0$ case, and which in $d = 1$ coincides with the usual 2d free field theory restricted to a line, as we will recall below.

We review here in simplified terms the construction of the GFF probability measure and the associated action functional, referring for more details to the original literature from which we draw our presentation [61, 57, 103, 105].

The main idea behind the rigorous construction of a GFF is to use the Bochner-Minlos theorem. Let $\mathcal{S}(\mathbb{R})$ be the Schwartz space of test functions $f : \mathbb{R} \to \mathbb{R}$ whose derivatives of all orders exist and decay faster than any polynomial at infinity, and let $Z[f]$ be a given complex-valued functional on such space. Let us introduce also the space $\mathcal{S}'(\mathbb{R})$ of tempered distributions, i.e. the space of continuous linear functionals on $\mathcal{S}(\mathbb{R})$, so that if $\varphi \in \mathcal{S}'(\mathbb{R})$ and $f \in \mathcal{S}(\mathbb{R})$, then the canonical pairing $\varphi[f] = (\varphi, f) \in \mathbb{R}$ exists, and we formally write

$$\varphi[f] = \int_{\mathbb{R}} \mathrm{d}x\, \varphi(x) f(x)\,. \tag{A.1}$$

One nice feature of $\mathcal{S}(\mathbb{R})$ and $\mathcal{S}'(\mathbb{R})$ is that the Fourier transform and its inverse act as linear endomorphisms on these spaces.

The Bochner-Minlos theorem states that there exists a unique probability measure $\mu$ on $\mathcal{S}'(\mathbb{R})$, such that $Z[f]$ is the characteristic function (or generating functional) of $\mu$, i.e. its Fourier transform

$$Z[f] = \int_{\mathcal{S}'(\mathbb{R})} d\mu(\varphi)\, e^{\mathrm{i}\varphi[f]}\,, \tag{A.2}$$

if and only if $Z[f]$ is continuous, $Z[0] = 1$, and $Z[f]$ is positive definite, i.e.

$$\sum_{j,k=1}^{n} z_j \bar{z}_k Z[f_j - f_k] \geq 0\,, \qquad \forall f_1, \dots, f_n \in \mathcal{S}(\mathbb{R}),\ \text{and}\ z_1, \dots, z_n \in \mathbb{C}\,. \tag{A.3}$$

A GFF is then defined by the assignment of a characteristic function of the form

$$Z_{\mathrm{GFF}}[f] = e^{-\frac{1}{2}\langle\varphi[f]\varphi[f]\rangle} = e^{-\frac{1}{2}\int \mathrm{d}x\mathrm{d}y\, C(x-y)f(x)f(y)}\,. \tag{A.4}$$

If the exponent can be cast in the form of an inner product in some space, then it is straightforward to show that $Z_{\mathrm{GFF}}[f]$ is positive (proposition 2.4 of [57]) and the Bochner-Minlos theorem applies. In following this argument, it is important to treat separately the cases of positive and negative scaling dimension, as we will now explain.

This construction might seem beyond the scope of our paper: after all, except for the choice of covariance, $Z_{\mathrm{GFF}}[f]$ is the standard generating functional for Gaussian correlators, familiar to any field theorist. And indeed at the end things are essentially as straightforward as they seem, if $0 < \Delta_\varphi < 1/2$. However, outside such range things become more involved, and following the steps of a rigorous construction leads to important insights.

Assume that $C(x)$ is normalized in such a way that in momentum space it reads

$$\tilde{C}(p) = \frac{1}{|p|^{1-2\Delta_\varphi}} \ . \tag{A.5}$$

In other words, the covariance is precisely defined as the fractional Laplacian (e.g. [106]) of power $-\zeta = \Delta_\varphi - 1/2$. One would like to apply the Bochner-Minlos theorem with the characteristic functional being given as above, with the choice

$$\langle \varphi[f]\varphi[f] \rangle = ||f||^2_{H^{\Delta_\varphi - 1/2}(\mathbb{R})} \equiv \int \frac{\mathrm{d}p}{2\pi} \, \tilde{f}(-p) \, |p|^{2\Delta_\varphi - 1} \, \tilde{f}(p) \ , \tag{A.6}$$

where $||f||_{H^{-\zeta}(\mathbb{R})}$ is by definition the norm on the Sobolev space $H^{-\zeta}(\mathbb{R})$, i.e. the Hilbert space completion of the set of Schwartz functions having finite norm. For $0 < \Delta_\varphi < 1/2$, i.e. $0 < \zeta < 1/2$, the norm (A.6) is finite for all Schwartz functions, hence the existence of the GFF as a probability measure is established.[27] On the contrary, for $\Delta_\varphi < 0$, i.e. $\zeta > 1/2$, the singularity at $p = 0$ renders the norm divergent, unless $\tilde{f}^{(n)}(0) = 0$ for $n \leq \lfloor -\Delta_\varphi \rfloor$. Therefore, in the latter case one introduces the subspace $\mathcal{S}_r(\mathbb{R})$ of Schwartz space spanned by test functions that in momentum space vanish at the origin, together with all the derivatives of order less or equal to the nonnegative integer $r$ (equivalently, such that in position space $\int \mathrm{d}x \, \mathcal{P}(x)f(x) = 0$ for all polynomials $\mathcal{P}(x)$ of degree $r$). The norm (A.6) is then finite in the restricted space of test functions $\mathcal{S}_{\lfloor -\Delta_\varphi \rfloor}(\mathbb{R})$, and we can establish the existence of the corresponding GFF measure on its dual space $\mathcal{S}'_{\lfloor -\Delta_\varphi \rfloor}(\mathbb{R})$. We refer to [57] for more details.[28]

In the remainder of this appendix we review some other aspects of the GFF, distinguishing the cases of positive and negative scaling dimensions, and introducing some of the notation used in the main body of the paper.

## A.1 GFF with positive scaling dimension

As remarked above, the GFF can be introduced without the need of an action functional. Nevertheless, it is convenient to write an explicit action, as for example this simplifies the derivation of Schwinger-Dyson equations. Moreover, this will make the nonlocal nature of the GFF explicit.

For positive scaling dimension, we write $\Delta_\varphi = (1-s)/2$, with $s < 1$, as in Section 2, which corresponds to taking $\zeta = s/2$. In path integral language, we can view the 1d GFF as a functional integral over $\varphi$, and write for example[29]

$$C(x) \equiv \langle \varphi(x)\varphi(0) \rangle \equiv \int [d\varphi] e^{-S_{\mathrm{GFF}}[\varphi]} \varphi(x)\varphi(0) \ . \tag{A.7}$$

The action $S_{\mathrm{GFF}}[\varphi]$ is most naturally defined in momentum space. We normalize it as

$$S_{\mathrm{GFF}}[\varphi] = \frac{1}{2} \int_{-\infty}^{+\infty} \frac{\mathrm{d}p}{2\pi} \, \tilde{\varphi}(-p)|p|^s \tilde{\varphi}(p) \ , \tag{A.8}$$

---

[27]The upper bound $\Delta_\varphi < 1/2$ comes from demanding that the norm is finite also in position space.

[28]Notice that in [57] the parameter $s$ corresponds to half our $s$, i.e. it coincides with the $\zeta$ introduced above (the latter being a notation used for example also in [25, 26]).

[29]The normalization of the Gaussian is implicit in the functional measure $[d\varphi]$, i.e. $\int [d\varphi] e^{-S_{\mathrm{GFF}}[\varphi]} = 1$.

so that the covariance takes the precise form (A.5), as can be seen starting from (A.2) with $d\mu(\varphi) = [d\varphi]e^{-S_{\mathrm{GFF}}[\varphi]}$, and using the translation invariance of $[d\varphi]$ to obtain (A.4) with norm (A.6).

The appearance of $|p|^s$ in the action makes evident its definition in terms of fractional Laplacian. Indeed, among several equivalent definitions [106], the easiest definition of the fractional Laplacian is in Fourier space, where $(-\partial_x^2)^{s/2}\varphi(x)$ is defined as the multiplication operator $|p|^s\tilde{\varphi}(p)$. Going to position space one finds a representation as a hypersingular integral operator:

$$(-\partial^2)^{s/2}\varphi(x) = \lim_{r\to 0} c_s \int_{|x-x'|>r} dx' \frac{\varphi(x) - \varphi(x')}{|x - x'|^{1+s}} , \tag{A.9}$$

where

$$c_s = -\frac{2^s \Gamma(\frac{1+s}{2})}{\pi^{1/2}\Gamma(-s/2)} = \Gamma(s+1)\sin(\frac{\pi s}{2})/\pi , \tag{A.10}$$

This can be derived by first writing

$$(p^2)^{s/2} = \frac{1}{\Gamma(-s/2)} \int_0^{+\infty} dt \frac{e^{-tp^2} - 1}{t^{1+s/2}} ,$$

whose validity is trivially checked by rescaling $t \to t/p^2$ and recognizing that the integral reduces to $|p|^s$ times the Cauchy-Saalschütz representation of $\Gamma(-s/2)$ for $0 < s < 2$. The singular integral representation is then found by going back to position space and exchanging the order of integration [105].

Multiplying (A.9) by $\frac{1}{2}\varphi(x)$ and integrating over $x$, we arrive at the action written in position space:

$$S_{\mathrm{GFF}}[\varphi] = \frac{c_s}{4} \int_{-\infty}^{+\infty} dx_1 dx_2 \frac{(\varphi(x_1) - \varphi(x_2))^2}{|x_1 - x_2|^{1+s}} , \tag{A.11}$$

Written as in (A.11), the action is manifestly positive for $0 < s < 2$, and the integral is regular at $x_1 \sim x_2$.[30]

The covariance in position space is of course the Fourier transform of $1/|p|^s$, which reads

$$C(x) = \frac{-c_{-s}}{|x|^{1-s}} , \tag{A.12}$$

and is a well-defined tempered distribution for $0 < s < 1$. Notice that the restriction to positive $s$, i.e. the upper bound $\Delta_\varphi < 1/2$, is also understood from the lattice point of view, as it is known that a positive $s$ is needed for the existence of the thermodynamic limit [107]. The restriction to $s < 1$ is instead needed for the positivity of $\Delta_\varphi$, without which the Schwartz space needs to be restricted, as explained above, and as further elaborated in the next subsection.

## A.2 GFF with negative scaling dimension

We now consider the case $-1 < \Delta_\varphi < 0$, relevant to the construction of our model in Section 3. In this case, adopting a notation consistent with Section 3, we rename the fundamental field as $\phi$, we denote the expectation value in this GFF theory by $\langle \cdot \rangle_0$, and we set $\Delta_\phi = -\delta/2$, i.e. $2\zeta = 2 - s = 1 + \delta$.

As explained above, in this range of scaling dimensions, the 1d GFF can be rigorously contructed as a probability measure on the space of distributions in $\mathcal{S}_0'(\mathbb{R})$, the dual space of $\mathcal{S}_0(\mathbb{R})$, the latter being the space of Schwartz functions satisfying the constraint $\int dx f(x) = 0$. Therefore, the covariance kernel is defined only up to an additive constant. Indeed, the covariance kernel $C(x)$ is defined as

$$\langle \phi[f_1]\phi[f_2] \rangle_0 = \int dx\, dy\, C(x - y) f_1(x) f_2(y) , \tag{A.13}$$

---

[30]Sometimes the action is written as proportional to $\int dx_1 dx_2 \varphi(x_1)\varphi(x_2)/|x_1 - x_2|^{1+s}$. This expression is clearly singular at $x_1 \sim x_2$ due to the kernel diverging with a power greater than one. It could be defined by analytic continuation from $s < 0$, but that would make properties like positivity and shift invariance of the action less transparent.

where $f_1, f_2 \in \mathcal{S}_0(\mathbb{R})$. As a consequence, if a function $C(x)$ satisfies (A.13), so does $C(x) + c$, for some constant $c$. Notice, that by such definition, we also have

$$\langle \phi[f_1]\phi[f_2]\rangle_0 = \int \mathrm{d}x\,\mathrm{d}y \,\langle \phi(x)\phi(y)\rangle_0 f_1(x) f_2(y) = -\frac{1}{2} \int \mathrm{d}x\,\mathrm{d}y \,\langle (\phi(x) - \phi(y))^2\rangle_0 f_1(x) f_2(y) \,. \tag{A.14}$$

However, one should keep in mind that this equality only holds for $f_1, f_2 \in \mathcal{S}_0(\mathbb{R})$. In particular, while $\langle \phi(x)\phi(y)\rangle_0$ is defined only up to a constant, $\langle (\phi(x) - \phi(y))^2\rangle_0$ is a well-defined quantity, the variance of the random variable $\phi(x) - \phi(y)$. The latter is indeed the stationary-increment form of the fractional Brownian motion with Hurst parameter $H = -\Delta_\phi$. In field theoretic applications we are rather interested in local observables, hence the appropriate well-defined random variables are derivatives of $\phi$, and their products: since for example $\langle \partial\phi(x)\partial\phi(y)\rangle_0 = \partial_x\partial_y\langle \phi(x)\phi(y)\rangle_0$, it is clear that no additive constant ambiguity survives.

The GFF theory has a similar action as before

$$S_0[\phi] = \frac{\mathcal{N}_\delta}{4} \int_{-\infty}^{+\infty} \mathrm{d}x_1\mathrm{d}x_2 \frac{(\phi(x_1) - \phi(x_2))^2}{|x_1 - x_2|^{2+\delta}} \,. \tag{A.15}$$

We can also (formally) express the action as $S_0[\phi] = \frac{1}{2}\phi \cdot C^{-1} \cdot \phi$, with obvious dot notation, and $C^{-1} \cdot \phi$ to be interpreted as (proportional to) the fractional Laplacian (equation (A.9) with $s \to 2 - s$).

For convenience, we will choose the normalization and the additive constant so that the covariance has our desired limit for $\delta \to 0$:

$$C(x) = -\frac{2}{\delta}\left(|x|^\delta - \kappa^{-\delta}\right) = -2\log(\kappa|x|) + O(\delta) \,. \tag{A.16}$$

This normalization corresponds to choosing $\mathcal{N}_\delta = \frac{\delta}{2}c_{1+\delta}c_{-1-\delta}$ in the action.

**GFF as a boundary theory.** At $\delta = 0$, the 1d GFF has a natural interpretation as a boundary theory. Consider the half plane $\Sigma \simeq \mathbb{R} \times \mathbb{R}^+$, where we use coordinates $\mathbf{x} \equiv (x, y)$ such that the boundary is located at $y = 0$. On $\Sigma$, we define the free theory of a bosonic bulk field $\Phi(\mathbf{x})$, with action normalized as

$$S_0[\Phi] = \frac{1}{4\pi} \int_\Sigma \mathrm{d}^2\mathbf{x}\,\partial_\mu\Phi\partial^\mu\Phi \,, \tag{A.17}$$

and with Neumann boundary conditions, i.e.

$$\partial_y\Phi(x, y)|_{y=0} = 0\,, \quad x \in \mathbb{R}\,. \tag{A.18}$$

In other words, the restriction of the bulk field $\Phi(x, y)$ to the boundary

$$\Phi(x, 0) = \phi(x)\,, \quad x \in \mathbb{R}\,, \tag{A.19}$$

is a dynamical field with propagator[31]

$$C(x) \equiv \langle \phi(x)\phi(0)\rangle_0 = -2\log(\kappa|x|) \,. \tag{A.20}$$

---

[31]If we denote $C(\mathbf{x}) = -\log|\mathbf{x}|$ the propagator on $\mathbb{R}^2$, such that $-\frac{1}{2\pi}\partial_\mu\partial^\mu C(\mathbf{x}) = \delta^{(2)}(\mathbf{x})$, then the propagator on $\mathbb{R} \times \mathbb{R}^+$ with Neumann boundary conditions at $y = 0$ is

$$C(\mathbf{x}_1 - \mathbf{x}_2) + C(\mathbf{x}_1 - \bar{\mathbf{x}}_2)\,,$$

where we defined $\bar{\mathbf{x}} = (x, -y)$. This gives an effective factor of two for the boundary-to-boundary propagator ($y_1 = y_2 = 0$).

The non-local action of eq. (A.15) is recovered from (A.17) with the help of a result by Caffarelli and Silvestre [108], according to which a Dirichlet boundary condition is mapped to a Neumann boundary condition with the aid of a fractional Laplacian. In two dimensions, this means that if $\Phi(\mathbf{x})$ satisfies

$$\partial^\mu \partial_\mu \Phi(x,y) = 0\,, \tag{A.21}$$

with Dirichlet boundary condition (A.19), then

$$\lim_{y \to 0^+} \partial_y \Phi(x,y) = -(-\partial_x^2)^{1/2}\phi(x) \equiv \frac{1}{\pi}\int \mathrm{d}x' \frac{\phi(x)-\phi(x')}{|x-x'|^2}\,. \tag{A.22}$$

Therefore, integrating (A.17) by parts and using the equations of motion of $\Phi(\mathbf{x})$, that in the path integral formalism is equivalent to performing the Gaussian integral, we are left with the boundary term

$$S_0[\phi] = \frac{1}{8\pi^2}\int_{-\infty}^{+\infty}\mathrm{d}x_1\mathrm{d}x_2 \frac{(\phi(x_1)-\phi(x_2))^2}{|x_1-x_2|^2}\,, \tag{A.23}$$

that coincides with the $\delta \to 0$ limit of (A.15), and now the Neumann boundary conditions for $\Phi(\mathbf{x})$ are equivalent to the field equations for $\phi(x)$.

By analytic continuation, this connection can be generalized to $\delta > 0$, so that the GFF becomes equivalent to conformal line defect for a free theory in noninteger dimension $D = 2 - \delta$. Although unitarity of the combined bulk and defect system is not guaranteed in fractional dimension (see e.g. [109]), this defect description has the advantage that it is manifestly local.

# B  From continuum 1d LRI to AYK model

We provide here a simple, but not rigorous, derivation of the AYK model, and its Coulomb gas limit, from the continuum long-range Ising model of equation (2.2). First, we rewrite the latter as

$$S[\varphi] = -\frac{c_s}{2s(1-s)}\int_{-\infty}^{+\infty}\mathrm{d}x\mathrm{d}y\,\left(|x-y|^{1-s}-a^{1-s}\right)\partial_x\varphi(x)\partial_y\varphi(y) + \frac{\lambda_4}{4}\int_{-\infty}^{+\infty}\mathrm{d}x\,\left(\varphi(x)^2-\rho^2\right)^2$$
$$-\lambda_1\int_{-\infty}^{+\infty}\mathrm{d}x\,\varphi(x) + \frac{\theta}{2}\int_{-\infty}^{+\infty}\mathrm{d}x\,\left(\partial_x\varphi(x)\right)^2\,, \tag{B.1}$$

where in the first term we have integrated by parts once in $x$ and once in $y$, and we have subtracted a vanishing term (the product of two integrals of total derivatives) so that the limit $s \to 1$ is manifestly going to produce a logarithm. In the potential we have instead introduced $\rho^2 = -\lambda_2/\lambda_4$ (assuming $\lambda_2 < 0$) and discarded a constant term, and we have introduced a constant source ($\lambda_1$) and a short-range term ($\propto \theta$) for full comparison to the original construction of Anderson and Yuval and of Kosterlitz.

We now consider the limit $\lambda_4 \to +\infty$ at finite $\rho$, which is justified by the behavior of the bare couplings near the fixed point, when using a momentum cutoff. In such limit, the potential enforces the condition $\varphi(x) = \pm\rho$, and a generic configuration is of the type drawn in figure 2, which is a continuous version of the blocks of up or down Ising spins. Therefore, the derivative of the field is nonzero (and divergent) only at the points where the sign changes (also known as kinks and anti-kinks). That is, the field derivative configurations are restricted to take the form

$$\partial\varphi_n(x) = \begin{cases} 0, & \text{if } n = 0, \\ \pm\rho\sum_{i=1}^{2n}(-1)^i\delta(x-x_i), & \text{if } n > 0 \text{ and } \frac{L}{2} > x_1 > x_2 > \ldots > x_{2n} > -\frac{L}{2}, \end{cases} \tag{B.2}$$

where the even number of delta functions is due to having assumed that outside the interval $[-\frac{L}{2}, \frac{L}{2}]$ there are no kinks and the field has the same sign on both sides of it.

In order to tame singular expressions, we use a heat kernel regularization for the delta functions, namely, we replace

$$\delta(x) \rightarrow \delta_a(x) \equiv \frac{\sqrt{2}}{a} e^{-2\pi \frac{x^2}{a^2}} , \tag{B.3}$$

where the normalization is chosen such that $\int_{-\infty}^{+\infty} dx\, \delta_a(x)^2 = 1/a$. The latter will thus provide a regularization of the short-range term.

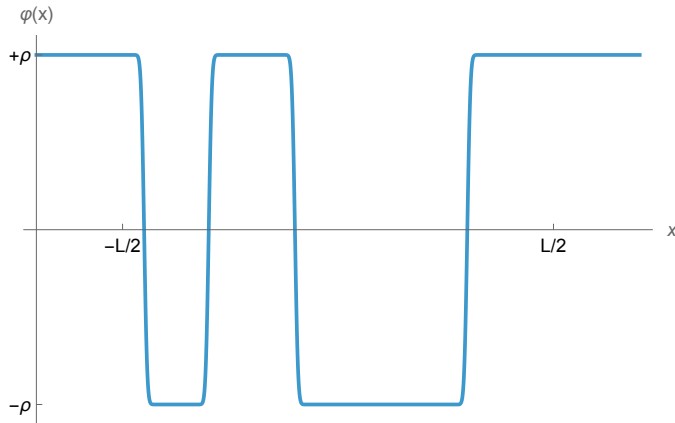

Figure 2: A typical low-temperature configuration of kinks and antikinks in the LRI model, with $n = 2$ and $a/L \sim 10^{-2}$.

Plugging the field configuration (B.2), with regularized delta functions, into the action (B.1), we obtain

$$S[\varphi_n] = -\frac{c_s\, \rho^2}{2s(1-s)} \sum_{i \neq j}^{1...2n} (-1)^{i+j} \left(|x_i - x_j|^{1-s} - a^{1-s}\right) + \lambda_1 \rho \sum_i (-1)^i (x_i - x_{i-1})$$
$$+ 2n\rho^2 \left(\frac{\theta}{2a} + \frac{c_s\, a^{1-s} f(1-s)}{2s(1-s)}\right) , \tag{B.4}$$

where we have discarded terms of order $O(a|x_i - x_j|^{-s})$ relative to those of order $O(|x_i - x_j|^{1-s})$, and we have introduced a function $f(z) = \kappa z + O(z^2)$, which results from the integration over pairs of kinks at $x_i = x_j$.

The resulting action has the same form as the classical Hamiltonian of the AYK model [23], see (2.20), and in the limit $s \rightarrow 1$, it becomes

$$S[\varphi_n] = -\frac{\rho^2}{2\pi} \sum_{i \neq j}^{1...2n} (-1)^{i+j} \log(|x_i - x_j|/a) + \lambda_1 \rho \sum_i (-1)^i (x_i - x_{i-1}) + n\rho^2(\theta/a + \kappa/\pi) , \tag{B.5}$$

which was previously obtained by Anderson and Yuval [22].

In the partition function, we accordingly replace the functional measure by a sum over configurations of the type (B.2), and we arrive at the partition function (2.18) of a Coulomb gas with charges of alternating sign, with the following identifications:

$$u = \exp(-\rho^2(\theta/a + \kappa/\pi)/2) , \quad K = \rho^2/2\pi , \quad H/a = 2\lambda_1 \rho . \tag{B.6}$$

Such relations are nonuniversal, so it is not surprising that in our derivation in the continuum they differ slightly from those obtained by Anderson and Yuval. However, they are qualitatively similar, and in particular they imply that the LRI model is a line in the $\{u, K\}$ plane, parametrized by $\rho^2$, because $\theta$ is an irrelevant coupling.

# C  Alternative formulations and gauging

## C.1  Coherent state representation

The partition function in (3.9) is expressed as mix of path integral and operator pictures. A more homogeneous representation is possible by noticing that the defect operator $\mathcal{D}$ in (3.13) can be thought as an evolution operator with a time-dependent Hamiltonian, if $x$ is interpreted as Euclidean time. This leads to the so-called coherent state representation (see for example [65, 52, 63, 64]), that is, a path integral representation in terms of a complex bosonic spinor $z(x) = \{z_1(x), z_2(x)\}$, subject to the constraint $\bar{z}(x)z(x) = 1$. Following the same construction as in the references above, we can write

$$\operatorname{tr}\mathcal{D} = \int [dz d\bar{z}]\delta(\bar{z}z - 1)e^{-S_{\mathcal{D}}[z,\phi]} \, , \tag{C.1}$$

with

$$S_{\mathcal{D}}[z,\phi] = \int \mathrm{d}x \, \left( \bar{z}(x)\partial z(x) - g\big(\hat{S}_+(x)V_+(x) + \hat{S}_-(x)V_-(x)\big) - h\,\hat{S}_3(x)\,\chi(x) \right) \, , \tag{C.2}$$

where now the spin operators are represented as

$$\hat{S}_\pm(x) = \bar{z}(x)\hat{\sigma}_\pm z(x) \, , \qquad \hat{S}_3(x) = \bar{z}(x)\hat{\sigma}_3 z(x) \, . \tag{C.3}$$

The coherent state representation makes it easier to exploit field equations, which now write

$$\frac{\delta(S_0 + S_{\mathcal{D}})}{\delta\phi} = 0 \quad \Rightarrow \quad C^{-1}\cdot\phi(x) = ig\big(\hat{S}_+(x)V_+(x) - \hat{S}_-(x)V_-(x)\big) - \frac{ih}{\sqrt{2}}\,\partial\hat{S}_3(x) \, . \tag{C.4}$$

In order to eliminate $\partial\hat{S}_3(x)$, and recover (3.30), we can then use a linear combination of the field equations for $z$ and $\bar{z}$:

$$\bar{z}\hat{\sigma}_3\frac{\delta S_{\mathcal{D}}}{\delta\bar{z}} - \frac{\delta S_{\mathcal{D}}}{\delta z}\hat{\sigma}_3 z = 0 \, , \tag{C.5}$$

with which we recover the equivalent of (3.27):

$$\partial_x\hat{S}_3 = 2g(\hat{S}_+V_+ - \hat{S}_-V_-) \, . \tag{C.6}$$

## C.2  Gauging

In the coherent state representation, the $U(1)$ symmetry acts as

$$\phi(x) \to \phi(x) + \alpha/b_0 \, , \qquad z(x) \to e^{i\frac{\alpha}{2}\hat{\sigma}_3}z(x) \, , \tag{C.7}$$

and it could thus be gauged by introducing a gauge field $A(x)$ via the replacements[32]

$$\partial_x\phi(x) \to \partial_x\phi(x) + A(x)/b_0 \, , \qquad \partial_x z(x) \to Dz(x) \equiv (\partial_x + \frac{i}{2}\hat{\sigma}_3 A(x))z(x) \, , \tag{C.8}$$

with the gauge transformation

$$A(x) \to A(x) - \partial_x\alpha(x) \, . \tag{C.9}$$

---

[32]The first replacement requires that we first integrate by part twice in the nonlocal kinetic term:

$$\int_{-\infty}^{+\infty} \mathrm{d}x\mathrm{d}y\frac{(\phi(x) - \phi(y))^2}{|x-y|^{3-s}} = -\frac{1}{(1-s)(2-s)} \int_{-\infty}^{+\infty} \mathrm{d}x\mathrm{d}y(\phi(x) - \phi(y))^2\partial_x\partial_y\frac{1}{|x-y|^{1-s}}$$

$$= \frac{2}{(1-s)(2-s)} \int_{-\infty}^{+\infty} \mathrm{d}x\mathrm{d}y\frac{\partial\phi(x)\partial\phi(y)}{|x-y|^{1-s}} \, .$$

We thus obtain the total action

$$
S_{\text{gauge}}[\phi, A, z, \bar{z}] = \frac{2\,\mathcal{N}_s}{(1-s)(2-s)} \int_{-\infty}^{+\infty} \mathrm{d}x\mathrm{d}y \frac{(\partial\phi(x) + A(x))(\partial\phi(y) + A(y))}{|x-y|^{1-s}}
$$
$$
+ \int \mathrm{d}x \left( \bar{z}(x) D z(x) - g\big(\hat{S}_+(x)V_+(x) + \hat{S}_-(x)V_-(x)\big) - h\,\hat{S}_3(x) \left( \chi(x) + \frac{\mathrm{i}}{\sqrt{2}}A(x) \right) \right) . \tag{C.10}
$$

Since in $d = 1$ gauge fields are non-dynamical, they could in principle be eliminated from the action. In fact $A(x)$ only appears quadratically, so it could be easily be integrated out. This would lead to new terms in the effective action, which however are irrelevant under the RG flow. Alternatively, we can fix the gauge $A(x) = 0$. Either way, the only effect of gauging would be to restrict observables to be gauge singlets. Moreover, we gain the equations of motion of $A(x)$:

$$
\frac{\delta S_{\text{gauge}}}{\delta A(x)} = 0 \quad \underset{A=0}{\Rightarrow} \quad (1 - \sqrt{2}h)\,\hat{S}_3(x) \propto \int_{-\infty}^{+\infty} \mathrm{d}y \frac{\partial\phi(y)}{|x-y|^{1-s}} , \tag{C.11}
$$

that we recognize being the shadow relation between $\hat{S}_3(x)$ and $\chi(x)$.

## C.3   Nonlinear sigma model formulation

It is actually possible to recast the model (3.9) in yet another form, one that makes the gauging look more standard. Introducing the complex field $\mathcal{U}(x)$, subject to the constraint $\bar{\mathcal{U}}(x)\mathcal{U}(x) = 1$, we claim that (3.9) is equivalent to the partition function for the following $U(1)$ nonlinear sigma model:

$$
S_{\text{NLSM}}[\phi] = -\frac{2\,c_{2-s}}{8\pi(1-s)(2-s)b_0^2} \int_{-\infty}^{+\infty} \mathrm{d}x\mathrm{d}y \frac{\bar{\mathcal{U}}(x)\partial_x\mathcal{U}(x)\,\bar{\mathcal{U}}(y)\partial_y\mathcal{U}(y)}{|x-y|^{1-s}}\mathbb{1} - \log\operatorname{Tr}\mathcal{D} , \tag{C.12}
$$

with

$$
\mathcal{D} \equiv \mathcal{P}\exp\left( \int_{-L/2}^{L/2} \mathrm{d}x \left( g\left(\hat{\sigma}_+\mathcal{U}(x) + \hat{\sigma}_-\bar{\mathcal{U}}(x)\right) + \frac{h}{\sqrt{2}\,b_0}\,\hat{\sigma}_3\bar{\mathcal{U}}(x)\partial_x\mathcal{U}(x) \right) \right) . \tag{C.13}
$$

We can indeed solve easily the constraint and identify $\mathcal{U}(x) = e^{\mathrm{i}b_0\phi(x)}$, and the correspondence of the interactions is obvious. For the Gaussian part, we have used again a double integration by parts.

In this formulation it is clear that the fundamental field is $\mathcal{U}(x)$, and not $\phi(x)$, which is introduced to solve the constraint and parametrize the circle. Notice that $\mathcal{M}(x) \equiv \bar{\mathcal{U}}(x)\partial_x\mathcal{U}(x)$ is the Maurer-Cartan form, so it is natural to express the kinetic term with it. In fact, if the theory was local, we would have a noninteracting action that is quadratic both in terms of $\mathcal{M}(x)$ and of $\mathcal{U}(x)$, because $\mathcal{M}(x)^2 = -\partial_x\bar{\mathcal{U}}(x)\partial_x\mathcal{U}(x)$, by virtue of the constraint. In the long-range case instead it is quadratic only in $\mathcal{M}(x)$.

The $U(1)$ symmetry is now

$$
\mathcal{U}(x) \to \mathcal{U}(x)e^{2\mathrm{i}\alpha} , \qquad \hat{\sigma}_i \to e^{-\mathrm{i}\alpha\hat{\sigma}_3}\hat{\sigma}_i e^{\mathrm{i}\alpha\hat{\sigma}_3} , \tag{C.14}
$$

and its gauging can be implemented by introducing a covariant derivative $D_x\mathcal{U}(x) = (\partial_x + 2A(x))\mathcal{U}(x)$, plus the same treatment as above for the $z$ sector.

Notice that the presence of the Pauli matrices sector, as well as the absence of rotation invariance in one dimension, allow the introduction of nontrivial interactions that would otherwise be incompatible with symmetries.

Notice also that the kinetic term in (C.12) differs from the long-range $O(2)$ nonlinear sigma model studied in [93] in a crucial way. The action in [93] involves the product $\partial_x\bar{\mathcal{U}}(x)\,\partial_y\mathcal{U}(y)$, which differs from $-\bar{\mathcal{U}}(x)\partial_x\mathcal{U}(x)\bar{\mathcal{U}}(y)\partial_y\mathcal{U}(y)$ for $x \neq y$. In particular, the latter is quadratic in $\phi$, while the former is not, thus explaining why the $O(2)$ model of [93] has a nontrivial beta function, even without our $g$ and $h$ terms.

# D  Correlators of $V_\pm$ and $\chi$ in the GFF

In this section, we compute some GFF correlators involving the operators

$$V_\pm(x) \equiv \kappa \,:\, e^{\pm i b_0 \phi(x)} : \quad \chi(x) \equiv \frac{i}{\sqrt{2}} \partial_x \phi(x) , \tag{D.1}$$

for generic $\delta$, where the normal ordering was defined in footnote 14, and we assume that $b_0 \to 1$ for $\delta \to 0$.

We recall that correlators between vertex operators read (here and below $x_{ij} \equiv x_i - x_j$ and $a$ is a UV cutoff)

$$\langle V_{n_1}(x_1) \cdots V_{n_m}(x_m) \rangle_0 = \delta_{0,\sum_i n_i} \, \kappa^m \, e^{2 b_0^2 \sum_{i<j} \frac{n_i n_j}{\delta}(|x_{ij}|^\delta - \kappa^{-\delta})} . \tag{D.2}$$

For example, we have

$$\langle V_+(x_1) V_-(x_2) \rangle_0 = \kappa^2 \, e^{-\frac{2b_0^2}{\delta}\left(|x_{12}|^\delta - \kappa^{-\delta}\right)} ,$$

$$\langle V_+(x_1) V_-(x_2) V_+(x_3) V_-(x_4) \rangle_0 = \kappa^4 \, e^{2 b_0^2 \sum_{i<j} \frac{(-1)^{i-j}}{\delta}(|x_{ij}|^\delta - \kappa^{-\delta})} . \tag{D.3}$$

In the $\delta = 0$ limit, we get:

$$\langle V_+(x_1) V_-(x_2) \rangle_0 = \frac{1}{|x_{12}|^2} , \quad \langle V_+(x_1) V_-(x_2) V_+(x_3) V_-(x_4) \rangle_0 = \frac{|x_{13}|^2 |x_{24}|^2}{|x_{14}|^2 |x_{23}|^2 |x_{34}|^2 |x_{12}|^2} . \tag{D.4}$$

The correlation functions of $\chi$ with itself are obtained from correlators of $\phi$ by acting with derivatives, giving:

$$\langle \chi(x_1) \chi(x_2) \rangle_0 = \frac{1-\delta}{|x_{12}|^{2-\delta}} , \quad \langle \chi(x_1) \chi(x_2) \chi(x_3) \rangle_0 = 0 ,$$

$$\langle \chi(x_1) \chi(x_2) \chi(x_3) \chi(x_4) \rangle_0 = (1-\delta)^2 \left( \frac{1}{|x_{12}|^{2-\delta}|x_{34}|^{2-\delta}} + \frac{1}{|x_{13}|^{2-\delta}|x_{24}|^{2-\delta}} + \frac{1}{|x_{14}|^{2-\delta}|x_{23}|^{2-\delta}} \right) . \tag{D.5}$$

To compute mixed correlation functions of $\chi$ with $V_\pm$, one can exploit the $\phi$-dependence of $V_\pm$ and the Wick theorem. To that end, we remind that Wick's theorem gives (in the following formula, $O_i[\phi]$ is a $\phi$-composite)

$$\langle \phi(x) \prod_{i=1}^n O_i[\phi](x_i) \rangle_0 = \sum_{i=1}^n \langle \phi(x) \phi(x_i) \rangle \langle O_1[\phi](x_1) \cdots \frac{\delta O_i[\phi]}{\delta \phi}(x_i) \cdots O_n[\phi](x_n) \rangle_0 , \tag{D.6}$$

which we can iterate. Alternatively, we can use the following trick. Introduce auxiliary vertex operators $V_\alpha =\, : e^{i\alpha\phi} :$, and use the identity:

$$\langle \chi(x_i) \dots \rangle = \left\langle \left( \lim_{\alpha_i \to 0} \frac{\partial_x V_{\alpha_i}(x_i)}{\alpha_i \sqrt{2}} \right) \dots \right\rangle , \tag{D.7}$$

in combination with the general formula (3.6) for correlators of vertex operators.

For mixed three-point functions with $V_\pm$ and $\chi$ we find

$$\langle \chi(x_1) V_\pm(x_2) V_\mp(x_3) \rangle_0 = \pm \sqrt{2} b_0 \left( x_{12}^{\delta-1} - x_{13}^{\delta-1} \right) \langle V_+(x_2) V_-(x_3) \rangle_0 ,$$

$$\langle V_\pm(x_1) V_\mp(x_2) \chi(x_3) \rangle_0 = \mp \sqrt{2} b_0 \left( x_{13}^{\delta-1} - x_{23}^{\delta-1} \right) \langle V_+(x_1) V_-(x_2) \rangle_0 ,$$

$$\langle V_\pm(x_1) \chi(x_2) V_\mp(x_3) \rangle_0 = \mp \sqrt{2} b_0 \left( x_{12}^{\delta-1} + x_{23}^{\delta-1} \right) \langle V_+(x_1) V_-(x_3) \rangle_0 , \tag{D.8}$$

where from now on we assume $x_i > x_{i+1}$ in order to get rid of some signum functions, and because anyway in our model we only need path-ordered correlators.

In the $\delta = 0$ limit we get

$$\langle \chi(x_1) V_\pm(x_2) V_\mp(x_3) \rangle_0 = \frac{\pm\sqrt{2}}{x_{12} x_{13} x_{23}},$$

$$\langle V_\pm(x_1) V_\mp(x_2) \chi(x_3) \rangle_0 = \frac{\pm\sqrt{2}}{x_{13} x_{23} x_{12}}, \quad \langle V_\pm(x_1) \chi(x_2) V_\mp(x_3) \rangle_0 = \frac{\mp\sqrt{2}}{x_{12} x_{23} x_{13}}. \tag{D.9}$$

Mixed four-point functions can be computed analogously. For simplicity, we report here the results in the $\delta = 0$ limit only:

$$\langle \chi(x_1) \chi(x_2) V_+(x_3) V_-(x_4) \rangle_0 = \langle V_+(x_1) V_-(x_2) \chi(x_3) \chi(x_4) \rangle_0 = \frac{1}{x_{12}^2 x_{34}^2} + \frac{2}{x_{13} x_{23} x_{14} x_{24}},$$

$$\langle \chi(x_1) V_+(x_2) \chi(x_3) V_-(x_4) \rangle_0 = \langle V_+(x_1) \chi(x_2) V_-(x_3) \chi(x_4) \rangle_0 = \frac{1}{x_{13}^2 x_{24}^2} - \frac{2}{x_{12} x_{23} x_{14} x_{34}},$$

$$\langle \chi(x_1) V_+(x_2) V_-(x_3) \chi(x_4) \rangle_0 = \langle V_+(x_1) \chi(x_2) \chi(x_3) V_-(x_4) \rangle_0 = \frac{1}{x_{14}^2 x_{23}^2} + \frac{2}{x_{12} x_{13} x_{24} x_{34}}. \tag{D.10}$$

## D.1 Insertions at infinity

Pushing the operator at position $x_1$ to infinity, according to the definition (4.18), the three-point functions (D.8) reduce to

$$\langle \chi(\infty) V_\pm(x_2) V_\mp(x_3) \rangle_0 = \pm\sqrt{2} b_0 \, x_{23} \, \langle V_+(x_2) V_-(x_3) \rangle_0,$$

$$\langle V_\pm(\infty) V_\mp(x_2) \chi(x_3) \rangle_0 = \pm\sqrt{2} b_0 \, x_{23}^{\delta-1},$$

$$\langle V_\pm(\infty) \chi(x_2) V_\mp(x_3) \rangle_0 = \mp\sqrt{2} b_0 \, x_{23}^{\delta-1}. \tag{D.11}$$

The limit $\delta \to 0$ are easily obtained and they coincide with what one obtains by using in (D.9) the standard CFT definition of operator at infinity.

As an example, we give also the case of homogeneous four-point functions:

$$\langle V_+(\infty) V_-(x_2) V_+(x_3) V_-(x_4) \rangle_0 = \kappa^2 \, e^{2b_0^2 \sum_{i<j}^{i,j=2,3,4} \frac{(-1)^{i-j}}{\delta} (|x_{ij}|^\delta - \kappa^{-\delta})} \xrightarrow{\delta \to 0} \frac{|x_{24}|^2}{|x_{23}|^2 |x_{34}|^2},$$

$$\langle \chi(\infty) \chi(x_2) \chi(x_3) \chi(x_4) \rangle_0 = (1 - \delta) \left( \frac{1}{|x_{34}|^{2-\delta}} + \frac{1}{|x_{24}|^{2-\delta}} + \frac{1}{|x_{23}|^{2-\delta}} \right). \tag{D.12}$$

# E  Logarithmic corrections to scaling near crossover

In this appendix, we study the logarithmic corrections to the scaling behavior of the critical 1d LRI Ising theory at the crossover. As discussed in the main text, at $s = 1$ the flow to the IR fixed-point theory at $g = h = 0$ is controlled by a marginally irrelevant operator, a linear combination of $\mathcal{O}_g$ and $\mathcal{O}_h$. A standard argument [110], also employed for the higher-dimensional LRI in appendix B of [9], shows that in the presence of marginally irrelevant operators, the CFT predictions for the IR behavior of correlators at criticality receive logarithmic corrections.

Starting from the $s = 1$ theory, let us turn on the $\mathcal{O}_h$ and $\mathcal{O}_g$ perturbations. The general solution of the the Callan–Symanzik equation for the two-point function of $\sigma(x) = \sigma_3$ is

$$\langle \sigma(r) \sigma(0) \rangle = \frac{c(r)}{r^{2\Delta_\sigma}}, \quad r > 0, \tag{E.1}$$

where $\Delta_\sigma$ is the conformal dimension in the unperturbed theory. Up to an overall function that is constant at leading order, we have that

$$c(r) \propto \exp\left\{ -2 \int_1^r d\log r' \, \gamma_\sigma(\bar{g}(r', g_0, h_0), \bar{h}(r', g_0, h_0)) \right\} . \tag{E.2}$$

In the equation above, $\bar{g}$ and $\bar{h}$ are the running couplings, whereas $g_0, h_0$ are the values of the couplings at $r = 1$, i.e. the starting point of the RG flow. Finally, $\gamma_\sigma$ is the anomalous dimension of $\sigma$, which in 4.4.2 was found to be

$$\gamma_\sigma = 2g^2 . \tag{E.3}$$

The running couplings are the solution to the differential equations

$$-\frac{dg}{d\log r} = \beta_g , \qquad -\frac{dh}{d\log r} = \beta_h , \tag{E.4}$$

where $\beta_g$ and $\beta_h$ are the beta functions of the $s = 1$ theory in eq. (4.12), which here we truncate to the leading non-trivial order, i.e. $\beta_g = -2\sqrt{2}gh$ and $\beta_h - \sqrt{2}g^2$. In this approximation, the quantity $W = 2h^2 - g^2$ is constant along the flow, i.e. $2h^2 - g^2 = 2h_0^2 - g_0^2$, and it parametrizes the deviation from the phase transition line, i.e. the separatrix between regions $I$ and $II$ in fig. 1, panel $(a)$.

The flow equations are easily solved. In the region $I$ (i.e. $W > 0$, $h_0 < 0$, $g_0 > 0$), we find:

$$\bar{g}(r, g_0, h_0) = 2\sqrt{W} \frac{\sqrt{A} e^{-2\sqrt{W}\log r}}{1 - A e^{-4\sqrt{W}\log r}} , \tag{E.5}$$

$$\bar{h}(r, g_0, h_0) = -\sqrt{W} \frac{\left(1 + A e^{-4\sqrt{W}\log r}\right)}{\sqrt{2}\left(1 - A e^{-4\sqrt{W}\log r}\right)} , \tag{E.6}$$

with $A = (\sqrt{2}h_0 + \sqrt{W})/(\sqrt{2}h_0 - \sqrt{W})$.

Tuning to the transition line, $W \to 0$, we obtain

$$\bar{g}^2(r, g_0, h_0) = \frac{4h_0^2}{\left(\sqrt{2} - 4h_0 \log r\right)^2} , \tag{E.7}$$

and thus

$$c(r) \propto \exp\left\{ -4 \int_1^r d\log r' \, \bar{g}^2(r', g_0, h_0) \right\} \propto \exp\left\{ -4 \frac{h_0}{\sqrt{2} - 4h_0 \log r} \right\}$$
$$\simeq \exp\left\{ \frac{1}{\log r} \right\} \simeq 1 + \frac{1}{\log r} + \dots , \tag{E.8}$$

where in the last two steps we used the large $r$ limit. We see that the logarithmic corrections to scaling appear, but only as subleading behavior. However, remembering that $\Delta_\sigma = 0$ and that at the critical temperature the 1d LRI model at $s = 1$ has a non-vanishing magnetization, the logarithmic correction gives a nontrivial result for the connected two-point function, as noticed early on in [111].

It is interesting to also consider the two-point function of vertex operators, as it leads to a more standard outcome. Consider the two-point function of $\mathcal{O}_g$, that at the fixed point $g = h = 0$ of the $s = 1$ theory is simply

$$\langle \mathcal{O}_g(r)\mathcal{O}_g(0)\rangle_0 = \frac{1}{r^2} . \tag{E.9}$$

Turning on $g$ and $h$, this will get corrections, starting at order $h$, and then with quadratic corrections $g^2$ and $h^2$, and so on. On the critical line ($W = 0$), $g$ and $h$ are proportional to each other, hence to leading order we can retain just the linear contribution in $h$. The latter leads to the correction

$$\langle \mathcal{O}_g(r)\mathcal{O}_g(0)\rangle = \frac{1}{r^{2(1-2\sqrt{2}h)}} + O(g^2, h^2) = \frac{1}{r^{2(1-\sqrt{2}h)^2}} + O(g^2)\,, \tag{E.10}$$

where in the second step we used the exact result at $g = 0$, see Section 3.2.1. This show that the anomalous dimension of $\mathcal{O}_g$ is $-2\sqrt{2}h$, in agreement with what we obtain from the beta functions in the leading-order approximation. In order to take into account the effect of $g > 0$, we use again the CS equation, again tuned to the critical line $W = 0$. Using similar formulas as above, but with $\gamma_{\mathcal{O}_g} = -2\sqrt{2}h$, we find

$$c(r) \propto \exp\left\{4\sqrt{2}\int_1^r \mathrm{d}\log r'\, \bar{h}(r', g_0, h_0)\right\} \propto \exp\left\{-2\log(1 - 2\sqrt{2}h_0\log r)\right\}$$
$$\simeq \frac{1}{(1 - 2\sqrt{2}h_0\log r)^2} \sim \frac{1}{(\log r)^2}\,. \tag{E.11}$$

Therefore, also the two-point function of $\mathcal{O}_g$ displays logarithmic corrections to scaling, with exponent $-2$.

# F    One- and two-point functions to $O(\delta)$.

In this appendix, we compute one- and two-point correlation functions with $\sigma, \chi$ and $\mathcal{O}_\pm$. At the IR fixed point the LRI is a 1d CFT, and we check that such correlation functions are consistent with the expected form for correlators of conformal primaries, see Section 2.2.1.

## F.1    One-point functions

The case of $\sigma$ and $\chi$ is trivial: their one-point functions vanish identically to all orders of perturbation theory, as a consequence of $\mathbb{Z}_2$ symmetry.[33]

For $\mathcal{O}_g$, we find that $O(g^2, h)$ terms vanish (by the neutrality condition) and in the $L \to \infty$ limit we have:

$$\langle \mathcal{O}_g(x)\rangle_\mathcal{D} = \frac{g}{2}\int \mathrm{d}y\, \mathrm{tr}\langle \mathrm{P}\mathcal{O}_g(x)\mathcal{O}_g(y)\rangle_0 = \frac{2g}{a} + O(g^3, hg)\,.$$

We can set the one-point function to zero by including a mixing term with the identity, i.e. we define the shifted operator $[\mathcal{O}_g]$ as

$$[\mathcal{O}_g] = \mathcal{O}_g + Z_{g\mathbb{1}}\mathbb{1}\,, \tag{F.1}$$

with $Z_{g\mathbb{1}} = -2g/a + O(g^3, hg)$, so that $\langle[\mathcal{O}_g]\rangle_\mathcal{D} = O(g^3, hg)$.

For $\mathcal{O}_h$ we find:

$$\langle \mathcal{O}_h(x)\rangle_\mathcal{D} = \frac{g^2}{4}\int \mathrm{d}y_1 \int \mathrm{d}y_2\, \mathrm{tr}\langle \mathrm{P}\mathcal{O}_h(x)\mathcal{O}_g(y_1)\mathcal{O}_g(y_2)\rangle_0 + \frac{h}{2}\int \mathrm{d}y\, \mathrm{tr}\langle \mathrm{P}\mathcal{O}_h(x)\mathcal{O}_h(y)\rangle_0$$
$$= 6\sqrt{2}\log 2\,\frac{g^2}{a} + \frac{2h}{a} + O(g^4, hg^2)\,, \tag{F.2}$$

and so we define

$$[\mathcal{O}_h] = \mathcal{O}_h + Z_{h\mathbb{1}}\mathbb{1}\,, \tag{F.3}$$

---

[33] Away from criticality, we should find a symmetric phase where they still vanish, as well as a broken phase with $\langle\sigma\rangle_\mathcal{D} \neq 0$. However, the latter can only be seen in the limit of vanishing $\mathbb{Z}_2$-breaking external field, or from the large-distance behavior of the two-point function. In the $s = 1$ case, we expect symmetry breaking even at the critical temperature, and this is associated to the decoupling of the GFF and $\mathbb{C}^2$ sectors at the fixed point, with the $\mathbb{C}^2$ sector reproducing the zero-temperature 1d SRI physics.

with $Z_{h\mathbb{1}} = -(6\sqrt{2}\log 2\, g^2 + 2h)/a + O(g^4, hg^2)$, so that $\langle[\mathcal{O}_h]\rangle_\mathcal{D} = O(g^4, hg^2)$.

Lastly, we rewrite $\mathcal{O}_\pm$ by replacing the bare $\mathcal{O}_{g,h}$ in (4.27) with (F.1) and (F.3),

$$a^{\Delta_\pm}\mathcal{O}_\pm = \frac{1}{\sqrt{2}}([\tilde{\mathcal{O}}_h] \pm [\tilde{\mathcal{O}}_g]) + \frac{\sqrt{\delta}}{8}[\tilde{\mathcal{O}}_g] + O(\delta)\,, \tag{F.4}$$

and therefore we obtain $\langle\mathcal{O}_\pm(x)\rangle_\mathcal{D} = O(\delta^{3/2})$, where we have used $g \sim \sqrt{\delta}$ and $h \sim \delta$.

## F.2 Two-point functions

We move on to the computation of the two-point functions. We have already computed the anomalous dimensions in Section 4.4, and now focus on the finite part to extract the normalization of the operators, which is needed for the computation of OPE coefficients in Section 4.5.

### F.2.1 For $\sigma$ and $\chi$

Let us start with the two-point function of $\sigma$. Since $\mathrm{tr}\langle\sigma(x)\sigma(0)\rangle_0 = 1$, we have that:

$$\langle\sigma(x)\sigma(0)\rangle_\mathcal{D} = 1 + \frac{g^2}{4}\int \mathrm{d}y_1\,\mathrm{d}y_2\,\mathrm{tr}\langle\mathrm{P}\sigma(x)\sigma(0)\mathcal{O}_g(y_1)\mathcal{O}_g(y_2)\rangle_{0,c} + \frac{h}{2}\int \mathrm{d}y\,\mathrm{tr}\langle\mathrm{P}\sigma(x)\sigma(0)\mathcal{O}_h(y)\rangle_0$$

$$= 1 + 4g^2\log\left(\frac{2a}{x}\right) + O(g^4, hg^2)\,. \tag{F.5}$$

(Above and below, we take $L/2 > x > 0 > -L/2$.) We can remove the UV divergence as $a \to 0$ by defining the renormalized operator

$$[\sigma]_r = Z_\sigma\sigma + \text{other possible mixings}\,, \tag{F.6}$$

with $Z_\sigma = 1 - 2g^2\log(a/L) + O(g^4, hg^2)$. The renormalized two-point function then reads

$$\langle[\sigma(x)]_r[\sigma(0)]_r\rangle_\mathcal{D} = 1 + 4g^2\log\left(\frac{2L}{x}\right) + O(g^4, hg^2)\,. \tag{F.7}$$

In this renormalization scheme, the renormalized correlator does not depend on the UV cutoff, and thus one obtains the Callan–Symanzik equation for the bare correlator:

$$0 = a\frac{\mathrm{d}}{\mathrm{d}a}\langle[\sigma(x)]_r[\sigma(0)]_r\rangle_\mathcal{D} = Z_\sigma^2\left(a\frac{\partial}{\partial a} - \beta_i\frac{\partial}{\partial g_i} - 2\gamma_\sigma\right)\langle\sigma(x)\sigma(0)\rangle_\mathcal{D}\,, \tag{F.8}$$

with the beta functions given in (4.21) and anomalous dimension $\gamma_\sigma \equiv -a\frac{d\log Z_\sigma}{da} = 2g^2$. We can use the Callan–Symanzik equation to resum the large logarithms in the *bare* correlator, and to find the explicit form of the latter at the IR fixed point

$$\langle\sigma(x)\sigma(0)\rangle_\mathcal{D} = \frac{a^\delta\mathcal{N}_\sigma^2}{(x^2)^{\delta/2}} + O(\delta^{3/2})\,, \quad \mathcal{N}_\sigma^2 = 1 + \delta\log 2\,, \tag{F.9}$$

where we have used the perturbative result (F.5) for the normalization factor. Except for the latter, the result above turns out to be valid to all orders, thanks to the Schwinger-Dyson equation (3.30). In fact, combining the latter with (3.27), we have

$$\partial_{x_1}\partial_{x_2}\langle\sigma(x_1)\sigma(x_2)\rangle_\mathcal{D} = -\frac{1}{g^2(b_0 - \sqrt{2}h)^2}\langle C^{-1}\cdot\phi(x_1)\,C^{-1}\cdot\phi(x_2)\rangle_\mathcal{D} \propto \frac{1}{|x_1 - x_2|^{2(1+d/2)}}\,, \tag{F.10}$$

where in the last step we used the fact that the dimension of $\phi$ is not corrected, as usual in long-range models.

We now consider two-point functions involving $\chi$. First, note that there is a non-vanishing mixing with $\sigma$, as

$$\langle\sigma(x)\chi(0)\rangle_{\mathcal{D}} = \frac{g^2}{4}\int \mathrm{d}y_1\,\mathrm{d}y_2\,\mathrm{tr}\langle\mathrm{P}\sigma(x)\chi(0)\mathcal{O}_g(y_1)\mathcal{O}_g(y_2)\rangle_{0,c} + \frac{h}{2}\int \mathrm{d}y\,\mathrm{tr}\langle\mathrm{P}\sigma(x_1)\chi(x_2)\mathcal{O}_h(y)\rangle_0 + O(g^3,gh)\,.$$

$$= \frac{2h}{a} + \frac{2\sqrt{2}g^2\log 2}{a} + O(g^3,gh)\,. \tag{F.11}$$

We can orthogonalize the two operators by defining a shifted $\chi$ as

$$[\chi] = \chi + Z_{\chi\sigma}\sigma + \text{other possible mixings}\,, \tag{F.12}$$

with $Z_{\chi\sigma} = -\frac{2h}{a} - \frac{2\sqrt{2}g^2\log 2}{a} + O(g^4,hg^2)$.

Finally, for the two-point of $\chi$, corrections linear in $g$ and in $h$ are zero due to an odd number of Pauli matrices, while the $O(g^2)$ term

$$\frac{g^2}{4}\int \mathrm{d}y_1\,\mathrm{d}y_2\,\mathrm{tr}\langle\mathrm{P}\chi(x)\chi(0)\mathcal{O}_g(y_1)\mathcal{O}_g(y_2)\rangle_{0,c}\,, \tag{F.13}$$

vanishes, or in other words, the correlator $\int \mathrm{tr}\langle\mathrm{P}\chi(x)\chi(0)\mathcal{O}_g(y_1)\mathcal{O}_g(y_2)\rangle_0$ is totally disconnected. Hence, at the fixed point we have

$$\langle[\chi](x)[\chi](0)\rangle_{\mathcal{D}} = \frac{1-\delta}{(x^2)^{1-\delta/2}} + O(\delta^{3/2})\,, \tag{F.14}$$

which again is exact, up to normalization, because $\chi \propto \partial\phi$ and the dimension of $\phi$ is not corrected.

### F.2.2 For $\mathcal{O}_g$, $\mathcal{O}_h$ and $\mathcal{O}_\pm$

We now want to compute the two-point functions of $\mathcal{O}_\pm$. As a first step, let us consider those of $\mathcal{O}_g$ and $\mathcal{O}_h$. For the diagonal parts of their correlators, we have that (neglecting $O(g^3,gh)$ terms, as well as corrections in $\delta$ to the perturbative terms)

$$\langle\mathcal{O}_g(x)\mathcal{O}_g(0)\rangle_{\mathcal{D}} = \kappa^2 e^{-\frac{2b_0^2\left(|x|^\delta - \kappa^{-\delta}\right)}{\delta}}$$

$$+ \frac{g^2}{4}\int \mathrm{d}y_1\,\mathrm{d}y_2\,\mathrm{tr}\langle\mathrm{P}\mathcal{O}_g(x)\mathcal{O}_g(0)\mathcal{O}_g(y_1)\mathcal{O}_g(y_2)\rangle_{0,c}$$

$$+ \frac{h}{2}\int \mathrm{d}y\,\mathrm{tr}\langle\mathrm{P}\mathcal{O}_g(x)\mathcal{O}_g(0)\mathcal{O}_h(y)\rangle_0$$

$$= \kappa^2 e^{-\frac{2b_0^2\left(|x|^\delta - \kappa^{-\delta}\right)}{\delta}}$$

$$+ \frac{g^2}{x^2}\left(2\log\left(\frac{x}{a}\right)\left(10\log\left(\frac{x}{a}\right) - 3\right) - 10 + \log 4\right) + \frac{4g^2}{a^2}$$

$$+ \frac{4\sqrt{2}h}{x^2}\log\left(\frac{x}{a}\right)$$

$$\langle\mathcal{O}_h(x)\mathcal{O}_h(0)\rangle_{\mathcal{D}} = \frac{1-\delta}{(x^2)^{1-\delta/2}} + \frac{g^2}{4}\int \mathrm{d}y_1\,\mathrm{d}y_2\,\mathrm{tr}\langle\mathrm{P}\mathcal{O}_h(x)\mathcal{O}_h(0)\mathcal{O}_g(y_1)\mathcal{O}_g(y_2)\rangle_{0,c}$$

$$= \frac{1-\delta}{(x^2)^{1-\delta/2}} + \frac{2g^2}{x^2}\left(8\log^2\left(\frac{x}{a}\right) - 2\log\left(\frac{x}{2a}\right)\right)\,. \tag{F.15}$$

The power-law divergent term cancels exactly upon including the mixing term with the identity, as in eq. (F.1). For the off-diagonal piece we have

$$\langle \mathcal{O}_g(x)\mathcal{O}_h(0)\rangle_{\mathcal{D}} = \frac{g}{2}\int \mathrm{d}y\,\mathrm{tr}\langle \mathrm{P}\mathcal{O}_g(x)\mathcal{O}_h(0)\mathcal{O}_g(y)\rangle_0 + O(g^3, gh) = \frac{4\sqrt{2}g\log(x/a)}{x^2} + O(g^3, gh)\,. \quad \text{(F.16)}$$

Turning to the scaling operators $\mathcal{O}_{\pm}$ of equation (F.4), and using the above results, at the IR fixed point we find (neglecting $O\left(\delta^{3/2}\right)$ terms)

$$\langle \mathcal{O}_{\pm}(x)\mathcal{O}_{\pm}(0)\rangle_{\mathcal{D}} = \frac{\mathcal{N}_{\pm}^2}{(x^2)^{\Delta_{\pm}}}\,, \qquad \langle \mathcal{O}_+(x)\mathcal{O}_-(0)\rangle_{\mathcal{D}} = 0\,, \quad \text{(F.17)}$$

with

$$\mathcal{N}_{\pm}^2 = 1 \pm \frac{\sqrt{\delta}}{4\sqrt{2}} + \frac{\delta}{96}\left(96\log 2 - 16\pi^2 - 93\right)\,. \quad \text{(F.18)}$$

In the diagonal correlators $\langle \mathcal{O}_{\pm}(x)\mathcal{O}_{\pm}(0)\rangle_{\mathcal{D}}$ we have resummed the logarithms into an exponent form via an argument based on the Callan-Symanzik equation, as we have done above for $\sigma$. Concerning the mixed two-point function $\langle \mathcal{O}_+(x)\mathcal{O}_-(0)\rangle_{\mathcal{D}}$, in order to ensure that it is indeed zero, up to $O\left(\delta^{3/2}\right)$, the order-$\delta$ term of (F.4) has been fixed to

$$a^{\Delta_{\pm}}\mathcal{O}_{\pm} = \frac{1}{\sqrt{2}}(\tilde{\mathcal{O}}_h \pm \tilde{\mathcal{O}}_g) + \frac{\sqrt{\delta}}{8}\tilde{\mathcal{O}}_g \pm \frac{\delta}{27}\left(48\log 2 - 16\pi^2 - 45 + 96\log^2(a\kappa)\right)\tilde{\mathcal{O}}_g + O(\delta^{3/2})\,. \quad \text{(F.19)}$$

This should in principle be derivable from the beta functions as we did at order $\sqrt{\delta}$ for (4.27), but it would require us to push them to the next order.

# G  The defect description of LRI

As demonstrated in [33], the $p$-dimensional LRI CFT can be realized as a co-dimension $q = D - p$ conformal defect for the free bulk massless scalar field $\Phi$, whihc propagates in the $D$-dimensional bulk. Reference [14] investigated the advantages of this construction to systematically constrain the dynamics of LRI using the defect conformal bootstrap approach [112].

In this defect realization, the bulk is fictitious. The spectrum of the LRI CFT is built out of operators constrained to lie on the $p$-dimensional defect, in the zero transverse spin sector. This latter restriction follows from the fact that the conformal defect description enjoys a manifest $SO(p+1, 1) \times SO(q)$ symmetry, while in the original LRI there is no $SO(q)$ symmetry.

## G.1  The defect description from the $\varphi^4$ formulation

As we discussed in Section 2, the $\varphi^4$ description of the $p$-dimensional LRI is given in terms of a generalized-free scalar field $\varphi$ with quartic interaction – see equation (1.2). The non-local equation of motion implies that, at the IR fixed point, $\varphi, \varphi^3$ form a shadow pair of operators with protected scaling dimensions

$$\Delta_{\varphi} = \frac{p - \epsilon}{4}, \quad \Delta_{\varphi^3} = \frac{p + \epsilon}{4}\,, \quad \text{(G.1)}$$

where we have set $s = p/2 + \epsilon$.

In the defect description, we interpret the field $\varphi(x)$ as a defect mode of a free bulk massless scalar field $\Phi(y, x)$, which propagates in $D$ fractional dimensions ($y$ represent $q$ transverse directions in the bulk, while $x$ are coordinates along the $p$-dimensional defect). The action is[34]

$$S = \int \mathrm{d}^q y\,\mathrm{d}^p x\,\frac{1}{2}(\partial\Phi)^2 + \int \mathrm{d}^d x\,\frac{\lambda_4}{4}\varphi(x)^4\,, \quad \text{(G.2)}$$

_____________
[34]Compared to (1.2), here we have set $\lambda_2 = 0$ in order to reach the IR fixed point.

with the condition that $\Phi(0,x) = \varphi(x)$, so that integrating out the $q$ transverse directions gives back the original non-local action. In this description, we have that $\Delta_\Phi = D/2 - 1 = \frac{p-\epsilon}{4}$, and so the co-dimension of the defect is $q = 1 + \frac{p}{2} - \frac{\epsilon}{2}$.

The 'defect modes' of $\Phi$, $\varphi$ and $\varphi^3$, form a shadow pair. In other words, the bulk-defect OPE of $\Phi$ (in the zero transverse spin sector) at the IR fixed point reads

$$\Phi(y,x) = b_0^{\Phi,+}\psi_0^+(x) + b_0^{\Phi,-}|y|^p\psi_0^-(x) + \dots, \tag{G.3}$$

where the ellipsis denotes contributions from defect conformal descendants. The scaling dimensions of $\psi_0^\pm$ are then protected by the free-bulk equation of motion [112]:

$$\Delta_+ = \frac{p-\epsilon}{4}, \quad \Delta_- = \frac{p+\epsilon}{4}, \tag{G.4}$$

and, up to a normalization, $\varphi$ ($\varphi^3$) can be identified with $\psi_0^+$ ($\psi_0^-$). Other defect operators such as $\varphi^2$, $\varphi^4$, and so on are instead not protected.

It is interesting to note that the extension to the bulk necessarily comes with an ambiguity, corresponding to a change of boundary condition. For example, we are free to represent the same LRI as a defect in a different bulk theory, i.e. with the boundary condition $\Phi(0,x) = \varphi(x)^3$. Compared with the other boundary condition, this would be a dual conformal defect with co-dimension $q' = 4 - q = 3 - \frac{p}{2} + \frac{\epsilon}{2}$ and $(\psi_0^\pm)' = \psi_0^\mp$.

## G.2 The defect description from the crossover for $p = 1$

As we discussed in Section 3, the weakly coupled model near crossover is given in term of a compact generalized-free scalar $\phi(x)$, with dimension $\Delta_\phi = -\delta$, coupled to a two-level system – see equation (3.2). By gauge invariance, $\phi$ is not part of the physical spectrum of the theory. In this description, at the IR fixed point the operators $\sigma(x)$ and $\chi(x) \equiv \frac{i}{\sqrt{2}}\partial_x\phi(x)$ form a shadow pair of protected operators with dimensions

$$\Delta_\sigma = \delta/2, \quad \Delta_\chi = 1 - \delta/2. \tag{G.5}$$

In the defect description, we can interpret $\chi$ as the defect mode of a free massless bulk scalar $\Phi(y,x)$ in co-dimension $q = 3 - \delta$, i.e. we write $\Phi(0,x) = \chi(x)$ and identify $\chi \sim \psi_0^+$ and $\sigma \sim \psi_0^-$. We can view the vertex operators $V_n(x)$ of the description in Section 3.2 as disorder operators that induce a discontinuity of $2\pi n$ in the field $\phi(x)$, corresponding to a delta-function source of magnitude $2\pi n$ in $\partial_x\phi(x)$ at the point $x$. In equations:

$$V_n(x) = \kappa^{n^2} : e^{inb_0\phi(x)} :\sim \kappa^2 e^{inb_0\int_{-\infty}^x dx'\chi(x')}, \quad n \in \mathbb{Z}. \tag{G.6}$$

The action reads

$$S = \int d^q y\, dx \frac{1}{2}(\partial\Phi)^2 + \log \text{tr}\, \text{Pexp}\left\{\int_{-L/2}^{L/2} dx\, \left[g\,\tilde{\mathcal{O}}_g(x) + h\,\mathcal{O}_h(x)\right]\right\}, \tag{G.7}$$

where again $\mathcal{O}_h(x) = \hat{\sigma}_3\chi(x)$ and $\tilde{\mathcal{O}}_g \equiv \hat{\sigma}_+V_+(x) + \hat{\sigma}_-V_-(x)$, and $\hat{\sigma}$'s are the same combinations of Pauli matrices as before.

An even more complicated description is obtained by switching the role of $\psi_0^+$ and $\psi_0^-$. Compared with the other boundary condition, this would be a conformal defect with co-dimension $q' = 4 - q = 1 - \delta$ and $(\psi_0^\pm)' = \psi_0^\mp$.

## G.3 OPE relations

Regardless of which particular description we use, defect OPE coefficients with one defect mode $\psi_0^{(\pm)}$ and any other defect operator are constrained into OPE relations. When a non-local Lagrangian description of LRI – such as (1.2) – becomes available, such OPE relations follow from the non-local equation of motion, see e.g. discussion in [33, 43]. In the realization of $p$-dimensional LRI as a conformal defect, as we now explain, the same OPE relations follow from a non-perturbative argument: bulk locality.[35]

Consider unit-normalized defect primaries (restricted to the zero transverse-spin sector, as usual). For any scalar defect primary $O$ with scaling dimension $\Delta_O$, and a defect primary $T$ with scaling dimension $\Delta_T$ and symmetric traceless spin $J$, the corresponding three-point function takes the form:[36]

$$\langle \psi_i(x_1) O(x_2) T^{(J)}(\theta, \infty) \rangle = \frac{c_{iOT}}{(x_{12})^{(\Delta_i + \Delta_O - \Delta_T)}} P_{\parallel}^{(J)}(\hat{x}_{12}, \theta) , \tag{G.8}$$

(in the following, $\psi_1 \equiv \psi_0^{(+)}$ and $\psi_2 \equiv \psi_0^{(-)}$) with

$$P_{\parallel}^{(J)}(\hat{x}_{12}, \theta) \equiv (-\hat{x}_{12} \bullet I(\hat{x}_3) \bullet \theta)^J , \quad \hat{x}^a \equiv \frac{x^a}{|x|}, \quad I^{ab}(\hat{x}) \equiv \delta^{ab} - 2\hat{x}^a \hat{x}^b . \tag{G.9}$$

The OPE relations for LRI read [113, 14]:

$$c_{2OT} = -\frac{1}{R(a_{\Phi^2})} \frac{\Gamma\left(1 - \frac{p}{2} + \Delta_1\right) \Gamma\left(\frac{J+p-\Delta_1+\Delta_O-\Delta_T}{2}\right) \Gamma\left(\frac{J+p-\Delta_1-\Delta_O+\Delta_T}{2}\right)}{\Gamma\left(1 + \frac{p}{2} - \Delta_1\right) \Gamma\left(\frac{J+\Delta_1+\Delta_O-\Delta_T}{2}\right) \Gamma\left(\frac{J+\Delta_1-\Delta_O+\Delta_T}{2}\right)} c_{1OT}, \tag{G.10}$$

where $R(a_{\Phi^2})$ is the following bulk-dependent factor

$$R(a_{\Phi^2}) \equiv b_0^{\Phi,-}/b_0^{\Phi,+} , \quad \langle \Phi^2(x) \rangle = \frac{a_{\Phi^2}}{|x_\perp|^{D-2}} , \tag{G.11}$$

and $b_0^{\Phi,\pm}$ and $a_{\Phi^2}$ are related by bulk-defect crossing as [14]:

$$(b_0^{\Phi,-})^2 = a_{\Phi^2} \frac{\Gamma(p)\Gamma\left(\frac{q-2}{2}\right)}{\Gamma\left(\frac{p}{2}\right)\Gamma\left(\frac{p+q-2}{2}\right)} , \quad (b_0^{\Phi,+})^2 = 1 - \frac{\Gamma\left(\frac{p+q-2}{2}\right)\Gamma\left(\frac{4-q}{2}\right)}{\Gamma\left(\frac{q}{2}\right)\Gamma\left(\frac{p-q+2}{2}\right)} (b_0^{\Phi,-})^2 . \tag{G.12}$$

For $O = \psi_i$, we find

$$c_{11T} = \kappa_1(\Delta_T, J) c_{12T} , \quad c_{22T} = \kappa_2(\Delta_T, J) c_{12T} , \tag{G.13}$$

with

$$\kappa_1(\Delta_T, J) = -R(a_{\Phi^2}) \frac{\Gamma\left(\frac{4-q}{2}\right) \Gamma\left(\frac{J+\Delta_T}{2}\right) \Gamma\left(\frac{J+p+q-2-\Delta_T}{2}\right)}{\Gamma\left(\frac{q}{2}\right) \Gamma\left(\frac{J+p-\Delta_T}{2}\right) \Gamma\left(\frac{J+2-q+\Delta_T}{2}\right)} ,$$

$$\kappa_2(\Delta_T, J) = -\frac{1}{R(a_{\Phi^2})} \frac{\Gamma\left(\frac{q}{2}\right) \Gamma\left(\frac{J+\Delta_T}{2}\right) \Gamma\left(\frac{J+p-q+2-\Delta_T}{2}\right)}{\Gamma\left(\frac{4-q}{2}\right) \Gamma\left(\frac{J+p-\Delta_T}{2}\right) \Gamma\left(\frac{J-2+q+\Delta_T}{2}\right)} . \tag{G.14}$$

---

[35]See [113, 77, 78] for a more general application of this principle to constraints theories with boundaries and defects, and [114, 115] for more applications of this principle to general QFTs in AdS.

[36]For any symmetric and traceless $SO(p)$ tensor of spin $J$ we define $\mathcal{O}^{(J)}(\theta, x) \equiv \theta^{a_1} \dots \theta^{a_J} \mathcal{O}^{a_1 \cdots a_J}(x), \quad \theta \bullet \theta = 0.$

For the $p = 1$ case, starting with the most general three-point function with $\psi_i$ and two arbitrary local primaries $\mathcal{O}_k$, with quantum numbers $(\Delta_{O_k}, J_k)$

$$\langle \psi_i(x_1)\mathcal{O}_1(x_2)\mathcal{O}_2(\infty)\rangle = \frac{c_{iO_1O_2}}{(x_{12})^{\Delta_i + \Delta_{O_1} - \Delta_{O_2}}}(\text{sign } x_{12})^J, \quad J \equiv J_1 + J_2 \mod 2, \tag{G.15}$$

bulk locality implies OPE relations as (G.10), where $J \to J_i + J_j \mod 2$, $\Delta_O \to \Delta_{O_1}$, and $\Delta_T \to \Delta_{O_2}$ (see [113] for a detailed derivation). From there, taking $\psi_1 \sim \sigma$, $\psi_2 \sim \chi$, as well as $O_1$ and $O_2$ in the set $\{\mathcal{O}_i, \mathcal{O}_j, \mathcal{O}_k, \mathcal{O}_l\}$, we see that, equation (G.10) implies equation (5.116). Similarly, equation (G.13) implies (2.16) for $p = 1$, using $\psi_1 \sim \sigma$, $\psi_2 \sim \chi$ and $T \sim \mathcal{O}$.

**A tower of protected spin-odd operators** For certain special values of the parameters, the gamma functions in (G.14) develop poles and the OPE relations degenerate. For non-integer $q$ this only happens when

$$\Delta_T = p + J + 2n, \quad J \text{ odd}. \tag{G.16}$$

Due to this poles, we must have that $\kappa_1(\Delta_T, J) = \kappa_2(\Delta_T, J) = 0$ and hence $c_{11T} = c_{22T} = 0$, while leaving $c_{12T}$ unconstrained. Hence, all odd-spin operators in $\psi_1 \times \psi_2$ must have protected dimensions as above.

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
