# Peer review of "A strong-weak duality for the 1d long-range Ising model"

_SciPost Physics_

## Round 1 · Referee Report · Connor Behan (Referee 1) · 2025-10-20

Report

In this work, the authors develop a powerful weakly coupled description of the 1d long-range Ising model (LRI) in the region of parameter space where it looks almost short-ranged. In so doing, they resolve a problem raised 8 years ago, concerning a duality for the dimension $d \geq 2$ LRI and how it can be extended to one dimension. This problem is highly non-trivial because the "CFT" governing the critical short-range Ising model (SRI) in $d = 1$ is really the simplest TQFT with two ground states. Coupling the spin field of this SRI to a generalized free theory, which is the naive expectation, fails to generate a sensible flow.

The improved proposal, explained in detail here after appearing in a shorter version of this paper last year, draws inspiration from work by Anderson, Yuval and Kosterlitz (AYK) in the 70s. Their statistical model, which is valid at leading order in the expansion parameter $\delta$, rewrites the LRI as a gas of alternating kinks and anti-kinks. The gas is dilute since the critical point of the LRI at small $\delta$ is close to the ordered phase of the SRI. The same physics can be reproduced in a field theory which involves exponentials of a bosonic field with a shift symmetry. Although this is most often seen for free fields in 2d, the authors consider generalized free fields in 1d and carefully define normal ordering so that the propagator becomes logarithmic at $\delta = 0$. Using Pauli matrices and path ordering to enforce the alternation of kinks and anti-kinks leads to a novel theory which has the symmetries of the LRI plus an extra $U(1)$. A fully predictive framework follows by restricting to the $U(1)$ singlet sector and this is shown to pass many checks.

While some parts of the paper are more fast-paced than others, it is very self-contained. Some aspects which are likely to make it a useful reference for many years are the modern review of the AYK model and the careful discussion of test functions which require a certain number of vanishing derivatives. It is also nice to see that this theory occupies a sort of "sweet spot" for implementing perturbation theory with the analytic bootstrap, being more complicated than the most commonly studied deformations of generalized free fields but more controlled than the LRI near the crossover in $d = 2$. All in all, I have mostly minor changes to suggest.

Requested changes

  1. It looks strange to jump from $\frac{ax + b}{cx + d}$ to another convention in equation (2.5).

  2. The Figure 1 caption says $s = 0$ instead of $s = 1$.

  3. While most lines in Figure 1 are easy to reproduce using the beta functions (2.21), it would be nice to state where the equation of the dashed blue line can be found. In particular, the notation in equation (B.6) is not consistent with the rest of the paper.

  4. Section 2.4 says that references 9 and 10 are about the $d > 1$ LRI but it would be more correct to say $d \geq 2$.

  5. Under (4.9), the dots seem to be terms that vanish as $a \to 0$. Terms that are independent of $a$ and therefore regular are already shown.

  6. Under (4.20), I am not sure why $b^2$ is a more physical parameter than $h$. But it seems that something interesting is going on here. Even though it is easy to check that (4.21) follows from (4.20), the fixed points $(g_, h_)$ can only obtained by using (4.21) and then translating back to the $h$ variable. Getting this directly from (4.20) would fail because when $g_*^2$ and $h_*$ are both $O(\delta)$, the included term $g^2 h$ and the excluded term $g^4$ are equally important. It would be nice to comment on this.

  7. The presentation of equation (4.32) seems a bit strange. This quantity is being considered in order to compute the scaling dimension $\Delta_\sigma$ so it should not be necessary to already know $\Delta_\sigma$ in order to define it.

  8. Given the other novel aspects (matrix degrees of freedom and non-scaling operators for instance) a reader who is new to conformal perturbation theory might come away from Appendix F with the impression that mixing with the identity is also a peculiar feature of this theory. It should be stated that this is not the case. Indeed, other theories also need this to ensure that perturbative corrections to one-point functions vanish. This type of additive renormalization is equivalent to subtracting power-law divergences. Most papers use the latter terminology exclusively but this paper uses both.

  9. Both limits are the same in (5.22) but one should probably be $+i\infty$.

  10. Under (5.61), it seems that $y$ is being used as a shorthand for $\frac{z}{z - 1}$.

  11. There is a typo "whihc" on page 65 and the indentation after equation (G.2) looks like a typo as well.

  12. In section G.1, $s$ is normally defined as $p/2 + \epsilon/2$ and the expressions in equation (G.1) are wrong because they are not defect shadows. Same with (G.4).

  13. When the codimensions $q$ and $q'$ are worked out in section G.1, there is another mistake. We should have $q = \frac{4 - \epsilon - p}{2}$.

  14. In the last paragraph of the paper, it sounds like the implication goes the other way. The vanishing of $c_{11T}$ and $c_{22T}$ is what we know a priori. The fact that the dimension of an odd-spin operator cannot leave the pole (G.16) is then a consequence.

Recommendation

Ask for minor revision

  • validity: -
  • significance: -
  • originality: -
  • clarity: -
  • formatting: -
  • grammar: -

Author:  Dalimil Mazac  on 2025-11-20  [id 6054]

(in reply to Report 1 by Connor Behan on 2025-10-20)

We thank the referee for his careful reading of the manuscript and insightful comments. We provide our responses below, using the same numbering.

  1. In our opinion, the displayed formula for the transformation of the field is natural for the following reason. The transformation $\tfrac{d x-b}{-cx +a}$ is the inverse of $\tfrac{ax+b}{cx+d}$. In general, when a symmetry $g$ acts on a space by $x\mapsto g x$ , the induced action on functions sends $f$ to $f'(x) = f(g^{-1}x)$. For this reason, we have decided to keep the formula as it is, but added a clarifying remark in a footnote.

  2. We fixed this typo in the new version.

  3. We added in the caption of Figure 1 a reference to equation (B.6), whose notation has been fixed.

  4. Fixed.

  5. Fixed.

  6. $b^2$ is a more physical parameter than $h$, because, as remarked above (4.21), among the two couplings $h$ and $b_0$, one of them is a redundant coupling, and only the combination $b_0-\sqrt{2}h$ is an essential coupling. The reason for this goes back to section 3.2.1. We also added a paragraph below equation (4.23), to emphasize that it is the fixed point of $\beta_{b^2}$ that matters, not that of $\beta_h$.

  7. In the new version we have improved the presentation of (4.23) and of few other details in section 4.4.2.

  8. We have added a clarifying sentence between (F.1) and (F.2).

  9. We corrected this typo in the version.

  10. We clarified this in the new version.

  11. Fixed.

  12. We kept $s = d/2+\varepsilon$, but we fixed both typos in previous equations G.2 and G.4.

  13. Fixed.

  14. We modified as follows: By Bose symmetry, for $J$ odd we must have that $c_{11T}=c_{22T}=0$, while leaving $c_{12T}$ unconstrained. Combining with the OPE relations, for non-integer $q$ this condition requires that:

    $$ \Delta_{{T}}=p + J + 2n, \quad J \;\; \text{odd}\,. $$
    Hence, all odd-spin operators in $\psi_1 \times \psi_2$ must have protected dimensions as above.

---

## Round 1 · Referee Report · Slava Rychkov (Referee 2) · 2025-10-31

Report

This paper studies the phase transition the 1D long-range model Ising model - the 1D Ising model with powerlike interactions among spins. When the power decay of interactions is not too strong, the phase transition happens at a nonzero temperature and is described by a nontrivial 1D CFT with powerlike correlations. When the power decay is too strong, there is no phase transition, like for the short-range 1D Ising model. The question is how the transition happens between the two regimes. A similar question in 2D and 3D was addressed by Behan et al (completing old work by Fisher-Nickel and Sak) who found a field theory description, coupling the short-range Ising CFT to a generalized free field (GFF). The paper under review makes a compelling proposal for a solution in 1D, completing old work by Yuval-Anderson and Kosterlitz. Their construction couples the "1D Ising TQFT" to a compactified GFF.

This long paper explains in detail the construction and performs many checks. I am convinced that their construction is correct at a technical level. I do however have some questions about what this construction means and how it should be presented.

I will give a list of questions in the order they occurred to me while reading the paper.

  1. p.4 "Thus, while the generalized..." They have in mind here some construction which does not work. Which is it?

  2. p.5, before Section 1.1 - What's the prediction for what will happen beyond the crossover, for $s>1$? Behan et al had a prediction in this regime as well.

  3. p.8 - Earlier work by S. Ferrara, R. Gatto, and A. F. Grillo, “Properties of Partial Wave Amplitudes in Conformal Invariant Field Theories,” Nuovo Cim. A26 (1975) 226 (see Eq. (3.4)) as well as Belavin, Polyakov and Zamolodchikov (see App.A) should, I believe, be included along with [41].

  4. p.13 - Section 3.1. This central section introduces a crucial field theoretic ingredient - compactified 1D GFF with negative dimension. As this is the first time that this theory is discussed in detail in the physics literature, it would be great to have it marked with the usual QFT labels. Is it a fixed point? Is it a CFT? Is it a gapped or gapless theory? What's the relation of theories with different values of $b_0$? (As of now, the reader has to ask and answer these basic questions themselves.) It would be also great to develop already here an interesting hint currently found in the conclusions, where the parameter $b_0$ is compared to the mass term of the free scalar theory - it may help the reader greatly.

  5. Concerning the mathematical literature, it seems to me the theories are related but not quite identical, since there is no compactification there. The class of observables that the mathematicians consider is more narrow. They deal with the random continuous field $\phi$, but are not so interested in other operators like derivatives of $\phi$ and the vertex operators. There is probably no theorem in those mathematics paper proving the existence of the QFT that the authors use, with all needed properties. So ultimately the existence of that mathematics work does not seem sufficient for tranquilizing the reader that the author's construction makes sense.

  6. p.13 "In fact, the resulting random distribution is almost surely a continuous function (modulo an additive constant). In other words, such GFF preserves shift symmetry." What does "In other words" refer to, to the additive constant or to continuity? In fact, is continuity used anywhere? Is there a simple physics way to see that paths are continuous?

  7. App.A, p.51 "can be defined without introducing any action functional" - is this an important point? Can it also be introduced with an action functional, and if it can, why should we not use the standard physics way rather than the mathematics construction? Is this a specific lesson about 1D GFF, or a general lesson urging us to change the way we should think about Gaussian theories?

  8. Third paragraph on p.51 - the authors cite many mathematics papers, but do any of those refer to their work as "conformal GFF"? If not it would be good to stress that the point of view taken here is a bit different.

  9. What's the physical meaning of the last but one paragraph on p.52?

  10. p.53 "the integral is regular at $x_1\sim x_2$" Is this significant? Of course if $\phi$ is differentiable the integral is convergent, but if it's not, then not necessarily. For random $\phi$'s distributed according to $\exp(-S)$ the integral will be finite almost surely, but this is a tautology.

  11. Section A.2 - I got a bit lost in the discussion of observables. I understood that $<\phi[f]>$ is defined only for $f$'s of zero mean. But what's the meaning of "well-defined quantity" in the phrase "$<(\phi(x)-\phi(y))^2>$ is a well-defined quantity" below (A.14), in the rigorous language of random distributions being used here. Whatever it is, how does this fact follow from the above discussion?

  12. Section 3.2, 1st line "1d continuum model". The word "model" is used in modern QFT for discrete models like the Ising model, apart from a few historically important theories like the Schwinger model. Otherwise continuous theories are referred to as QFTs. Is there a reason the authors don't call (3.9) a QFT? Is there a message that the considered theory somehow falls short of being a QFT?

  13. The coupling to Pauli matrices in (3.9) through the path ordered exponential appears without much ado. In the introduction, the authors describe nicely their construction as coupling a 1D TQFT to a 1D GFF. Is (3.9) the ONLY coupling of these two theories which exists, or is it the only one which works? There seems to be an interesting general lesson here about what it means to couple a 1D TQFT to a 1D QFT, but the authors avoid stating it.

  14. p.16 3rd paragraph "The first justification" - are any others and where in the paper do they occur?

  15. p.16 4th paragraph - the formulation in terms of bosonic spinors - can the authors comment about advantages and disadvantages of different formulations?

  16. It would be great if this section culminated in classifying theory (3.9) as an RG flow originating from some UV theory through some relevant deformation. That's what we do with any QFT, and I hope it's a QFT.

  17. Another question is how their theory differs from the original theory (2.20), and where it lives on the diagram of Fig. 1(b). If it's precisely the same theory, then it would be great to state early on what's the advantage of such a reformulation. If it's a different theory which shares the same fixed point then it's interesting and would be great to know it early on.

  18. p.18 below (3.26), "Interestingly, $D_x O$ can be a descendant..." - I believe what the authors what to say here is that $D_x O$ can be nonvanishing. It's a descendant by definition.

  19. General question/comment for Section 3.2.2: That $O_h$ can be compensated by a shift of $b_0$ - could this be interpreted by saying that some operator of the theory is redundant? But what is the operator which implements the shift of $b_0$? Is there such an operator?

  20. p.19 below (3.32) "the only relevant or marginal operators" Here the authors are treating $O_g$ and $O_h$ as if they were scaling operators, but at the beginning of Section 4 and elsewhere they stress they are not. The way to reduce the reader's confusion would be to develop the already mentioned analogy between $b_0$ and the mass of the free massive scalar, earlier than in the conclusions. Could it be that their flow is then similar to the $\phi^4$ flow in massive free scalar theory in 2d (except of course $\phi^4$ is strongly relevant)?

  21. Where is the relevant perturbation which needs to be tuned to reach the phase transition? In the beta-function calculations it probably drops out because power-law divergences are dropped, but who is it? Is it by any chance the operator which deforms the two-state Quantum Mechanics to weakly gapped (from being topological), while preserving $\mathbb{Z}_2$ invariance? Is this coupling forbidden or does it have to be tuned? In the 2D and 3D work of Behan et al, the thermal perturbation of the 2D Ising CFT had to be tuned to reach the fixed point. It would be great to know if here the situation is similar or different.

  22. p.20 What would the theory flow to if the couplings were turned off?

  23. p.20 "it should be said that the UV completeness of the model is not needed for our purposes, as we are only interested in its IR behavior." I sense this comment is off the mark. UV completeness is not essential, but perturbative renormalizability is. Has this work solved the problem of finding a field theoretic description capable to compute critical exponents to all orders near the long-short crossover point? (If the model is not perturbatively renormalizable, then in what aspects is it better than Kosterlitz's model (2.20)? Can the computations they performed be performed from (2.20)?) Perhaps the authors don't have a rigorous proof of perturbative renormalizability yet, but can they build on their insight about $b_0$ and $m^2$ analogy and give at least intuitive arguments that it is?

This also concerns the discussion in the second paragraph on p.49 in the conclusions where the authors also seem to conflate UV completeness with perturbative renormalizability.

  1. p.24 1st line - where was $\chi=i/\sqrt{2}\partial\phi$ identification made?

  2. 2nd line - which of the two marginal operators is exactly marginal?

  3. Section 5 - can the authors state advantages and disadvantages of the analytic bootstrap approach vs perturbative RG? Could Section 5 be done completely independently from the previous sections? Given Section 5, is perturbative RG obsolete for this model? Or will Section 5 treatment fail at higher powers of perturbing parameter?

I would be grateful to the authors if they could shed light on these questions and incorporate the answers whenever possible into the revised version of the paper, for the benefit of the readers.

Recommendation

Ask for minor revision

  • validity: -
  • significance: -
  • originality: -
  • clarity: -
  • formatting: -
  • grammar: -

Author:  Dalimil Mazac  on 2025-11-20  [id 6055]

(in reply to Report 2 by Slava Rychkov on 2025-10-31)

Warnings issued while processing user-supplied markup:

  • Inconsistency: plain/Markdown and reStructuredText syntaxes are mixed. Markdown will be used.
    Add "#coerce:reST" or "#coerce:plain" as the first line of your text to force reStructuredText or no markup.
    You may also contact the helpdesk if the formatting is incorrect and you are unable to edit your text.

We thank the referee for his careful reading of the manuscript and insightful comments. We provide our responses below, using the same numbering.

  1. We have in mind a simple-minded generalization of the $d\geq 2$ case: coupling a GFF $\chi$ to the Ising spin $\sigma$, that in $d=1$ we identify with the topological operator of the previous paragraph in the paper. This can be represented as the Pauli matrix $\hat\sigma_3$, hence the operator $\sigma\chi$ would be essentially the same as our $\mathcal{O}_h$. We have added some clarification and a footnote on page 4 (see also the fourth paragraph of the conclusions, "There is a loose analogy ...").

  2. We added a comment about $s>1$ in the second bullet of page 5, plus a more detailed discussion at the end of section 4.3.

  3. References added.

  4. We have added some remarks on page 14, hopefully clarifying some of the aspects of the GFF $\phi$.

  5. The main lesson we drew from the mathematical literature is the fact (reviewed in appendix A) that the two-point function $\langle\phi(x)\phi(y)\rangle$ of the GFF $\phi$ with $-1<\Delta_{\phi}<0$ is only defined up to an additive constant. This is the crucial ingredient for compatibility with compactification $\phi(x)\sim\phi(x)+2\pi/b_0$. Besides that, in our opinion the theory is really not so exotic: it is a Gaussian theory, and the computation of arbitrary correlators for the derivatives of $\phi$ or for the vertex operators is completely standard.

  6. On p.13, "In other words" refers to `"modulo an additive constant". We have now removed that sentence from the parenthesis, hopefully making it more clear. We do not specifically use continuity in our work, and we do not have a simple physical explanation for it.

  7. Yes, a GFF can also be introduced with an action functional, as indeed we do in sections A.1 and A.2. The definition of a Gaussian measure from its properties is however more general, as in particular it holds also for theories with a covariance whose inverse is singular. For the GFF the inversion of the covariance is a slightly delicate operation, hence we prefer to postpone it to A.1 and keep the initial discussion of the appendix focused on different aspects. We added a sentence in the first paragraph of the appendix to anticipate that an explicit action will be written later on.

  8. In fact we also did not use the expression "conformal GFF" except for the beginning of the third paragraph on p.51. We have now changed that sentence.

  9. The physical meaning of the last but one paragraph before section A.1 is that in the case of negative scaling dimension the two-point function $\langle\phi(x)\phi(y)\rangle$ is only defined modulo an additive constant (or a higher polynomial in $x$ and $y$, depending on how negative is the dimension).

  10. The regularity of the integral in (A.11) at $x_1\sim x_2$ is of course a classical statement, but we are making it here because often people are surprised by the presence of the squared difference $(\varphi(x_1)-\varphi(x_2))^2$ rather than just the product $\varphi(x_1)\varphi(x_2)$. We are thus simply remarking that the former is well-defined (for sufficiently good functions) at $x_1\sim x_2$, while the latter is not (see also footnote 32 -- formerly footnote 30), in order to provide a simple intuition of why (A.11) is the correct Fourier transform of (A.8) (as explained more precisely around (A.9)).

  11. Below (A.14), "a well-defined quantity" means a quantity that has a precise value and is not defined modulo a constant. That $\langle(\phi(x)-\phi(y))^2\rangle$ respects such criterion follows by expanding the square. In physics language, it is such because $\phi(x)-\phi(y)$ is a shift-invariant observable.

  12. There is no implicit message in our choice of calling the one in equation (3.9) a "model", and honestly we had not given much thought about it, but it is an interesting question. The physics literature is rather inconsistent in the choice of "theory" versus "model", but it seems to us that there are plenty of examples of QFTs that are commonly referred to as models: the majority of those with a person's name (Gross-Neveu model, Thirring model, Nambu–Jona-Lasinio model, Kondo model, Wess-Zumino-Witten model, Sachdev-Ye-Kitaev model, etc), but also typical models with a technical name ($O(N)$ model, non-linear sigma model, sine-Gordon model, etc). In particular, we were probably subconsciously influenced by the similarity of our proposal to general impurity models, and specifically the relation with the Kondo model.

  13. The coupling of the GFF to Pauli matrices in (3.9) is arrived at constructively, in order to reproduce the AYK model: at the end of section 3.1 we notice that the vertex operators of the GFF reproduce the Boltzmann weights of the AYK model, but perturbing the GFF by just a sine-Gordon type of interaction would miss the constraint of having alternating charges. This is then implemented by introducing a non-commuting algebra, represented in terms of Pauli matrices. This is the way we arrive at it, and as far as we could see, it is the only one that works for the 1d LRI problem. Moreover, it is unique if we require $U(1)$ symmetry (see section 3.2.3) and discard operators that are irrelevant at $s=1$. This kind of coupling has appeared before, most notably at $s=1$ our model reduces to the Kondo model: see for example reference [54] (formerly [53]), where also a generalization has been considered with the Pauli matrices replaced by arbitrary spin-$j$ representations of $SU(2)_q$. We do not see at the moment a general lesson to be drawn from this sort of coupling.

  14. "The first justification" can be read as the fact that our model (3.9) passes the initial test of reproducing AYK's partition function and RG flow. Further justification can be found in the agreement between the perturbative computation of CFT data and the bootstrap analysis, the fact that the model (3.9) has the right symmetries corresponding to the 1d LRI, and that it satisfies constraints on the spectrum of operators (protected operators) and on OPE coefficients, which are characteristic of the LRI. We have added a remark at the end of that paragraph (now on p.16-17).

  15. We have added on p.17 some comments about advantages and disadvantages of different formulations.

  16. The way the paper is organized, the RG flow is studied in section 4, so we have added a discussion about the UV fixed point at the end of section 4.3, but also a short comment pointing towards that just before section 3.2.1.

  17. The AYK model in (2.20) is not a QFT, it is a diluted gas model. The main point about our (3.9) is that it is a QFT whose perturbative expansion reproduces the AYK model (as explained in the "The first justification" paragraph discussed in point 14 above). This is similar to the relation between the sine-Gordon model and a Coulomb gas in two dimensions. Our beta functions (4.21) reproduce those of AYK, given in (2.21), and add one subleading term to them, therefore the flow is essentially the same as in Fig.1 (as explained below (4.21)). As stressed in the introduction and conclusions, the advantage of our (3.9) is that it is a QFT, and this allows the systematic computation of CFT data (sec.4), as well as logarithmic corrections to scaling at $s=1$ (App.E).

  18. Fixed.

  19. We show in Section 3.2.1 that the operator implementing the shift of $b_0$ is the operator $\mathcal{O}_h$, and we comment below equation (4.20) that indeed this implies that one of the couplings $b_0$ and $h$ is redundant.

  20. This was an oversight, $\mathcal{O}_g$ and $\mathcal{O}_h$ are marginal at $s=1$, which is what we meant. We corrected the sentence.

  21. As shown in (4.25), the fixed point (4.22) has one relevant ($\mathcal{O}_-$) and one irrelevant ($\mathcal{O}_+$) perturbation. This is also manifest from the flows in figure 1. Therefore, the relevant perturbation which needs to be tuned to reach the phase transition is $\mathcal{O}_-$. Again this is manifest in figure 1, where the LRI is represented by the dashed blue line (remember that the LRI has only one coupling, thus it is a one-dimensional subspace in this two-dimensional plot): the phase transition occurs when the LRI is tuned to the red trajectory separating region $I$ from region $II$.

  22. If the couplings were turned off, the only RG flow would be that associated to the classical mass dimension of $b_0$, which is positive for $\delta>0$, and hence the theory would have a UV fixed point at vanishing $b_0$, while in the IR it would flow to large $b_0$. This is similar to the mass flow in a massive free theory.

  23. We agree that the original sentence could be misleading. We have reformulated the last paragraph before Sec.4.1, discussing also one reason why we expect the model to be renormalizable. We removed also a similar misleading sentence from the conclusions.

  1. We defined $\chi=\frac{i }{\sqrt{2}}\partial \phi$ in eq. (3.10). We added a reference to that equation at the end of section 4.2.

  2. Since in section 4.2 we are discussing the CFT at $s=1$ and $g=h=0$, all the operators in the table (4.17) are marginal. However, only $\mathcal{O}_h$ is "exactly marginal", if by this it is meant that it can be added to the action without inducing a renormalization group flow: indeed such deformation corresponds to the line of fixed points of Fig.1(a).

  3. This is a very interesting question. We do believe that the full crossing symmetry for all four-point functions does completely fix the model, perhaps even at finite $\delta>0$. However, we are not aware of any theorem that would guarantee this even perturbatively in $\delta$. In any case, proceeding to higher orders in $\delta$ will require considering additional external operators. In response to this question, we added two paragraphs in the Conclusions, on page 52-53.

---

## Round 1 · Referee Report · Anonymous (Referee 3) · 2025-11-5

Report

The authors have studied the long-range Ising model (LRI) in one dimension with interaction decaying as $ \frac{1}{r^{1+s}} $. This model shows nontrivial critical behaviour for $ \frac{1}{2} \leq s \leq 1 $. Near s close to $ \frac{1}{2} $, it has a perturbative description around a mean-field theory. On the other hand, near s close to $ 1 $, there was no weakly coupled description available before the work by these authors.

In higher dimensions, it has been proposed that the LRI interpolates between the short-range Ising model (SRI) and mean-field theory. Therefore, the LRI admits a perturbative description in terms of the short-range Ising model coupled to a Gaussian free field. The puzzle in one dimension is that SRI is a trivial TQFT, and it is not clear how to generate an RG flow. The authors propose a field-theoretic description of the 1D LRI in this limit, near s close to $ 1 $.

The paper is well written and clearly explained. The authors give a good introduction to 1D CFT and then introduce their model. The weakly coupled model turns out to be a compact GFF perturbed by a vertex operator, which they explain in detail. Then they carry out a detailed study of this dual description and validate it in two independent ways: (1) by performing a standard perturbative renormalisation of the dual field-theoretic model near $ s = 1 $, and (2) by applying the analytic conformal bootstrap (in 1D) to compute CFT data and show agreement between the two methods. I am convinced by their proposal and would recommend it for publication after addressing a few minor concerns that I have mentioned below.

  1. In Section 5, the authors have shown that correlators involving $\sigma$, $\chi$, $O_+$, and $O_-$ are fixed up to some high order in $\delta = (1 - s)$, essentially up to the order where an infinite set of primary operators first appears. For the first few orders, the crossing equation can be satisfied with only a finite number of primary operators, which is a unique feature of 1D CFT. This analysis is impressive, as the CFT data are uniquely fixed using crossing symmetry and an additional constraint that demands symmetry under the flip $\sqrt{\delta} \to -\sqrt{\delta}$. This ambiguity appears at order $O(\delta^0)$ and, as such, it is not a symmetry of the full finite-$\delta$ theory. Nevertheless, in perturbation theory, this feature enables the authors to isolate the LRI model in the space of theories. I wonder if there is any physical meaning to the invariance of the data under this flip, since if so, imposing it could also help numerically isolate the theory non-perturbatively. It is interesting that once they impose this symmetry, the data also automatically satisfy the expected OPE relations.

  2. Another question is related to analytic functionals. One could have chosen to work with a Regge-bounded basis from the beginning, in which case one would not have to worry about its action on unitary CFT correlators. However, that approach would introduce additional subtractions, and I wonder whether that would lead to additional ambiguities. Therefore, it was important that the authors chose to work with a single-zero functional with a fixed kernel, whose action is not guaranteed to be well-defined on unitary CFT correlators in general. A related issue is that, on the field-theory side, the perturbing operators do not have well-defined UV dimensions. My concern is whether this flow could actually be an irrelevant flow. In my opinion, any such feature would manifest in the correlator when nontrivial infinite sums appear in the expansion. The authors have addressed this in the conclusion, but I wonder if they have further insight on this point.

  3. The vanishing of equation (5.70) does not seem obvious. In particular, the blocks contain unknown parameters, but perhaps the blocks simplify here, and the unknowns are just factored out nicely so that the integration over z can be performed. I think an explanation of this fact will be helpful for the reader.

  4. In equation (5.79), there appears to be a conformal block corresponding to dimension 1. Hence, in principle, there is an ambiguity, and a choice has been made. Could the authors clarify this to understand what assumptions go into fixing the data uniquely?

  5. Below equation (5.61), $y=z/z-1$ is not defined.

  6. Given the data at order $O(\delta^0)$, can one predict that operators with $\Delta = n$ will appear in the $\sigma \times \sigma$ OPE at $O(\delta^2)$, or is this taken as an input?

  7. This is more of a curiosity. The proposed model possesses a $U(1)$ global symmetry, and the $U(1)$ singlets are identified with the operators of the LRI model. However, these singlets must also satisfy crossing equations involving charged operators, even though such charged operators are not visible within the LRI model itself. Does this imply that, in a numerical bootstrap study, including these additional crossing equations could lead to stronger constraints for the singlet sector? Or should one disregard them altogether when isolating the LRI sector?

Recommendation

Ask for minor revision

  • validity: top
  • significance: top
  • originality: top
  • clarity: top
  • formatting: perfect
  • grammar: perfect

Author:  Dalimil Mazac  on 2025-11-20  [id 6056]

(in reply to Report 3 on 2025-11-05)

We thank the referee for their careful reading of the manuscript and insightful comments. We provide our responses below, using the same numbering.

  1. The physical reason for the invariance of the CFT data under the $2\pi$ monodromy around $\delta = 0$ is the presence of an additional $\mathbb{Z}_2$ symmetry of the undeformed theory at $\delta = g = h =0$. This symmetry is explicitly broken by $\mathcal{O}_g$ and is discussed in more detail below eq.~(5.13). This symmetry is fully analogous to the invariance of correlation functions of the Ising model in a nonzero magnetic field under the flip of the sign of the magnetic field, accompanied by the sign flip of all spins. We do not know how to use this invariance in a non-perturbative numerical bootstrap setting because the latter would presumably operate at a fixed value of $\delta$, whereas the invariance only says something about what happens to the data as $\delta$ is varied.

  2. We agree with this interesting comment but we do not have any additional insights on this point.

  3. Indeed, as the referee correctly guesses, the unknown parameters only appear through a common prefactor, multiplying the function $(2z-1)/z$. The latter function satisfies eq.~(5.70). We added a sentence to that effect in the new version.

  4. Indeed, prior to fixing undetermined parameters, the RHS appears to contain a conformal block of $\Delta=1$. However, the LHS shows that such a block must be absent. The vanishing of its coefficient on the RHS then imposes a constraint on the undetermined parameters, as discussed immediately after eq.~(5.79). In other words, the vanishing of this coefficient is not a new assumption, but necessarily follows.

  5. We have fixed this issue in the new version.

  6. Here the referee raises a good point that was somewhat glossed over in our submission. Here is the complete argument. The four-point function $\langle\sigma\sigma\sigma\sigma\rangle$ is invariant under the monodromy symmetry mentioned above, so its expansion proceeds in integer powers of $\delta$. The question asks how do we know that the operators $\mathcal{P}$ with $n\geq 2$ make no contribution already at $O(\delta)$, i.e.~why do we have $c^{(1)}_{\sigma\sigma\mathcal{P}} = 0$. This claim is proved in Section 5.2 using the $\langle\sigma\sigma\mathcal{O}_a\mathcal{O}_b\rangle$ crossing equation, but only for operators which appear in the $\mathcal{O}_a\times\mathcal{O}_b$ OPE at $\delta = 0$. While this is an infinite set of operators, it does not exhaust all operators of the undeformed theory.

    To obtain a general argument, we need to consider the $\langle\sigma\sigma\sigma\sigma\rangle$ crossing equation at $O(\delta)$. The crucial point is that the total contribution of $1$, $\mathcal{O}_{+}$, and $\mathcal{O}_-$ to $\langle\sigma\sigma\sigma\sigma\rangle$ at $O(\delta)$ is already crossing symmetric on its own. We can then use the analytic functionals $\omega^{0,0,0,0}_n$ to solve for the sum of $(c^{(1)}_{\sigma\sigma\mathcal{P}})^2$ over all $\mathcal{P}$ with $\Delta^{(0)}_{\mathcal{P}} = n \geq 2$, analogously to what is done in Section 5.3.2 for $(c^{(2)}_{\sigma\sigma\mathcal{P}})^2$. The difference is that at $O(\delta)$, $\mathcal{G}^{s,L}(1-z)-\mathcal{G}^{s,L}(z) = 0$, and applying the analytic functionals gives vanishing total squared OPE, which was to be proved.

    We added clarifying comments in the new version.

  7. The $\mathrm{U}(1)$ symmetry of the model should truly be viewed as a gauge symmetry, meaning that correlation functions of charged operators are not well-defined. The reason is that correlation functions involving charged operators depend not only on the combination $b_0-\sqrt{2}h$, but rather on $b_0$ and $h$ separately. To describe the critical LRI, the combination $b_0-\sqrt{2}h$ is at an RG fixed point, but the couplings $b_0$ and $h$ are not separately at a fixed point. Therefore, correlators of charged operators are not scaling operators. Consequently, it does not seem possible to include them in the conformal bootstrap analysis. A new paragraph that we added below equation (4.23) should also help clarify this point.

---

## Round 2 · Author Response

We have corrected the mistake after eq. (G.1) pointed out by Referee 1.

---

## Round 2 · List of Changes

page 68: corrected expression for $s=(p+\epsilon)/2$

---

## Editorial Decision

refereeing_in_preparation